# Towards Federated Reinforcement Learning Free of Problem-Parameter-Based Tuning

## Abstract

Federated reinforcement learning (FRL) has emerged as a promising framework for decentralized decision-making in privacy-sensitive environments. However, its practical deployment is often impeded by the need for extensive hyperparameter tuning, a challenge further exacerbated by non-stationary data distributions and environment heterogeneity across agents. To overcome these limitations, we propose a **problem-parameter-free FRL framework** that eliminates manual stepsize selection through a novel integration of adaptive strategies and momentum-based updates. To further address environment heterogeneity, we introduce carefully designed control variates at both the client and server levels. Building on this framework, we develop two algorithms: **PFEDPG-VR**, which integrates an adaptive scheme into variance-reduced policy gradient updates, and **PFEDPG-HA**, which refines the approach using Hessian-aided corrections. Through rigorous theoretical analysis, we prove that both algorithms achieve state-of-the-art convergence rates, with a sample complexity of $\mathcal{O}(\epsilon^{-3})$ and a communication complexity of $\mathcal{O}(\epsilon^{-2})$. In addition, they enjoy linear convergence speedups with respect to the number of agents and local update steps at each global round. Notably, our methods eliminate the reliance on environment heterogeneity bounds required in previous FRL approaches, significantly broadening their applicability. Extensive experiments on benchmark FRL tasks further demonstrate the superior performance and robustness of our proposed algorithms compared to existing baselines.

## 1 Introduction

Federated learning (FL) has regarded as a key framework for enabling collaborative model training across decentralized agents while preserving data privacy (McMahan et al., 2017; Cheng et al., 2023). Driven by the rising demand for intelligent decision-making across decentralized systems, federated reinforcement learning (FRL) has emerged as a promising extension of FL, enabling distributed agents to jointly learn policies without sharing their raw trajectories (Qi et al., 2021; Jin et al., 2022; Wang et al., 2024a). FRL addresses the data inefficiency of traditional reinforcement learning (RL) and has shown strong potential in privacy-critical and distributed environments such as robotics (Liu et al., 2019), autonomous driving (Liang et al., 2022), and intelligent IoT control (Yu et al., 2020a).

However, a fundamental challenge in FRL lies in the spatio-temporal non-stationarity of local data distributions. Unlike supervised FL, where each client learns from a fixed dataset, FRL requires agents to interact with their environments and collect data continuously. As each agent updates its policy locally, the underlying data distribution shifts—leading to non-stationary and inconsistent trajectories across agents and training rounds. Due to this instability, most existing FRL methods are designed for relatively simple settings. Some assume all agents interact with the same environment (Fan et al., 2021; Khodadadian et al., 2022). Others allow different environments but require them to be structurally similar (Jin et al., 2022; Xie & Song, 2023; Wang et al., 2023a). These assumptions simplify training but limit generalization to more realistic cases. Recently, Wang et al. (2024a); Yue et al. (2024) introduced momentum-based methods designed to handle more complex scenarios. These approaches achieve a sample complexity of $\mathcal{O}(\epsilon^{-3})$ even in settings where agents exhibit significant differences in reward functions, state transition kernels, or initial state distributions.

Despite recent advancements, current FRL methods face critical limitations, as their performance heavily relies on carefully tuned hyperparameters, such as learning rates and momentum coefficients. This tuning process often requires substantial computational resources and detailed knowledge of

problem-specific parameters, which are typically unknown. For instance, selecting appropriate learning rates often depends on access to smoothness constants, heterogeneity bounds, stochastic gradient variances, importance weight variances, and initial optimality gaps. Table 1 (Section 5) illustrates how the learning rates of existing methods are tightly coupled with these unknown parameters, complicating practical deployment and limiting adaptability in diverse real-world scenarios.

The inherent properties of FRL significantly amplify the difficulty of hyperparameter tuning. First, local data distributions in FRL evolve unpredictably as agents interact with their environments, requiring hyperparameters that can adapt to these dynamic changes (Yue et al., 2024). Second, agents often face substantial heterogeneity in reward functions, transition dynamics, and exploration behaviors (Jin et al., 2022). Such diversity demands hyperparameters that remain effective across a broad spectrum of scenarios. Third, identifying such problem-dependent settings is computationally intensive and often impractical due to the distributed nature of data and the privacy-preserving constraints of FRL (Qi et al., 2021). These challenges not only hinder practical deployment but also raise barriers to adoption, underscoring the urgent need for problem-parameter-free FRL algorithms capable of automatically adapting to dynamic and heterogeneous environments without manual intervention.

These limitations raise a critical question:

*Can we design an FRL framework that adapts automatically to arbitrary heterogeneous environments while eliminating the need for manual hyperparameter tuning?*

## 1.1 MAIN CONTRIBUTIONS

In this paper, we provide an affirmative answer to this question by introducing a problem-parameter-free FRL framework that combines adaptive stepsize strategies with momentum-based updates. The adaptive stepsize mechanism dynamically adjusts learning rates based on the optimization landscape, ensuring efficient adaptation. At the same time, momentum-based updates accelerate optimization by promoting consistent gradient alignment across agents and iterations. To address the challenges posed by environmental heterogeneity, we integrate carefully designed control variates into the framework. These designs allow learning rates to be determined solely from observable training constants, eliminating the need for explicit knowledge of problem-specific parameters or trial-and-error tuning. Our primary contributions are as follows:

- We propose a novel problem-parameter-free FRL framework that operates independently of problem-specific parameters, such as smoothness and Lipschitz constants, bounded policy gradient norms, or gradient variance estimates. All stepsizes in our framework are automatically configured using only system-level constants, including the number of agents, the number of local updates, and the total communication rounds. Furthermore, the framework is specifically designed for FRL environments with arbitrary heterogeneity, where agents may vary in their reward structures, transition dynamics, and initial state distributions. Based on this framework, we propose two problem-parameter-free FRL algorithms: PFEDPG-VR, which integrates variance-reduced policy gradient updates, and PFEDPG-HA, which incorporates Hessian-aided corrections.

- Through rigorous theoretical analysis, we prove that both PFEDPG-VR and PFEDPG-HA achieve state-of-the-art convergence results (e.g., (Wang et al., 2024a)). Specifically, they attain an optimal sample complexity of $\mathcal{O}(\epsilon^{-3})$ and a communication complexity of $\mathcal{O}(\epsilon^{-2})$ to reach $\|J(\theta)\| \leq \epsilon$ accuracy. Moreover, both algorithms exhibit linear acceleration with respect to the number of local updates and participating agents. Importantly, our approaches address the limitations of conventional methods that rely on heterogeneity constraints (e.g., (Jin et al., 2022; Xie & Song, 2023; Wang et al., 2023a)), diminishing stepsizes (e.g., (Wang et al., 2019; Karimireddy et al., 2020; Yang et al., 2021)), or achieve only suboptimal convergence rates (e.g., (Fan et al., 2021)). This represents a significant advancement in FRL, offering both theoretical rigor and practical scalability.

- We validate our theoretical findings through extensive empirical studies, demonstrating that the proposed algorithms consistently outperform state-of-the-art FRL baselines, including PAVG (Jin et al., 2022), FEDSVRPG-M, and FEDHAPG-M (Wang et al., 2024a). Furthermore, the experimental results show that our proposed methods maintain robust performance across a diverse range of hyperparameter settings, highlighting their adaptability and effectiveness in practical scenarios.

## 2 PROBLEM SETUP

**Reinforcement Learning (RL).** A standard RL task is modeled as a discrete-time Markov Decision Process (MDP), defined by $\mathcal{M} = (\mathcal{S}, \mathcal{A}, \mathcal{P}, \mathcal{R}, \gamma, \rho)$, where $\mathcal{S}$ is the state space, $\mathcal{A}$ is the action

space, and $\rho$ represents the initial state distribution. The transition kernel $\mathcal{P}(\cdot \mid s, a)$ specifies the probability of transitioning from state $s$ to state $s'$ after taking action $a$. The reward function $\mathcal{R} : \mathcal{S} \times \mathcal{A} \rightarrow [0, R_{\max}]$ maps each state-action pair to a bounded reward, where $R_{\max} > 0$. The discount factor $\gamma \in (0, 1)$ controls the trade-off between immediate and future rewards.

A stationary stochastic policy $\pi_\theta : \mathcal{S} \rightarrow \Delta(\mathcal{A})$, parameterized by $\theta \in \mathbb{R}^d$, defines a probability distribution over actions given a state. Executing a policy $\pi_\theta$ yields a trajectory $\tau = (s_0, a_0, s_1, a_1, \ldots, s_H)$, where $H$ is the finite horizon. The cumulative discounted reward of a trajectory $\tau$ is given by $\mathcal{R}(\tau) = \sum_{h=0}^{H-1} \gamma^h \mathcal{R}(s_h, a_h)$. The objective of RL is to learn a policy $\pi_\theta$ that maximizes the expected return:

$$\max_\theta \mathbb{E}_{\tau \sim (\pi_\theta, \mathcal{P}, \rho)} \left[ \sum_{h=0}^{H-1} \gamma^h \mathcal{R}(s_h, a_h) \right], \tag{1}$$

where the expectation is taken over trajectories sampled according to the initial distribution $\rho$, the policy $\pi_\theta$, and the transition dynamics $\mathcal{P}$.

**Federated Reinforcement Learning (FRL).** FRL extends the setup in Eq. (1) to decentralized settings, where $N$ distributed agents aim to collaboratively learn a shared policy without exchanging their raw trajectories. Each agent interacts with its own environment and shares only model updates or aggregated information with a central coordinator. Nevertheless, environment heterogeneity is the key challenge: in practice, agents may face different reward functions, state transition kernel, or initial-state distributions (Wang et al., 2024a). For instance, an audio-streaming platform uses FRL to train a global subscription pricing policy that balances diverse listening behaviors, while autonomous vehicles rely on FRL to learn a unified driving policy that generalizes across a wide range of environments—from congested city streets to rural highways—by aggregating distributed driving experiences. A shared policy that bridges such variations not only captures the common structural patterns across environments but could also serve as a strong initialization for fine-tuning in specialized contexts, as explored in meta- and few-shot RL (Finn et al., 2017; Yu et al., 2020b). Despite this, most FRL research (Fan et al., 2021; Khodadadian et al., 2022; Xie & Song, 2023) assumes relatively homogeneous settings, such as a single shared environment or structurally similar environments, whereas real-world applications typically exhibit far greater diversity across agents.

We now formalize the notion of heterogeneity in the $N$-agent FRL setting. Each agent $i \in [N]$ is associated with its own MDP: $\mathcal{M}_i = (\mathcal{S}, \mathcal{A}, \mathcal{R}^{(i)}, \mathcal{P}^{(i)}, \gamma, \rho^{(i)})$, where $\mathcal{P}^{(i)}$ is the local transition probability, $\mathcal{R}^{(i)}$ is the agent-specific reward function, and $\rho^{(i)}$ denotes the initial state distribution. Despite differences in transitions, rewards, and initial states, all agents share a common state space $\mathcal{S}$ and action space $\mathcal{A}$.

The overarching goal of FRL in this heterogeneous setting is to collaboratively learn a shared policy $\pi_\theta$ that achieves uniformly strong performance across environments of all agents. Crucially, to preserve data privacy, agents are prohibited from exchanging raw state, action, or reward information; only aggregated model updates are communicated. Formally, the optimization problem is

$$\max_\theta \left\{ J(\theta) \triangleq \frac{1}{N} \sum_{i=1}^N J_i(\theta) \right\}, \tag{2}$$

where the local objective $J_i(\theta)$ for agent $i$ is defined as

$$J_i(\theta) = \mathbb{E} \left[ \sum_{h=0}^{H-1} \gamma^h \mathcal{R}^{(i)}(s_h, a_h) \mid s_0 \sim \rho^{(i)}, a_h \sim \pi_\theta(\cdot \mid s_h), s_{h+1} \sim \mathcal{P}^{(i)}(\cdot \mid s_h, a_h) \right].$$

This formulation captures the core challenge of FRL under heterogeneous, decentralized, and privacy-constrained environments. To optimize the shared policy across diverse agents, we build upon the standard policy gradient framework and address the distribution mismatch introduced by local policy updates through importance sampling (IS), as formalized below.

**Policy Gradient.** Denote the probability of a trajectory $\tau^{(i)} = (s_0, a_0, s_1, a_1, \ldots, s_H)$ under a policy $\pi_\theta$ as $p^{(i)}(\tau^{(i)} \mid \theta) = \rho^{(i)}(s_0) \prod_{h=0}^{H-1} \mathcal{P}^{(i)}(s_{h+1} \mid s_h, a_h) \pi_\theta(a_h \mid s_h)$. By applying the likelihood-ratio trick, the gradient of $J_i(\theta)$ can be rewritten as $\nabla J_i(\theta) = \mathbb{E}_{\tau^{(i)} \sim p^{(i)}(\cdot \mid \theta)} \left[ \left( \sum_{h=0}^{H-1} \nabla_\theta \log \pi_\theta(a_h \mid s_h) \right) \mathcal{R}^{(i)}(\tau^{(i)}) \right]$. To estimate $\nabla J_i(\theta)$ from sampled trajectories, we define an unbiased gradient estimator $g_i(\tau^{(i)} \mid \theta)$, following standard approaches such as REINFORCE (Williams, 1992) or GPOMDP (Baxter & Bartlett, 2001). In this paper, we adopt

$$g_i(\tau^{(i)} \mid \theta) = \sum_{t=0}^{H-1} \left( \sum_{h=t}^{H-1} \gamma^h \mathcal{R}^{(i)}(s_h, a_h) \right) \nabla_\theta \log \pi_\theta(a_t \mid s_t).$$

**Importance Sampling.** In FRL settings, local policy updates cause the sampling distributions to drift over time. As a result, for two policies $\theta$ and $\theta'$, the relation $\mathbb{E}_{\tau^{(i)} \sim p^{(i)}(\cdot|\theta)} \left[ g_i \left( \tau^{(i)} \mid \theta \right) - g_i \left( \tau^{(i)} \mid \theta' \right) \right] \neq \nabla J_i(\theta) - \nabla J_i(\theta')$ generally holds, since trajectories are collected under different distributions. To correct for this discrepancy, we introduce IS weight, which is defined as $w^{(i)} \left( \tau^{(i)} \mid \theta', \theta \right) = \frac{p^{(i)}\left(\tau^{(i)}|\theta'\right)}{p^{(i)}\left(\tau^{(i)}|\theta\right)} = \prod_{h=0}^{H-1} \frac{\pi_{\theta'}(a_h|s_h)}{\pi_\theta(a_h|s_h)}$. Using these weights, we can ensure the unbiased relation $\mathbb{E}_{\tau^{(i)} \sim p^{(i)}(\cdot|\theta)} \left[ g_i \left( \tau^{(i)} | \theta \right) - w^{(i)} \left( \tau^{(i)} | \theta', \theta \right) g_i \left( \tau^{(i)} | \theta' \right) \right] = \nabla J_i(\theta) - \nabla J_i(\theta')$.

Existing FRL algorithms critically depend on problem-specific parameters to set appropriate stepsizes, such as the Lipschitz constant, bounds on the policy log-density gradients, bounded policy gradient norms, and the variance of stochastic gradients. However, accurately estimating these parameters is highly impractical in federated settings. The smoothness constants reflect global properties of the optimization landscape that require access to full environment information, which is incompatible with privacy constraints. Similarly, bounding policy log-gradients or gradient norms would necessitate detailed knowledge of local dynamics, and estimating gradient variance would introduce significant sampling and communication overhead. As a result, reliance on such parameters not only complicates algorithm design but also impedes the scalability and practicality of FRL systems.

In addition to problem-parameter dependency, many existing methods further restrict the level of environment heterogeneity by assuming bounded differences in policy gradients, transition dynamics, and reward functions (Jin et al., 2022; Xie & Song, 2023; Wang et al., 2023b; 2024b). While these assumptions simplify theoretical analysis, they substantially limit the practical applicability of FRL in diverse real-world settings. Furthermore, such methods typically ensure convergence only to a neighborhood of a stationary point, with the neighborhood size increasing alongside the degree of heterogeneity. Motivated by these challenges, we develop the problem-parameter-free FRL framework that eliminates the need for problem-specific parameter tuning and naturally accommodates arbitrarily heterogeneous environments, as introduced in the next section.

## 3 ALGORITHM DEVELOPMENT

In this section, we present the design of our problem-parameter-free FRL framework and its two algorithmic instantiations. The first variant, PFEDPG-VR, incorporates a first-order variance-reduction (VR) technique (Gargiani et al., 2022) into the framework to improve convergence speed and reduce gradient variance. To enhance robustness against non-stationary dynamics and gradient variability, we further develop the Hessian-aided (HA) algorithm PFEDPG-HA by integrating stochastic Hessian-vector products (Shen et al., 2019). We rigorously prove that both algorithms converge to an $\epsilon$-stationary point of problem (2) at a state-of-the-art rate.

### 3.1 ALGORITHM DEVELOPMENT OF PFEDPG-VR

We first describe the development of the VR-based problem-parameter-free FRL algorithm (PFEDPG-VR). The detailed procedure is outlined in Algorithm 1. Our algorithm integrates the techniques of control variates, momentum, and local adaptive stepsizes into a unified federated framework. This design ensures stable and efficient policy optimization under arbitrary environment heterogeneity, without the need for manual hyperparameter tuning. Below, we elaborate on these key components.

**Control Variates.** At each global round $t$, agent $i \in [N]$ initializes its local policy model from the shared parameter $\theta_t$ and performs $K$ local updates based on the trajectories $\left\{ \tau_t^{(i,k)} \right\}$ sampled from its own environment. To mitigate variance and stabilize learning, we introduce a local control variate defined as $c_t^{(i)} = \frac{1}{K} \sum_{k=0}^{K-1} g_i \left( \tau_t^{(i,k)} \mid \theta_t^{(i,k)} \right)$, where each agent averages gradients computed from $K$ local updates. The server then updates the global control variate by aggregating the local changes: $c_t = c_{t-1} + \frac{1}{N} \sum_{i=1}^{N} \left( c_t^{(i)} - c_{t-1}^{(i)} \right)$. A correction term $c_{t-1} - c_{t-1}^{(i)}$ is incorporated into the local update to adjust for the discrepancy between global and local gradient statistics. Since $c_t$ captures the global evolution of agent updates, this mechanism effectively reduces agent drift and ensures stability under severe heterogeneity. Theoretically, the control-variate mechanism ensures that the gap between local gradients and the global gradient diminishes over communication rounds, effectively driving the heterogeneity-induced bias to vanish asymptotically. This property allows the analysis to control the optimization error without requiring explicit bounds on differences in rewards, transition kernels, or initial-state distributions across agents, and is therefore essential for guaranteeing convergence under arbitrarily heterogeneous environments.

---

**Algorithm 1** Description of PFEDPG-VR

---

1: **Input:** Initial model $\theta_0$, control variates $c_{-1}^{(i)} = \frac{1}{K} \sum_{k=0}^{K-1} g_i\left(\tau_{-1}^{(i,k)} \mid \theta_0\right)$ for all $i$, aggregated control variates $c_{-1} = \frac{1}{N} \sum_i c_{-1}^{(i)}$, momentum $u_{-1} = c_{-1}$, local learning rate $\eta$, global learning rate $\lambda$, momentum parameter $\beta$

2: **for** $t = 0, 1, \ldots, T-1$ **do**

3:    **for** each agent $i \in [N]$ **do**

4:       Initialize local model: $\theta_t^{(i,0)} = \theta_t$

5:       **for** $k = 0, 1, \ldots, K-1$ **do**

6:          Sample trajectory: $\tau_t^{(i,k)} \sim p^{(i)}(\tau \mid \theta_t^{(i,k)})$

7:          Compute importance weight: $w^{(i)}\left(\tau_t^{(i,k)} \mid \theta_{t-1}, \theta_t^{(i,k)}\right) \triangleq \frac{p^{(i)}\left(\tau_t^{(i,k)} \mid \theta_{t-1}\right)}{p^{(i)}\left(\tau_t^{(i,k)} \mid \theta_t^{(i,k)}\right)}$

8:          Compute update direction:

9: 
$$u_t^{(i,k)} = \beta\left(g_i\left(\tau_t^{(i,k)} \mid \theta_t^{(i,k)}\right) - c_{t-1}^{(i)} + c_{t-1}\right)$$
$$+ (1-\beta)\left[u_{t-1} + g_i\left(\tau_t^{(i,k)} \mid \theta_t^{(i,k)}\right) - w^{(i)}\left(\tau_t^{(i,k)} \mid \theta_{t-1}, \theta_t^{(i,k)}\right) g_i\left(\tau_t^{(i,k)} \mid \theta_{t-1}\right)\right]$$

10:          Update local model: $\theta_t^{(i,k+1)} = \theta_t^{(i,k)} + \eta \frac{u_t^{(i,k)}}{\left\|u_t^{(i,k)}\right\|}$

11:       **end for**

12:       Update control variate $c_t^{(i)} = \frac{1}{K} \sum_{k=0}^{K-1} g_i\left(\tau_t^{(i,k)} \mid \theta_t^{(i,k)}\right)$

13:       Send $\Delta_t^{(i)} = \theta_t^{(i,K)} - \theta_t$ and $c_t^{(i)}$ to server

14:    **end for**

15:    Server aggregates: $\hat{u}_t = \frac{1}{\eta N K} \sum_{i=1}^{N} \Delta_t^{(i)}$

16:    Update global model: $\theta_{t+1} = \theta_t + \lambda \hat{u}_t$

17:    Aggregate control variate $c_t = c_{t-1} + \frac{1}{N} \sum_{i=1}^{N}(c_t^{(i)} - c_{t-1}^{(i)})$

18:    Aggregate momentum $u_t = \beta\left(\frac{1}{N} \sum_{i=1}^{N}(c_t^{(i)} - c_{t-1}^{(i)}) + c_{t-1}\right) + (1-\beta)u_{t-1}$

19: **end for**

---

**Variance Reduction.** Beyond control variates, we further incorporate an additional correction term computed using the gradient at the previous global model: $g_i\left(\tau_t^{(i,k)} \mid \theta_{t-1}\right)$. This design follows the principle of STORM (Cutkosky & Orabona, 2019), which allows efficient reuse of past samples to reduce variance in stochastic optimization. Unlike supervised learning with stationary data distributions, RL agents interact with dynamic environments, leading to non-stationary trajectories. To address this challenge, we incorporate an IS weight, $w^{(i)}\left(\tau_t^{(i,k)} \mid \theta_{t-1}, \theta_t^{(i,k)}\right) \triangleq \frac{p^{(i)}\left(\tau_t^{(i,k)} \mid \theta_{t-1}\right)}{p^{(i)}\left(\tau_t^{(i,k)} \mid \theta_t^{(i,k)}\right)}$, which corrects for the distribution mismatch between different policies. This ensures valid variance-reduced updates despite the evolving data distribution in RL environments.

**Client-Side Momentum.** Building on the control variates and VR mechanism, we incorporate client-side momentum to stabilize the optimization trajectory and accelerate convergence across agents. Accordingly, rather than relying on plain policy gradient updates, PFEDPG-VR computes the local ascent direction using a carefully constructed estimator that corrects for distribution drift and reduces gradient variance:

$$u_t^{(i,k)} = \beta\left(g_i\left(\tau_t^{(i,k)} \mid \theta_t^{(i,k)}\right) - c_{t-1}^{(i)} + c_{t-1}\right)$$
$$+ (1-\beta)\left[u_{t-1} + g_i\left(\tau_t^{(i,k)} \mid \theta_t^{(i,k)}\right) - w^{(i)}\left(\tau_t^{(i,k)} \mid \theta_{t-1}, \theta_t^{(i,k)}\right) g_i\left(\tau_t^{(i,k)} \mid \theta_{t-1}\right)\right], \quad (3)$$

where $\beta \in (0, 1]$ is the momentum coefficient, $g_i\left(\tau_t^{(i,k)} \mid \theta_t^{(i,k)}\right)$ represents the stochastic gradient estimator computed at the current local model $\theta_t^{(i,k)}$ using the sampled trajectory $\tau_t^{(i,k)}$, and $u_{t-1}$ refers to the global momentum vector from the previous communication round, which plays a key role in maintaining stable updates and accelerating convergence in our framework.

**Local Adaptive Stepsize.** With the above local ascent estimator, the local model for each agent $i$ is updated using a normalized ascent step: $\theta_t^{(i,k+1)} = \theta_t^{(i,k)} + \eta u_t^{(i,k)} / \left\| u_t^{(i,k)} \right\|$. Here, the adaptive stepsize $\eta / \left\| u_t^{(i,k)} \right\|$ is realized by scaling the ascent direction to a unit norm. This normalization ensures that all agents make updates of equal magnitude, preventing any single agent from dominating the global model evolution. Notably, it facilitates precise analysis of model dynamics, as the distance between consecutive updates remains fixed as $\left\| \theta_t^{(i,k+1)} - \theta_t^{(i,k)} \right\| = \eta, \forall i, k, t$.

In our theoretical analysis, this fixed-step property is exactly what enables a problem-parameter-free learning rate. Once each local update satisfies $\left\| \theta_t^{(i,k+1)} - \theta_t^{(i,k)} \right\| = \eta$, the total local drift within a round is deterministically bounded by $K\eta$, and all problem-specific parameters enter the recursion only as multiplicative factors of $\eta$. This structure allows us to choose $\eta$ purely as a function of the system parameters $(N, K, T)$ so that ascent and variance terms are properly balanced, while problem-dependent constants appear only as multiplicative constants in the final convergence bound. As a result, our stepsizes depend solely on $(N, K, T)$, without requiring any explicit knowledge of unknown problem parameters, while still matching the best-known convergence rates. Furthermore, unlike prior methods such as (Huang et al., 2020), which require diminishing local learning rates to ensure convergence, our approach employs a constant learning rate across all iterations.

**Aggregated Momentum Update.** At the end of each communication round $t$, the global momentum vector is updated by aggregating the smoothed local statistics from all agents: $u_t = \beta \left( \frac{1}{N} \sum_{i=1}^N \left( c_t^{(i)} - c_{t-1}^{(i)} \right) + c_{t-1} \right) + (1-\beta)u_{t-1}$. This global update aggregates ascent directions from all agents and their local updates, capturing the collective learning dynamics across the system. By incorporating shared momentum information through $u_t$, policy update on each agent benefits from the stabilized directionality informed by others. This coordination mechanism strengthens resilience to inconsistencies in local updates, effectively mitigating the adverse effects of environment heterogeneity on the overall optimization process.

## 3.2 ALGORITHM DEVELOPMENT OF PFEDPG-HA

To further enhance robustness, we integrate HA optimization (Shen et al., 2019) into the problem-parameter-free framework and propose PFEDPG-HA. Rather than computing an explicit Hessian inverse, PFEDPG-HA leverages stochastic Hessian–vector products, ensuring that the per-iteration computational cost is linear with respect to the parameter dimension $d$. As a result, PFEDPG-HA achieves a sample complexity of $\mathcal{O}(\epsilon^{-3})$, matching that of PFEDPG-VR.

The detail of PFEDPG-HA is presented in Algorithm 2 in Section E due to space limits. PFEDPG-HA adopts the same framework as PFEDPG-VR, incorporating control variates, momentum techniques, and local adaptive stepsizes. However, it introduces a key difference in the design of local update rules. Specifically, following the approach of Furmston et al. (2016) and Shen et al. (2019), we assume that the local objective function $J_i(\theta)$ is twice differentiable for all $i \in [N]$. Under this assumption, the local update incorporates a curvature-aware correction term, which is computed as:

$$u_t^{(i,k)} = \beta \left( w^{(i)} \left( \tau_t^{(i,k)} | \theta_t^{(i,k)}, \theta_t^{(i,k)}(\alpha) \right) g_i \left( \tau_t^{(i,k)} | \theta_t^{(i,k)} \right) - c_{t-1}^{(i)} + c_{t-1} \right) + (1-\beta) \left[ u_t + \Lambda_t^{(i,k)} \right],$$

where

$$\Lambda_t^{(i,k)} \triangleq \left\langle \nabla \log p \left( \tau_t^{(i,k)} | \theta_t^{(i,k)}(\alpha) \right), v_t^{(i,k)} \right\rangle g_i \left( \tau_t^{(i,k)} | \theta_t^{(i,k)}(\alpha) \right) + \nabla \left\langle g_i \left( \tau_t^{(i,k)} | \theta_t^{(i,k)}(\alpha) \right), v_t^{(i,k)} \right\rangle, \quad (4)$$

and $v_t^{(i,k)} \triangleq \theta_t^{(i,k)} - \theta_{t-1}$. Here, $\alpha$ is sampled uniformly (as illustrated in Algorithm 2), ensuring that $\Lambda_t^{(i,k)}$ is an unbiased estimator of $\nabla J(\theta_t^{(i,k)}) - \nabla J(\theta_{t-1})$. Compared to the VR term in Eq. (3) employed by PFEDPG-VR, the local update direction in PFEDPG-HA incorporates a correction term derived from second-order information. This enhancement allows for more accurate and robust update directions. Moreover, the second term in Eq. (4) can be efficiently computed via automatic differentiation of the scalar quantity $g_i \left( \tau_t^{(i,k)} | \theta_t^{(i,k)}(\alpha) \right)$ (Fatkhullin et al., 2023). Thus, the computational cost of PFEDPG-HA does not increase and remains at $\mathcal{O}(Hd)$. In summary, PFEDPG-HA allows each agent to incorporate second-order information via stochastic Hessian-vector product approximations, enhancing optimization efficiency without incurring high computational cost.

## 4 THEORETICAL ANALYSIS

We first provide the standard assumptions used in our analysis.

**Assumption 4.1.** Let $\pi_\theta^{(i)}(a \mid s)$ denote the policy of the $i$-th agent at state $s$. There exist constants $G, M > 0$ such that the log-density of the policy function satisfies

$$\left\| \nabla_\theta \log \pi_\theta^{(i)}(a \mid s) \right\| \leq G, \quad \left\| \nabla_\theta^2 \log \pi_\theta^{(i)}(a \mid s) \right\|_2 \leq M,$$

for all $a \in \mathcal{A}$, $s \in \mathcal{S}$, and $i \in [N]$.

Assumption 4.1 provides the basis for the smoothness assumption on the objective function $J(\theta)$ commonly used in non-convex optimization (Allen-Zhu & Hazan, 2016; Reddi et al., 2016). It is also widely used in policy-gradient analysis, including (Papini et al., 2018; Xu et al., 2019; Shen et al., 2019; Liu et al., 2020; Pirotta et al., 2013; Wang et al., 2024a).

**Assumption 4.2.** For each agent $i \in [N]$, the variance of stochastic gradient $g_i(\tau \mid \theta)$ is bounded, i.e., there exists a constant $\sigma > 0$ for all policies $\pi_\theta$ such that $\mathrm{Var}(g_i(\tau \mid \theta)) = \mathbb{E}[\|g_i(\tau \mid \theta) - \nabla J_i(\theta)\|^2] \leq \sigma^2$.

Assumption 4.2 is a classical bounded-variance condition in stochastic non-convex optimization (Allen-Zhu & Hazan, 2016; Lei et al., 2017), and it is routinely assumed in policy-gradient literature such as (Papini et al., 2018; Xu et al., 2019; Shen et al., 2019; Liu et al., 2020; Pirotta et al., 2013; Wang et al., 2024a).

**Assumption 4.3.** For each agent $i \in [N]$, the variance of the IS weight $w^{(i)}(\tau \mid \theta_1, \theta_2)$ is bounded, i.e., there exists a constant $W > 0$ such that $\mathrm{Var}(w^{(i)}(\tau \mid \theta_1, \theta_2)) \leq W$ holds for any $\theta_1, \theta_2 \in \mathbb{R}^d$ and $\tau \sim p^{(i)}(\cdot \mid \theta_2)$.

Assumption 4.3 is also standard in policy-gradient analyses and appears in (Papini et al., 2018; Xu et al., 2019; Liu et al., 2020; Cortes et al., 2010; Wang et al., 2024a). With these assumptions, we provide the convergence guarantees for PFEDPG-VR and PFEDPG-HA algorithms.

**Theorem 4.4.** *Under Assumptions 4.1, 4.2, and 4.3, let the local and global learning rates of* PFEDPG-VR *be* $\eta = \frac{1}{\sqrt{KT}}$ *and* $\lambda = \frac{(NK)^{1/3}}{T^{2/3}}$, *respectively, the momentum parameter be* $\beta = \frac{(NK)^{1/3}}{T^{2/3}}$, *and* $\{\theta_t\}_{t \geq 0}$ *be the iterates generated by Algorithm 1. Then, it holds for all* $T \geq 1$ *that*

$$\frac{1}{T} \sum_{t=0}^{T-1} \mathbb{E} \left\| \nabla J(\theta_t) \right\| \leq \mathcal{O} \left( \frac{\Delta + \hat{L}}{(NKT)^{1/3}} + \frac{\hat{L}(NK)^{1/3}}{T^{2/3}} \right),$$

*where* $\Delta := \max_\theta J(\theta) - J(\theta_0)$ *and* $\hat{L} := L + L_g + C_w C_g + \sigma$, *with* $L = HR_{\max}(M + HG^2)/(1-\gamma)$, $L_g = HMR_{\max}/(1-\gamma)$, $C_g = HGR_{\max}/(1-\gamma)$, *and* $C_w = \sqrt{H(2HG^2 + M)(W+1)}$.

**Theorem 4.5.** *Under Assumptions 4.1, 4.2, and 4.3, let the local and global learning rates of* PFEDPG-HA *be* $\eta = \frac{1}{\sqrt{KT}}$ *and* $\lambda = \frac{(NK)^{1/3}}{T^{2/3}}$, *respectively, the momentum parameter be* $\beta = \frac{(NK)^{1/3}}{T^{2/3}}$, *and* $\{\theta_t\}_{t \geq 0}$ *be the iterates generated by Algorithm 2. Then, it holds for all* $T \geq 1$ *that*

$$\frac{1}{T} \sum_{t=0}^{T-1} \mathbb{E} \left\| \nabla J(\theta_t) \right\| \leq \mathcal{O} \left( \frac{\Delta + \bar{L}}{(NKT)^{1/3}} + \frac{\bar{L}(NK)^{1/3}}{T^{2/3}} \right),$$

*where* $\bar{L} := L + \sqrt{L_g + C_w C_g} + \sigma$.

*Remark* 4.6. According to Theorems 4.4 and 4.5, both PFEDPG-VR and PFEDPG-HA converge in expectation to an $\epsilon$-stationary point within $\mathcal{O}(\epsilon^{-3}/NK)$ communication rounds, exhibiting a linear speedup in the number of agents $N$ and local update steps $K$. Moreover, by choosing $NK = \mathcal{O}(\sqrt{T})$, our algorithms attain the optimal sample complexity $\mathcal{O}(\epsilon^{-3})$ together with a communication complexity of $\mathcal{O}(\epsilon^{-2})$ for finding an $\epsilon$-stationary point (Fatkhullin et al., 2023; Wang et al., 2024a).

*Remark* 4.7. In existing FRL methods, selecting optimal learning rates often requires knowledge of problem-dependent constants—such as smoothness or heterogeneity levels—which are typically unknown or difficult to estimate in practice due to the privacy-preserving nature of FRL. As a result, practitioners must rely on manual tuning, which is costly, time-consuming, and prone to suboptimal performance. In contrast, our proposed methods eliminate this dependency by setting learning rates solely based on system-level constants: the number of agents $N$, local update epochs $K$, and communication rounds $T$. This design makes our approach inherently problem-parameter-free, significantly simplifying implementation while improving robustness and scalability.

Table 1: Comparison of the results for policy-based methods in FRL.

| Algorithm | Heter◇ | Stepsize Restrictions | Complexity† | Stepsize-Related Parameters |
|---|---|---|---|---|
| **FEDPG-BR** (Fan et al. (2021)) | ✗ | $\eta \le \frac{1}{2(L(L_g+C_g^2 C_w))^{\frac{1}{3}} B^{\frac{2}{3}}}, \delta \in (0,1),$ $\delta \le \frac{1}{5NB}, e^{\frac{\delta B}{2(1-2\delta)}} \le \frac{2N}{\delta} \le e^{\frac{B}{2}}$ | $\mathcal{O}\left(\frac{\epsilon^{-10/3}}{N^{2/3}}\right)$ | $L, L_g, G, M,$ $C_g, C_w$ |
| **FEDNPG-ADMM** (Lan et al. (2023)) | ✗ | $\eta = \frac{\mu^2}{4G^2(56G^2+L)}$ | $\mathcal{O}\left(\frac{\epsilon^{-4}}{N}\right)$ | $\mu, L, L_g, G$ |
| **FEDSVRPG-M & FEDHAPG-M** (Wang et al. (2024a)) | ✓ | $\beta = \min\left\{1, \left(\frac{NK\tilde{L}^2\Delta^2}{\sigma^4 T^2}\right)^{\frac{1}{3}}\right\},$ $\lambda = \min\left\{\frac{1}{24\tilde{L}}, \sqrt{\frac{\beta NK}{72\tilde{L}^2}}\right\},$ $\eta K\tilde{L} \lesssim \min\left\{\sqrt{\frac{\tilde{L}\Delta}{G_0\lambda\tilde{L}T}}, \sqrt{\frac{\beta}{N}}, \left(\frac{\beta}{NK}\right)^{\frac{1}{4}}\right\}$ | $\mathcal{O}\left(\frac{\epsilon^{-3}}{N}\right)$ | $L, L_g, G, M,$ $C_g, C_w, \Delta, \sigma$ |
| **PFEDPG-VR & PFEDPG-HA** (This Paper) | ✓ | $\beta = \frac{(NK)^{\frac{1}{3}}}{T^{\frac{2}{3}}}, \lambda = \frac{(NK)^{\frac{1}{3}}}{T^{\frac{2}{3}}}, \eta = \frac{1}{\sqrt{KT}}$ | $\mathcal{O}\left(\frac{\epsilon^{-3}}{N}\right)$ | None |

◇ Heter indicates whether the environment heterogeneity is considered.

† Finding a parameter $\theta$ such that $\|J(\theta)\| \le \epsilon$, where $\epsilon$ is the target stationarity. When convergence is stated in terms of the average squared gradient norm, $\frac{1}{T}\sum_{t=1}^{T-1}\mathbb{E}\left[\|\nabla J(\theta_t)\|^2\right]$, one can recover a bound on the average gradient norm, $\frac{1}{T}\sum_{t=1}^{T-1}\mathbb{E}\left[\|\nabla J(\theta_t)\|\right] \le \sqrt{\frac{1}{T}\sum_{t=1}^{T-1}\mathbb{E}\left[\|\nabla J(\theta_t)\|^2\right]}$, by Jensen's inequality. Thus, results expressed in squared-norm can be equivalently interpreted in norm via a square-root transformation.

*Remark* 4.8. In Theorem 4.4, our convergence bound exhibits *linear* dependence on the problem constants $(L, L_g, C_w, C_g, \sigma)$; in Theorem 4.5, it is *linear* in $(L, \sigma)$ and *square-root* in $(L_g, C_w, C_g)$. In comparison, Wang et al. (2024a) attains *cube-root* dependence on most constants—while incurring a 2/3-*order* dependence on $C_w$—through problem-dependent stepsize balancing; Fatkhullin et al. (2023) achieves *square-root* dependence under similar tuning requirements; and Fan et al. (2021) also has *linear* dependence on $\sigma$, but with a slower convergence rate than ours. Our guarantees, however, are problem-parameter-free and remove environment heterogeneity bounds. The linear (or square-root) constant dependence in our results arises because gradient normalization fixes the effective step length without invoking unknown problem-specific constants, while momentum aggregates updates without the oracle balancing that compresses constant factors. This behavior is inherent to problem-parameter-free implementations and is commonly observed in related methods (e.g., (Yang et al., 2023; Li et al., 2024; Khaled & Jin, 2024)). Importantly, our convergence bounds share the best-known leading order in $T$ (with linear speedups in $N$ and $K$), so for moderately large $T$ the asymptotic regime dominates and the practical impact of having linear-root constants diminishes rapidly. Given the practical gains from eliminating problem-specific tuning, this modest increase in constant order is widely regarded as acceptable and well-justified in the current literature.

## 5 COMPARISON WITH PRIOR WORKS

We compare our algorithms with some representative FRL methods in Table 1. Those methods offering only inexact or asymptotic convergence are omitted from the table and discussed in the text.

**Enhanced Sample Efficiency and Scalability.** Existing methods in single-agent RL (Papini et al., 2018; Xu et al., 2019; Gargiani et al., 2022) typically require large batch sizes, limiting their practical efficiency. In contrast, our proposed algorithms sample only one trajectory per local iteration, significantly improving sample efficiency. Additionally, by naturally accommodating multiple local updates, our approach effectively alleviates communication bottlenecks prevalent in distributed environments. Importantly, theoretical analyses confirm that our methods achieve linear scalability in sample complexity with respect to the number of agents $N$ and local update steps $K$, even without assuming bounded environment heterogeneity—markedly outperforming prior work such as FEDPG-BR (Fan et al., 2021), which attains only sublinear speedup (as illustrated in Table 1).

**Problem-Parameter-Free Robustness to Arbitrary Environment Heterogeneity.** As demonstrated in Table 1, our proposed algorithms achieve convergence to an $\epsilon$-stationary point in heterogeneous FRL problems regardless of the degree of environment heterogeneity. This represents a significant improvement over existing methods such as FEDPG-BR (Fan et al., 2021), FEDNPG-ADMM

(Lan et al., 2023), and others (Jin et al., 2022; Xie & Song, 2023; Wang et al., 2023b; Zhang et al., 2024), which typically assume identical or similar environments across agents. Recent approaches like FEDSVRPG-M and FEDHAPG-M (Wang et al., 2024a), adopt similar client-side momentum techniques to accommodate environment heterogeneity. However, consistent with existing FRL methods, they generally rely on complex and problem-specific parameters—including the Lipschitz and smoothness constants ($L_g$, $L$); the initial optimality gap ($\Delta$); the gradient variance ($\sigma$); bounds on the log-density gradients of the policy ($G$, $M$); upper bounds on the gradient estimator ($C_g$) and the IS weight variance ($C_w$); and the lower bound on policy sensitivity ($\mu$)—all of which require extensive manual stepsize tuning (as summarized in Table 1). In contrast, we integrate adaptive stepsizes and control variates within the momentum framework, aligning local updates with the global objective and reducing gradient variance. This results in effectively problem-parameter-free algorithms that achieve state-of-the-art communication efficiency and robust performance under arbitrary environment heterogeneity.

**Algorithmic and Theoretical Advances Beyond Existing FRL Frameworks.** Currently, several works employ momentum to address data or environment heterogeneity in federated settings (e.g., (Xu et al., 2021; Cheng et al., 2023; Wang et al., 2024a)). However, in our problem-parameter-free regime, momentum alone is not sufficient. When stepsizes cannot be tuned against unknown problem parameters, momentum by itself cannot fully control agent drift or ensure that a problem-parameter-free stepsize remains valid under arbitrary environment heterogeneity. In our design, this is achieved only through the joint action of three key components: gradient normalization fixes the per-step update length and decouples it from unknown problem scales; control variates contract the bias between aggregated local gradients and the true global gradient under heterogeneous environments; and momentum stabilizes and accelerates these normalized, drift-corrected updates. Together, these components play distinct and indispensable roles in enabling a problem-parameter-free FRL framework that remains effective under arbitrary environment heterogeneity.

On the theoretical side, our analysis takes a fundamentally different path from prior FRL works such as (Wang et al., 2024a; Jin et al., 2022), because establishing problem-parameter-free guarantees introduces several new challenges: (i) **Handling normalized gradients:** Gradient normalization complicates the analysis of update dynamics, requiring new techniques to track its interaction with momentum. (ii) **Momentum analysis:** Momentum couples updates across rounds and agents, and new bounds are needed to capture how this coupling interacts with the control-variates recursion. (iii) **Problem-parameter-free guarantees:** Convergence must be established without inserting any problem-dependent constants into the stepsize formulas, which demands new arguments showing how normalization compensates for the absence of tuned stepsizes. Addressing these challenges requires proof techniques not present in existing FRL analyses. As a result, our framework represents a fundamental advancement in the practical deployment of FRL algorithms.

## 6 EXPERIMENTS

We conduct experiments under two cases: a tabular case and a deep RL case. In the tabular setting, we evaluate the performance of our proposed PFEDPG-VR algorithm. It is worth noting that PFEDPG-HA is excluded from this evaluation because the objective function $J_i(\theta)$ is not twice differentiable. For the deep RL case, we assess both PFEDPG-VR and PFEDPG-HA on CartPole and HalfCheetah tasks (Todorov et al., 2012). To provide a comprehensive comparison, we include three baselines: PAVG (Jin et al., 2022), FEDSVRPG-M, and FEDHAPG-M (Wang et al., 2024a). Detailed experimental configurations are provided in Section F.

In the tabular setting, we assess algorithm performance using random MDPs, where both state transitions and reward functions are randomly generated. To model environment heterogeneity, we follow the protocol of Jin et al. (2022), which introduces heterogeneity by mixing a nominal kernel $\mathcal{P}_0$ with agent-specific kernels $\mathcal{P}_i$ via $\mathcal{P}^{(i)} = \kappa \mathcal{P}_i + (1 - \kappa)\mathcal{P}_0$. By adjusting $\kappa$, we simulate varying levels of heterogeneity across agents. We benchmark our proposed PFEDPG-VR algorithm against the FEDSVRPG-M and PAVG baselines, and illustrate the average mean reward as defined in Eq. (2). Results in Table 2 show that PFEDPG-VR achieves superior performance relative to the baselines and remains robust across different levels of heterogeneity, consistent with our theoretical analysis.

In the deep RL setting, we evaluate our algorithms on CartPole and HalfCheetah tasks, two widely used benchmarks in the MuJoCo simulation environment (Todorov et al., 2012). To simulate environment heterogeneity, we vary the parameters of the initial state distribution across agents. The CartPole policy is modeled as a categorical distribution, while the HalfCheetah policy is modeled as a

Table 2: Performance under different values of $\kappa$ on Random MDPs.

| Random MDPs | | | | | | |
|---|---|---|---|---|---|---|
| Algorithm | $\kappa = 0$ | $\kappa = 0.2$ | $\kappa = 0.4$ | $\kappa = 0.6$ | $\kappa = 0.8$ | $\kappa = 1.0$ |
| PFEDPG-VR | $\mathbf{10.14}_{\pm 0.07}$ | $\mathbf{10.14}_{\pm 0.08}$ | $\mathbf{10.12}_{\pm 0.08}$ | $\mathbf{10.15}_{\pm 0.08}$ | $\mathbf{10.18}_{\pm 0.07}$ | $\mathbf{10.14}_{\pm 0.07}$ |
| FEDSVRPG-M | $10.11_{\pm 0.07}$ | $10.09_{\pm 0.07}$ | $10.09_{\pm 0.07}$ | $10.09_{\pm 0.07}$ | $10.06_{\pm 0.08}$ | $10.03_{\pm 0.08}$ |
| PAVG | $10.03_{\pm 0.07}$ | $10.01_{\pm 0.07}$ | $10.02_{\pm 0.08}$ | $10.00_{\pm 0.07}$ | $10.03_{\pm 0.08}$ | $9.96_{\pm 0.08}$ |

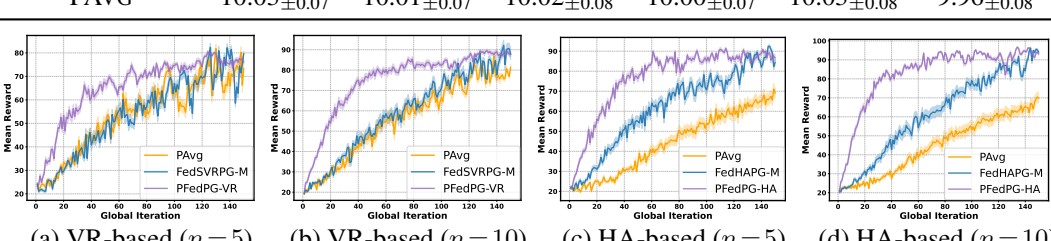

(a) VR-based ($n=5$)    (b) VR-based ($n=10$)    (c) HA-based ($n=5$)    (d) HA-based ($n=10$)

Figure 1: Mean rewards over global iterations for the CartPole task with varying numbers of agents.

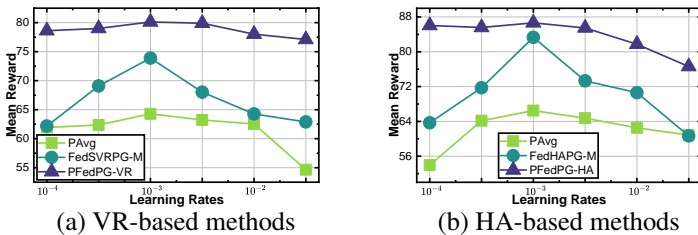

(a) VR-based methods      (b) HA-based methods

Figure 2: Mean rewards versus learning rates for different methods on the CartPole task.

Gaussian distribution, both parameterized by fully connected neural networks with two hidden layers (8 neurons for categorical policies and 32 for Gaussian policies) and hyperbolic tangent activations. In Figure 1, we present the mean reward curves over global iterations for both our proposed methods and the baseline algorithms on CartPole, under VR-based and HA-based settings with varying numbers of agents. The results show that our methods consistently achieve the highest performance across all configurations. This advantage is attributed to our problem-parameter-free design, which enables convergence at optimal rates without requiring learning rate tuning. Furthermore, as the number of agents increases, our algorithms demonstrate improved performance, aligning with the theoretical guarantee of linear speedup. Results on HalfCheetah, shown in Figure 5 in Section F, further illustrate that our method clearly surpasses all baselines, underscoring its robustness and effectiveness even in continuous RL tasks. Additional experiments on the impact of varying the local update steps and the momentum coefficient $\beta$, as well as ablation studies, are provided in Section F.

Figure 2 illustrates the mean rewards across different learning rates on the CartPole task for both our proposed methods and the baselines. In both the VR-based and HA-based algorithms, our methods consistently deliver stable and superior performance across a wide range of initial learning rates. This demonstrates that the combination of momentum, gradient normalization, and control variates in our framework significantly enhances learning rate robustness. In contrast, baseline methods exhibit sharp performance degradation when the learning rate deviates from a carefully tuned value. These results empirically validate the problem-parameter-free nature of our approach and highlight its ability to maintain strong performance without sensitive hyperparameter selection.

## 7 CONCLUSIONS

In this paper, we proposed two problem-parameter-free FRL algorithms—PFEDPG-VR and PFEDPG-HA. Our approaches effectively address critical challenges in FRL, notably the costly and impractical requirement of manually tuning problem-specific hyperparameters. By integrating control variates, momentum techniques, and adaptive stepsizes, our methods ensure robust convergence to the exact stationary point without imposing any boundedness assumptions on environment heterogeneity. Theoretical analysis confirms the linear scalability of our algorithms with respect to the number of agents and local update steps, demonstrating substantial advantages over existing FRL approaches. Empirical results further validate that our algorithms achieve superior robustness and ease of practical implementation, highlighting their suitability for real-world FRL tasks.

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

## A   ADDITIONAL DISCUSSION ON RELATED WORKS

**Federated Reinforcement Learning.** FRL has recently emerged as an effective approach for enabling collaborative policy training among multiple distributed agents, particularly addressing challenges related to data privacy and decentralized decision-making. Initial FRL methods primarily adapted supervised FL techniques, such as FEDAVG (McMahan et al., 2017), to RL settings. For example, Nadiger et al. (2019) introduced an FRL algorithm integrating Double Q-learning (Hasselt, 2010) and FEDAVG, which demonstrated the capability to obtain personalized policies among multiple participants by employing smoothing average techniques. Similarly, (Lim et al., 2020) proposed an FRL variant that merged Proximal Policy Optimization (Schulman et al., 2017) with FEDAVG, successfully extending federated methods into policy-based RL.

More recent research has shifted toward exploring FRL algorithms under heterogeneous environments (Qi et al., 2021; Wang et al., 2023a; Woo et al., 2025; Khodadadian et al., 2022). These studies highlight a key limitation of traditional FRL methods: their convergence guarantees often degrade in the presence of environmental heterogeneity, typically only ensuring convergence to a neighborhood around a stationary point. In response, several algorithms—such as FEDTD(0) (Wang et al., 2023a), FEDSARSA (Zhang et al., 2024), FEDSVRPG-M, and FEDHAPG-M (Wang et al., 2024a)—have been proposed to achieve linear speed-up even in heterogeneous settings.

Despite these advancements, a major challenge remains: the reliance on carefully tuned hyperparameters, particularly stepsizes, which are often tied to problem-specific parameters. This sensitivity limits the practicality of current FRL methods in real-world applications, where such parameters are typically unknown or difficult to estimate due to privacy constraints. To the best of our knowledge, no existing FRL algorithm offers a fully problem-parameter-free solution. Addressing this significant gap, developing a problem-parameter-free adaptive FRL algorithm remains an important and open research direction.

**Adaptive Methods.** Adaptive gradient techniques automatically tailor the stepsize to the local geometry of the objective, alleviating the need for extensive manual tuning. Early milestones include AdaGrad (McMahan & Streeter, 2010; Duchi et al., 2011), whose success in convex optimization spurred a family of variants such as Adadelta (Zeiler, 2012), RMSprop (Tieleman & Hinton, 2017), and Adam (Xie et al., 2024). More recent work has refined AdaGrad with normalized-gradient schemes—e.g., AdaNGD (Levy, 2017), AcceleGrad (Levy et al., 2018), and AdaGrad-Norm (Xie et al., 2020)—that remove explicit dependence on Lipschitz or curvature constants, yielding practically tuning-free stepsizes.

Adaptive stepsizes have also been adopted in supervised FL. Frameworks such as (Reddi et al., 2020; Yan et al., 2025) integrate AdaGrad, Adam, or Yogi into the FEDAVG scheme, improving convergence without stepsize search. However, these approaches were designed for static data and do not address the environment heterogeneity intrinsic to FRL settings, where each client interacts with a distinct and potentially non-stationary MDP. In FRL, gradient variance stems not only from data imbalance but also from policy–environment feedback loops, necessitating a more nuanced adaptation mechanism grounded in rigorous theoretical analysis.

To address these challenges, we propose the first FRL framework that is both problem-parameter-free and heterogeneity-agnostic. Our method combines momentum, control variates, and gradient normalization to reduce the effects of agent drift while maintaining fast local adaptation. These components interact in a subtle but critical way, necessitating a refined analytical framework to capture their combined effect. Importantly, our convergence analysis removes the need for explicit bounds on inter-agent heterogeneity while still achieving provably optimal rates. Together, these advances close the tuning gap in FRL and provide a practical solution for arbitrarily heterogeneous environments.

## B  NOTATION

We initialize $\mathcal{F}_0 = \emptyset$ and recursively define the $\sigma$-algebras $\mathcal{F}_t^{(i,k)} := \sigma\left(\left\{\theta_t^{(i,j)}\right\}_{0 \leq j \leq k} \cup \mathcal{F}_t\right)$ for each global iteration $t \geq 0$, agent $i \in [N]$, and local step $k = 0, 1, \ldots, K$, with $\mathcal{F}_{t+1} := \sigma\left(\cup_i \mathcal{F}_t^{(i,K)}\right)$. The conditional expectation $\mathbb{E}_t[\cdot] := \mathbb{E}[\cdot \mid \mathcal{F}_t]$ is taken with respect to the trajectories $\left\{\tau_t^{(i,k)}\right\}_{1 \leq i \leq N, 0 \leq k < K}$ generated during the $t$-th iteration, while $\mathbb{E}[\cdot]$ denotes the global expectation over all algorithmic randomness. We define $\Delta := J(\theta^*) - J(\theta_0)$ as the initial optimality gap and initialize $\theta_{-1} := \theta_0$, where $\theta^*$ is the optimal policy for the optimization problem (2).

## C  USEFUL LEMMAS AND INEQUALITIES

**Proposition C.1.** (Proposition 5.2 in (Xu et al., 2020)) *Under Assumption 4.1, both $J(\theta)$ and $\{J_i(\theta)\}_{i=1}^N$ are $L$-smooth, where the smoothness constant is given by $L = HR_{\max}(M + HG^2)/(1 - \gamma)$. In addition, for all $\theta_1, \theta_2 \in \mathbb{R}^d$, the gradient estimator satisfies*

$$\|g_i(\tau \mid \theta_1) - g_i(\tau \mid \theta_2)\|_2 \leq L_g\|\theta_1 - \theta_2\|_2$$

*and $\|g_i(\tau \mid \theta)\| \leq C_g$ for all $\theta \in \mathbb{R}^d$ and $i \in [N]$, where $L_g = HMR_{\max}/(1 - \gamma)$ and $C_g = HGR_{\max}/(1 - \gamma)$.*

**Lemma C.2.** (Lemma 6.1 in Xu et al. (2020)) *Under Assumption 4.1 and Assumption 4.3, the variance of the IS weight satisfies*

$$\text{Var}\left(w^{(i)}(\tau \mid \theta_1, \theta_2)\right) \leq C_w^2\|\theta_1 - \theta_2\|^2$$

*for any $\theta_1, \theta_2 \in \mathbb{R}^d$ and any $i \in [N]$, where $C_w^2 = H\left(2HG^2 + M\right)(W + 1)$.*

Throughout the appendix, we frequently employ the following inequalities:

- Given any two vectors $x, y \in \mathbb{R}^d$, for any $a > 0$, we have

$$\|x + y\|^2 \leq (1 + a)\|x\|^2 + \left(1 + \frac{1}{a}\right)\|y\|^2. \tag{5}$$

- Given any two vectors $x, y \in \mathbb{R}^d$, we have

$$\|x + y\| \leq \|x\| + \|y\|. \tag{6}$$

- Given any two vectors $x, y \in \mathbb{R}^d$, for any constant $a > 0$, we have

$$\langle x, y \rangle \leq \frac{a}{2}\|x\|^2 + \frac{1}{2a}\|y\|^2. \tag{7}$$

  This inequality is commonly known as Young's inequality.

- Given $m$ vectors $x_1, \ldots, x_m \in \mathbb{R}^d$, the following inequality follows from a direct application of Jensen's inequality:

$$\left\|\sum_{i=1}^m x_i\right\|^2 \leq m \sum_{i=1}^m \|x_i\|^2. \tag{8}$$

## D  CONVERGENCE ANALYSIS FOR PFEDPG-VR

**Lemma D.1.** *For any $t$, we have*

$$\frac{1}{NK} \sum_{i,k} \left\|\theta_t^{(i,k)} - \theta_t\right\|^2 \leq \frac{1}{3}\eta^2 K^2 \quad \text{and} \quad \frac{1}{NK} \sum_{i,k} \left\|\theta_t^{(i,k)} - \theta_t\right\| \leq \frac{1}{2}\eta K.$$

*Proof.* From the update rule of the local model, for any $i, k$, and $t$, we have

$$\left\| \theta_t^{(i,k+1)} - \theta_t^{(i,k)} \right\| = \eta \left\| \frac{u_t^{(i,k)}}{\left\| u_t^{(i,k)} \right\|} \right\| \leq \eta.$$

Then,

$$\left\| \theta_t^{(i,k)} - \theta_t \right\|^2 = \left\| \sum_{j=0}^{k-1} \left( \theta_t^{i,j+1} - \theta_t^{i,j} \right) \right\|^2 \leq k \sum_{j=0}^{k-1} \left\| \theta_t^{i,j+1} - \theta_t^{i,j} \right\|^2 \leq \eta^2 k^2.$$

Summing the above inequality over $i$ and $k$ yields

$$\frac{1}{NK} \sum_{i,k} \left\| \theta_t^{(i,k)} - \theta_t \right\|^2 \leq \frac{\eta^2}{K} \sum_{k=0}^{K-1} k^2 \leq \frac{\eta^2}{6K} K(K-1)(2K-1) \leq \frac{1}{3} \eta^2 K^2.$$

Similarly, we have

$$\frac{1}{NK} \sum_{i,k} \left\| \theta_t^{(i,k)} - \theta_t \right\| = \frac{1}{NK} \sum_{i,k} \left( \left\| \theta_t^{(i,k)} - \theta_t \right\|^2 \right)^{\frac{1}{2}} \leq \frac{\eta}{K} \sum_{k=0}^{K-1} k \leq \frac{1}{2} \eta K.$$

$\square$

These inequalities in Lemma D.1 are frequently used in our analysis.

From the $L$-smoothness of $J(\cdot)$, we have

$$J(\theta_{t+1}) - J(\theta_t)$$

$$\geq \nabla J(\theta_t)^\top (\theta_{t+1} - \theta_t) - \frac{L}{2} \left\| \theta_{t+1} - \theta_t \right\|^2$$

$$\overset{(a)}{\geq} \lambda \nabla J(\theta_t)^\top \left( \frac{1}{NK} \sum_{i,k} \frac{u_t^{(i,k)}}{\left\| u_t^{(i,k)} \right\|} \right) - \frac{\lambda^2 L}{2}$$

$$= \lambda \left( \nabla J(\theta_t) - u_t \right)^\top \left( \frac{1}{NK} \sum_{i,k} \frac{u_t^{(i,k)}}{\left\| u_t^{(i,k)} \right\|} \right) + \lambda u_t^\top \left( \frac{1}{NK} \sum_{i,k} \frac{u_t^{(i,k)}}{\left\| u_t^{(i,k)} \right\|} \right) - \frac{\lambda^2 L}{2}$$

$$\geq -\lambda \left\| \nabla J(\theta_t) - u_t \right\| - \lambda u_t^\top \left( \frac{1}{NK} \sum_{i,k} \frac{u_t^{(i,k)}}{\left\| u_t^{(i,k)} \right\|} - \frac{u_t}{\left\| u_t \right\|} \right) + \lambda \left\| u_t \right\| - \frac{\lambda^2 L}{2}$$

$$\overset{(b)}{\geq} -2\lambda \left\| \nabla J(\theta_t) - u_t \right\| + \lambda \left\| \nabla J(\theta_t) \right\| - \lambda \left\| u_t \right\| \left\| \frac{1}{NK} \sum_{i,k} \frac{u_t^{(i,k)}}{\left\| u_t^{(i,k)} \right\|} - \frac{u_t}{\left\| u_t \right\|} \right\| - \frac{\lambda^2 L}{2}$$

$$\overset{(c)}{\geq} -2\lambda \left\| \nabla J(\theta_t) - u_t \right\| + \lambda \left\| \nabla J(\theta_t) \right\| - \frac{\lambda}{NK} \sum_{i,k} \left\| u_t^{(i,k)} - u_t \right\| - \frac{\lambda^2 L}{2}, \tag{9}$$

where $(a)$ uses the inequality that $\left\| \theta_{t+1} - \theta_t \right\| = \left\| \frac{\lambda}{NK} \sum_{i,k} \frac{u_t^{(i,k)}}{\left\| u_t^{(i,k)} \right\|} \right\| \leq \lambda$, $(b)$ is based on $-(\lambda \| \nabla J(\theta_t) \| - \lambda \| u_t \|) \geq -\lambda \| \nabla J(\theta_t) - u_t \|$, and $(c)$ is from the following relation:

$$\left\| u_t \right\| \left\| \frac{1}{NK} \sum_{i,k} \frac{u_t^{(i,k)}}{\left\| u_t^{(i,k)} \right\|} - \frac{u_t}{\left\| u_t \right\|} \right\|$$

$$= \frac{\left\| u_t \right\|}{NK} \left\| \sum_{i,k} \left( \frac{u_t^{(i,k)}}{\left\| u_t^{(i,k)} \right\|} - \frac{u_t}{\left\| u_t \right\|} \right) \right\|$$

$$= \frac{\|u_t\|}{NK} \left\| \sum_{i,k} \frac{\|u_t\| - \left\|u_t^{(i,k)}\right\|}{\|u_t\| \left\|u_t^{(i,k)}\right\|} u_t^{(i,k)} \right\|$$

$$\leq \frac{\|u_t\|}{NK} \sum_{i,k} \frac{\left| \|u_t\| - \left\|u_t^{(i,k)}\right\| \right|}{\|u_t\| \left\|u_t^{(i,k)}\right\|} \left\|u_t^{(i,k)}\right\|$$

$$= \frac{1}{NK} \sum_{i,k} \left| \|u_t\| - \left\|u_t^{(i,k)}\right\| \right|$$

$$\leq \frac{1}{NK} \sum_{i,k} \left\| u_t^{(i,k)} - u_t \right\|. \tag{10}$$

Taking expectation on both sides of Eq. (9), we obtain

$$\lambda \mathbb{E} \|\nabla J(\theta_t)\| \leq \mathbb{E} \left[ J(\theta_{t+1}) - J(\theta_t) \right] + 2\lambda \mathbb{E} \|\nabla J(\theta_t) - u_t\|$$

$$+ \frac{\lambda}{NK} \mathbb{E} \left[ \sum_{i,k} \left\| u_t^{(i,k)} - u_t \right\| \right] + \frac{\lambda^2 L}{2}. \tag{11}$$

Summing the above inequality over $t$ and dividing it by $\lambda T$, we have

$$\frac{1}{T} \sum_{t=0}^{T-1} \mathbb{E} \|\nabla J(\theta_t)\| \leq \frac{1}{\lambda T} \mathbb{E} \left[ J(\theta_T) - J(\theta_0) \right] + \frac{2}{T} \sum_{t=0}^{T-1} \mathbb{E} \|\nabla J(\theta_t) - u_t\|$$

$$+ \frac{1}{NKT} \sum_{t=0}^{T-1} \mathbb{E} \left[ \sum_{i,k} \left\| u_t^{(i,k)} - u_t \right\| \right] + \frac{\lambda L}{2}. \tag{12}$$

We have the following results on the terms $\frac{1}{T} \sum_{t=0}^{T-1} \mathbb{E} \|\nabla J(\theta_t) - u_t\|$ and $\frac{1}{NKT} \sum_{t=0}^{T-1} \mathbb{E} \left[ \sum_i \left\| u_t^{(i,k)} - u_t \right\| \right]$ in Eq. (12).

**Lemma D.2.** *Under Assumptions 4.1, 4.2, and 4.3, the disparity $\frac{1}{T} \sum_{t=0}^{T-1} \mathbb{E} \|\nabla J(\theta_t) - u_t\|$ is upper bounded by:*

$$\frac{1}{T} \sum_{t=0}^{T-1} \mathbb{E} \|\nabla J(\theta_t) - u_t\|$$

$$\leq \frac{1}{\beta T} \left( \frac{\sigma}{\sqrt{NK}} + \frac{\beta C_w C_g \eta K}{2} + \frac{(2C_w C_g + L_g)\eta K}{2} + 2\beta\sigma \right) + \frac{\eta K (C_w C_g + L_g)}{2\beta}$$

$$+ \eta C_w C_g \sqrt{\frac{2K}{N\beta}} + \lambda L_g \sqrt{\frac{3}{NK\beta}} + \sigma \sqrt{\frac{\beta}{NK}}. \tag{13}$$

**Lemma D.3.** *Under Assumptions 4.1, 4.2, and 4.3, the gradient dissimilarity $\frac{1}{NKT} \sum_{t=0}^{T-1} \mathbb{E} \left[ \sum_i \left\| u_t^{(i,k)} - u_t \right\| \right]$ is upper bounded by:*

$$\frac{1}{NKT} \sum_{t=1}^{T} \mathbb{E} \left[ \sum_{i,k} \left\| u_t^{(i,k)} - u_t \right\| \right]$$

$$\leq 2\beta \left( \left( 1 + \frac{2}{\sqrt{K}} \right) \sigma + 5\eta K (C_w C_g + L_g) + 6\lambda (C_w C_g + L_g) \right)$$

$$+ \eta K (L_g + 3C_w C_g) + 2\lambda L_g. \tag{14}$$

The proofs of Lemma D.2 and Lemma D.3 are presented in Section D.1 and Section D.2, respectively. Together, these lemmas provide two key components needed to establish Theorem 4.4, with the detailed proof given in Section D.3.

### D.1 PROOF OF LEMMA D.2

**Lemma D.4.** *For any $i, t$, define $\phi_t^{(i)} = \mathbb{E}\|\nabla J_i(\theta_t) - c_t^{(i)}\|^2$, we have*

$$\phi_t^{(i)} \leq \frac{2\sigma^2}{K} + \frac{2}{3}\eta^2 K^2 (C_w^2 C_g^2 + L_g^2). \tag{15}$$

*Proof.* Since $c_t^{(i)} = \frac{1}{K} \sum_k g_i\left(\tau_t^{(i,k)} \mid \theta_t^{(i,k)}\right)$, using Young's inequality repeatedly, we can get that

$$\phi_t^{(i)} = \mathbb{E}\left\|\frac{1}{K}\sum_k \left(\nabla J_i(\theta_t) - g_i\left(\tau_t^{(i,k)} \mid \theta_t^{(i,k)}\right)\right)\right\|^2$$

$$= \mathbb{E}\left\|\frac{1}{K}\sum_k \left(\nabla J_i(\theta_t) \mp w^{(i)}\left(\tau_t^{(i,k)} \mid \theta_t, \theta_t^{(i,k)}\right) g_i\left(\tau_t^{(i,k)} \mid \theta_t\right) - g_i\left(\tau_t^{(i,k)} \mid \theta_t^{(i,k)}\right)\right)\right\|^2$$

$$\leq \frac{2\sigma^2}{K} + \frac{2C_w^2 C_g^2 + 2L_g^2}{K}\sum_k \mathbb{E}\left\|\theta_t^{(i,k)} - \theta_t\right\|^2$$

$$\leq \frac{2\sigma^2}{K} + \frac{2}{3}\eta^2 K^2 (C_w^2 C_g^2 + L_g^2), \tag{16}$$

where the last inequality uses the results in Lemma D.1. $\qquad\square$

Define $\mathcal{E}_t := \nabla J(\theta_t) - u_t$. From the update rule of momentum $u_t$, we have

$$\mathcal{E}_t = \nabla J(\theta_t) - \frac{1}{NK}\sum_{i,k} g_i\left(\tau_t^{(i,k)} \mid \theta_t^{(i,k)}\right) + \frac{\beta}{N}\sum_i (c_{t-1}^{(i)} - c_{t-1})$$

$$- (1-\beta)\left(u_{t-1} \mp \nabla J(\theta_{t-1}) - \frac{1}{NK}\sum_{i,k} w^{(i)}\left(\tau_t^{(i,k)} \mid \theta_t, \theta_t^{(i,k)}\right) g_i\left(\tau_t^{(i,k)} \mid \theta_t\right)\right)$$

$$= (1-\beta)\mathcal{E}_{t-1} + \frac{1}{NK}\sum_{i,k}\left(w^{(i)}\left(\tau_t^{(i,k)} \mid \theta_t, \theta_t^{(i,k)}\right) g_i\left(\tau_t^{(i,k)} \mid \theta_t\right) - g_i\left(\tau_t^{(i,k)} \mid \theta_t^{(i,k)}\right)\right)$$

$$+ \beta\left(\nabla J(\theta_t) - c_{t-1} - \frac{1}{NK}\sum_{i,k}\left(w^{(i)}\left(\tau_t^{(i,k)} \mid \theta_t, \theta_t^{(i,k)}\right) g_i\left(\tau_t^{(i,k)} \mid \theta_t\right) - c_{t-1}^{(i)}\right)\right)$$

$$+ (1-\beta)\left(\frac{1}{NK}\sum_{i,k}\left(w^{(i)}\left(\tau_t^{(i,k)} \mid \theta_{t-1}, \theta_t^{(i,k)}\right) g_i\left(\tau_t^{(i,k)} \mid \theta_{t-1}\right)\right.\right.$$

$$\left.\left. - w^{(i)}\left(\tau_t^{(i,k)} \mid \theta_t, \theta_t^{(i,k)}\right) g_i\left(\tau_t^{(i,k)} \mid \theta_t\right)\right) + \nabla J(\theta_t) - \nabla J(\theta_{t-1})\right)$$

$$= (1-\beta)^t \mathcal{E}_0 + \sum_{\tau=1}^t r_\tau (1-\beta)^{t-\tau} + \sum_{\tau=1}^t z_\tau (1-\beta)^{t+1-\tau} + \sum_{\tau=1}^t \beta q_\tau (1-\beta)^{t-\tau}, \tag{17}$$

where we define

$$r_t = \frac{1}{NK}\sum_{i,k}\left(w^{(i)}\left(\tau_t^{(i,k)} \mid \theta_t, \theta_t^{(i,k)}\right) g_i\left(\tau_t^{(i,k)} \mid \theta_t\right) - g_i\left(\tau_t^{(i,k)} \mid \theta_t^{(i,k)}\right)\right),$$

$$z_t = \nabla J(\theta_t) - c_{t-1} - \frac{1}{NK}\sum_{i,k}\left(w^{(i)}\left(\tau_t^{(i,k)} \mid \theta_t, \theta_t^{(i,k)}\right) g_i\left(\tau_t^{(i,k)} \mid \theta_t\right) - c_{t-1}^{(i)}\right),$$

$$q_t = \frac{1}{NK}\sum_{i,k}\left(w^{(i)}\left(\tau_t^{(i,k)} \mid \theta_{t-1}, \theta_t^{(i,k)}\right) g_i\left(\tau_t^{(i,k)} \mid \theta_{t-1}\right) - w^{(i)}\left(\tau_t^{(i,k)} \mid \theta_t, \theta_t^{(i,k)}\right) g_i\left(\tau_t^{(i,k)} \mid \theta_t\right)\right)$$

$$+ \nabla J(\theta_t) - \nabla J(\theta_{t-1}) \Big).$$

Based on the triangle inequality of $\ell_2$ norm and the concavity of the square root $(\cdot)^{\frac{1}{2}}$, we get:

$$\mathbb{E} \left\| \mathcal{E}_t \right\| \leq (1 - \beta)^t \mathbb{E} \left\| \mathcal{E}_0 \right\| + \sum_{\tau=1}^t \mathbb{E} \left\| r_\tau \right\| (1 - \beta)^{t-\tau}$$

$$+ \left( \mathbb{E} \left\| \sum_{\tau=1}^t z_\tau (1 - \beta)^{t+1-\tau} \right\|^2 \right)^{\frac{1}{2}} + \left( \mathbb{E} \left\| \sum_{\tau=1}^t \beta q_\tau (1 - \beta)^{t-\tau} \right\|^2 \right)^{\frac{1}{2}}. \tag{18}$$

Since $c_{-1}^{(i)} = \frac{1}{K} \sum_{k=0}^{K-1} g_i \left( \tau_{-1}^{(i,k)} \mid \theta_0 \right)$ for any $i$, $c_{-1} = \frac{1}{N} \sum_i c_{-1}^{(i)}$, and $u_{-1} = c_{-1}$, we have

$$\mathbb{E} \left\| \mathcal{E}_0 \right\|$$

$$= \mathbb{E} \left\| \nabla J(\theta_0) - \frac{1}{NK} \sum_{i,k} g_i \left( \tau_{-1}^{(i,k)} \mid \theta_0 \right) \right.$$

$$\left. + \frac{1}{NK} \sum_{i,k} \left( \beta g_i \left( \tau_{-1}^{(i,k)} \mid \theta_0 \right) + (1 - \beta) w^{(i)} \left( \tau_0^{(i,k)} \mid \theta_0, \theta_0^{(i,k)} \right) g_i \left( \tau_0^{(i,k)} \mid \theta_0 \right) - g_i \left( \tau_0^{(i,k)} \mid \theta_0^{(i,k)} \right) \right) \right\|$$

$$= \mathbb{E} \left\| \nabla J(\theta_0) \mp \frac{1}{NK} \sum_{i,k} w^{(i)} \left( \tau_{-1}^{(i,k)} \mid \theta_0, \theta_{-1}^{(i,k)} \right) g_i \left( \tau_{-1}^{(i,k)} \mid \theta_0 \right) - \frac{1}{NK} \sum_{i,k} g_i \left( \tau_{-1}^{(i,k)} \mid \theta_0 \right) \right.$$

$$+ \frac{1}{NK} \sum_{i,k} \left( \beta g_i \left( \tau_{-1}^{(i,k)} \mid \theta_0 \right) \mp \beta w^{(i)} \left( \tau_{-1}^{(i,k)} \mid \theta_0, \theta_{-1}^{(i,k)} \right) g_i \left( \tau_{-1}^{(i,k)} \mid \theta_0 \right) \right.$$

$$\left. \mp \beta \nabla J_i(\theta_0) + (1 - \beta) w^{(i)} \left( \tau_0^{(i,k)} \mid \theta_0, \theta_0^{(i,k)} \right) g_i \left( \tau_0^{(i,k)} \mid \theta_0 \right) - g_i \left( \tau_0^{(i,k)} \mid \theta_0^{(i,k)} \right) \right) \right\|$$

$$\leq \frac{\sigma}{\sqrt{NK}} + \frac{(1 + \beta) C_w C_g}{NK} \sum_{i,k} \mathbb{E} \left\| \theta_0 - \theta_{-1}^{(i,k)} \right\| + \frac{C_w C_g + L_g}{NK} \sum_{i,k} \mathbb{E} \left\| \theta_0^{(i,k)} - \theta_0 \right\| + 2\beta\sigma$$

$$\leq \frac{\sigma}{\sqrt{NK}} + \frac{\beta C_w C_g \eta K}{2} + \frac{(2 C_w C_g + L_g) \eta K}{2} + 2\beta\sigma, \tag{19}$$

where the last inequality uses the results in Lemma D.1. Then, we have

$$\left\| r_t \right\| \leq \frac{C_w C_g + L_g}{NK} \sum_{i,k} \mathbb{E} \left\| \theta_t^{(i,k)} - \theta_t \right\| \leq \frac{(C_w C_g + L_g) \eta K}{2}. \tag{20}$$

Additionally, conditioned on $\mathcal{F}_t$, the term $q_t$ depends only on the freshly sampled trajectories $\tau_t^{(i,k)}$ at round $t$. Since the IS–corrected estimator is unbiased, we have $\mathbb{E}[q_t \mid \mathcal{F}_t] = 0$. Hence, for any $0 \leq t_1 < t_2 \leq T - 1$, we can get that

$$\mathbb{E} \langle q_{t_1}, q_{t_2} \rangle = \mathbb{E} \langle q_{t_1}, \mathbb{E}[q_{t_2} \mid \mathcal{F}_{t_2}] \rangle = 0.$$

Futher, for any $t$, we have

$$\mathbb{E} \left\| q_t \right\|^2$$

$$\leq \mathbb{E} \left\| \frac{1}{NK} \sum_{i,k} \left( w^{(i)} \left( \tau_t^{(i,k)} \mid \theta_{t-1}, \theta_t^{(i,k)} \right) g_i \left( \tau_t^{(i,k)} \mid \theta_{t-1} \right) - w^{(i)} \left( \tau_t^{(i,k)} \mid \theta_t, \theta_t^{(i,k)} \right) g_i \left( \tau_t^{(i,k)} \mid \theta_t \right) \right) \right.$$

$$\left. + \nabla J(\theta_t) - \nabla J(\theta_{t-1}) \right\|^2$$

$$\leq 3\mathbb{E}\left\|\frac{1}{NK}\sum_{i,k}\left[w^{(i)}\left(\tau_t^{(i,k)}\mid\theta_t,\theta_t^{(i,k)}\right)-1\right]g_i\left(\tau_t^{(i,k)}\mid\theta_t\right)\right\|^2$$

$$+\frac{3}{N^2K^2}\sum_{i,k}\mathbb{E}\left\|g_i\left(\tau_t^{(i,k)}\mid\theta_t\right)-g_i\left(\tau_t^{(i,k)}\mid\theta_{t-1}\right)\right\|^2$$

$$+3\mathbb{E}\left\|\frac{1}{NK}\sum_{i,k}\left[w^{(i)}\left(\tau_t^{(i,k)}\mid\theta_{t-1},\theta_t^{(i,k)}\right)-1\right]g_i\left(\tau_t^{(i,k)}\mid\theta_{t-1}\right)\right\|^2$$

$$\leq\frac{3C_g^2C_w^2}{N^2K^2}\sum_{i,k}\mathbb{E}\left\|\theta_t^{(i,k)}-\theta_t\right\|^2+\frac{3L_g^2}{NK}\mathbb{E}\left\|\theta_{t-1}-\theta_t\right\|^2+\frac{3C_g^2C_w^2}{N^2K^2}\sum_{i,k}\mathbb{E}\left\|\theta_t^{(i,k)}-\theta_{t-1}\right\|^2$$

$$\leq\frac{2C_g^2C_w^2\eta^2K}{N}+\frac{3\lambda^2L_g^2}{NK}, \tag{21}$$

where the last inequality uses the results in Lemma D.1. Then, we have

$$\mathbb{E}\left\|\sum_{\tau=1}^t q_\tau(1-\beta)^{t+1-\tau}\right\|^2=\sum_{\tau=1}^t\mathbb{E}\left\|q_\tau\right\|^2(1-\beta)^{2(t+1-\tau)}\leq\frac{2C_g^2C_w^2\eta^2K}{N\beta}+\frac{3\lambda^2L_g^2}{NK\beta}. \tag{22}$$

Similarly, since $\mathbb{E}[z_t\mid\mathcal{F}_t]=0$, for any $0\leq t_1<t_2\leq T-1$, we have

$$\mathbb{E}\left\langle z_{t_1},z_{t_2}\right\rangle=\mathbb{E}\left\langle z_{t_1},\mathbb{E}[z_{t_2}\mid\mathcal{F}_{t_2}]\right\rangle=0.$$

For any $t$, we have

$$\mathbb{E}\left\|z_t\right\|^2\leq\mathbb{E}\left\|\nabla J(\theta_t)-\frac{1}{NK}\sum_{i,k}w^{(i)}\left(\tau_t^{(i,k)}\mid\theta_t,\theta_t^{(i,k)}\right)g_i\left(\tau_t^{(i,k)}\mid\theta_t\right)\right\|^2\leq\frac{\sigma^2}{NK}. \tag{23}$$

Then we have

$$\mathbb{E}\left\|\sum_{\tau=1}^t\beta z_\tau(1-\beta)^{t-\tau}\right\|^2\leq\beta^2\sum_{\tau=1}^t\mathbb{E}\left\|z_\tau\right\|^2(1-\beta)^{2(t+1-\tau)}\leq\frac{\beta\sigma^2}{NK}. \tag{24}$$

Plugging Eq. (19), Eq. (20), Eq. (22), and Eq. (24) into Eq. (18) yields

$$\mathbb{E}\left\|\mathcal{E}_t\right\|\leq(1-\beta)^t\left(\frac{\sigma}{\sqrt{NK}}+\frac{\beta C_wC_g\eta K}{2}+\frac{(2C_wC_g+L_g)\eta K}{2}+2\beta\sigma\right)+\frac{\eta K(C_wC_g+L_g)}{2\beta}$$

$$+\eta C_wC_g\sqrt{\frac{2K}{N\beta}}+\lambda L_g\sqrt{\frac{3}{NK\beta}}+\sigma\sqrt{\frac{\beta}{NK}}. \tag{25}$$

Summing the above inequality over $t$ yields

$$\frac{1}{T}\sum_{t=0}^{T-1}\mathbb{E}\left\|\mathcal{E}_t\right\|\leq\frac{1}{\beta T}\left(\frac{\sigma}{\sqrt{NK}}+\frac{\beta C_wC_g\eta K}{2}+\frac{(2C_wC_g+L_g)\eta K}{2}+2\beta\sigma\right)+\frac{\eta K(C_wC_g+L_g)}{2\beta}$$

$$+\eta C_wC_g\sqrt{\frac{2K}{N\beta}}+\lambda L_g\sqrt{\frac{3}{NK\beta}}+\sigma\sqrt{\frac{\beta}{NK}}. \tag{26}$$

## D.2 Proof of Lemma D.3

With variance reduction, we have

$$u_t^{(i,k)}=g_i\left(\tau_t^{(i,k)}\mid\theta_t^{(i,k)}\right)-\beta\left(c_{t-1}^{(i)}-c_{t-1}\right)$$

$$+ (1 - \beta) \left( u_{t-1} - w^{(i)} \left( \tau_t^{(i,k)} \mid \theta_{t-1}, \theta_t^{(i,k)} \right) g_i \left( \tau_t^{(i,k)} \mid \theta_{t-1} \right) \right).$$

Since $u_t = \frac{1}{NK} \sum_{i,k} u_t^{(i,k)}$, we have

$$\mathbb{E} \left[ \frac{1}{NK} \sum_{i,k} \left\| u_t^{(i,k)} - u_t \right\| \right]$$

$$\leq \frac{2}{NK} \sum_{i,k} \mathbb{E} \left\| g_i \left( \tau_t^{(i,k)} \mid \theta_t^{(i,k)} \right) - \beta c_{t-1}^{(i)} - (1-\beta) w^{(i)} \left( \tau_t^{(i,k)} \mid \theta_{t-1}, \theta_t^{(i,k)} \right) g_i \left( \tau_t^{(i,k)} \mid \theta_{t-1} \right) \right\|$$

$$\leq \frac{2}{NK} \sum_{i,k} \left( \beta \mathbb{E} \left\| g_i \left( \tau_t^{(i,k)} \mid \theta_t^{(i,k)} \right) - c_{t-1}^{(i)} \right\| + (1-\beta) \mathbb{E} \left\| g_i \left( \tau_t^{(i,k)} \mid \theta_t^{(i,k)} \right) \right. \right.$$

$$\left. \left. \mp w^{(i)} \left( \tau_t^{(i,k)} \mid \theta_t, \theta_t^{(i,k)} \right) g_i \left( \tau_t^{(i,k)} \mid \theta_t \right) - w^{(i)} \left( \tau_t^{(i,k)} \mid \theta_{t-1}, \theta_t^{(i,k)} \right) g_i \left( \tau_t^{(i,k)} \mid \theta_{t-1} \right) \right\| \right)$$

$$\leq \frac{2\beta}{NK} \sum_{i,k} \mathbb{E} \left\| g_i \left( \tau_t^{(i,k)} \mid \theta_t^{(i,k)} \right) - c_{t-1}^{(i)} \right\| + \frac{2(1-\beta)(L_g + 3C_w C_g)}{NK} \sum_{i,k} \mathbb{E} \left\| \theta_i^{t,k} - \theta^t \right\|$$

$$+ 2 L_g (1-\beta) \mathbb{E} \left\| \theta^t - \theta^{t-1} \right\|$$

$$\leq \frac{2\beta}{NK} \sum_{i,k} \mathbb{E} \left\| g_i \left( \tau_t^{(i,k)} \mid \theta_t^{(i,k)} \right) - c_{t-1}^{(i)} \right\| + \eta K (L_g + 3C_w C_g) + 2\lambda L_g.$$

Then we can derive that

$$\frac{2\beta}{NK} \sum_{i,k} \mathbb{E} \left\| g_i \left( \tau_t^{(i,k)} \mid \theta_t^{(i,k)} \right) - c_{t-1}^{(i)} \right\|$$

$$= \frac{2\beta}{NK} \sum_{i,k} \mathbb{E} \left\| g_i \left( \tau_t^{(i,k)} \mid \theta_t^{(i,k)} \right) \mp \nabla J_i(\theta_t^{(i,k)}) \mp \nabla J_i(\theta_t) \mp \nabla J_i(\theta_{t-1}) - c_{t-1}^{(i)} \right\|$$

$$= \frac{2\beta}{NK} \sum_{i,k} \mathbb{E} \left\| g_i \left( \tau_t^{(i,k)} \mid \theta_t^{(i,k)} \right) - \nabla J_i(\theta_t^{(i,k)}) \right.$$

$$+ g_i \left( \tau_t^{(i,k)} \mid \theta_t^{(i,k)} \right) - w^{(i)} \left( \tau_t^{(i,k)} \mid \theta_t, \theta_t^{(i,k)} \right) g_i \left( \tau_t^{(i,k)} \mid \theta_t \right)$$

$$+ w^{(i)} \left( \tau_t^{(i,k)} \mid \theta_t, \theta_t^{(i,k)} \right) g_i \left( \tau_t^{(i,k)} \mid \theta_t \right) - w^{(i)} \left( \tau_t^{(i,k)} \mid \theta_{t-1}, \theta_t^{(i,k)} \right) g_i \left( \tau_t^{(i,k)} \mid \theta_{t-1} \right)$$

$$\left. + \nabla J_i(\theta_{t-1}) - c_{t-1}^{(i)} \right\|$$

$$\leq 2\beta \left( \sigma + \frac{8 C_w C_g + 2 L_g}{NK} \sum_{i,k} \mathbb{E} \left\| \theta_t^{(i,k)} - \theta_t \right\| + (3 L_g + 6 C_w C_g) \mathbb{E} \left\| \theta_t - \theta_{t-1} \right\| \right)$$

$$+ \frac{2\beta}{N} \sum_i \mathbb{E} \left\| \nabla J_i(\theta_{t-1}) - c_{t-1}^{(i)} \right\|$$

$$\leq 2\beta \left( \sigma + 4\eta K C_w C_g + \eta K L_g + 6\lambda C_w C_g + 3\lambda L_g \right) + \frac{2\beta}{N} \sum_i \sqrt{\phi_i^{t-1}}. \tag{27}$$

From Lemma D.4, we know that

$$\sqrt{\phi_{t-1}^{(i)}} \leq \frac{\sqrt{2}\sigma}{\sqrt{K}} + \sqrt{\frac{2}{3}} \eta K (C_w C_g + L_g), \quad \forall i. \tag{28}$$

Thus, we have

$$\frac{2\beta}{NK} \sum_{i,k} \mathbb{E} \left\| g_i \left( \tau_t^{(i,k)} \mid \theta_t^{(i,k)} \right) - c_{t-1}^{(i)} \right\|$$

$$\leq 2\beta \left( \left(1 + \frac{2}{\sqrt{K}}\right) \sigma + 5\eta K (C_w C_g + L_g) + 6\lambda (C_w C_g + L_g) \right).$$

Then, we have

$$\mathbb{E}\left[ \frac{1}{NK} \sum_i \left\| u_t^{(i,k)} - u_t \right\| \right] \leq 2\beta \left( \left(1 + \frac{2}{\sqrt{K}}\right) \sigma + 5\eta K (C_w C_g + L_g) + 6\lambda (C_w C_g + L_g) \right)$$
$$+ \eta K (L_g + 3 C_w C_g) + 2\lambda L_g.$$

Summing the above inequality over $t$, we have

$$\frac{1}{NKT} \sum_{t=1}^{T} \mathbb{E}\left[ \sum_{i,k} \left\| u_t^{(i,k)} - u_t \right\| \right]$$

$$\leq 2\beta \left( \left(1 + \frac{2}{\sqrt{K}}\right) \sigma + 5\eta K (C_w C_g + L_g) + 6\lambda (C_w C_g + L_g) \right) + \eta K (L_g + 3 C_w C_g) + 2\lambda L_g. \tag{29}$$

### D.3 PROOF OF THEOREM 4.4

Set $\beta = \frac{\beta_0}{T^{\frac{2}{3}}}$, $\lambda = \frac{\lambda_0}{T^{\frac{2}{3}}}$, $\eta = \frac{1}{KT}$. From Lemma D.2, we know that

$$\frac{1}{T} \sum_{t=0}^{T-1} \mathbb{E} \left\| \nabla J(\theta_t) - u_t \right\|$$

$$\leq \frac{1}{\beta_0 T^{\frac{1}{3}}} \left( \frac{\beta_0 C_w C_g}{2 T^{\frac{5}{3}}} + \frac{2 C_w C_g + L_g}{2T} + \frac{2\beta_0 \sigma}{T^{\frac{2}{3}}} + \frac{\sigma}{\sqrt{NK}} \right) + \frac{C_w C_g + L_g}{2\beta_0 T^{\frac{1}{3}}}$$

$$+ \frac{C_w C_g}{T^{\frac{2}{3}}} \sqrt{\frac{2}{NK\beta_0}} + \frac{\lambda_0 L_g}{T^{\frac{1}{3}}} \sqrt{\frac{3}{NK\beta_0}} + \frac{\sigma}{T^{\frac{1}{3}}} \sqrt{\frac{\beta_0}{NK}}$$

$$\lesssim \frac{\sigma}{\beta_0 \sqrt{NK} T^{\frac{1}{3}}} + \frac{C_w C_g + L_g}{2\beta_0 T^{\frac{1}{3}}} + \frac{\lambda_0 L_g}{T^{\frac{1}{3}}} \sqrt{\frac{3}{NK\beta_0}} + \frac{\sigma}{T^{\frac{1}{3}}} \sqrt{\frac{\beta_0}{NK}}. \tag{30}$$

Similarly, from Lemma D.3, we have

$$\frac{1}{NKT} \sum_{t=1}^{T} \mathbb{E} \left[ \sum_{i,k} \left\| u_t^{(i,k)} - u_t \right\| \right]$$

$$\leq \frac{2\beta_0}{T^{\frac{2}{3}}} \left( \left(1 + \frac{2}{\sqrt{K}}\right) \sigma + \frac{5(L_g + C_w C_g)}{T} + \frac{6\lambda_0 (L_g + C_w C_g)}{T^{\frac{2}{3}}} \right)$$

$$+ \frac{L_g + 3 C_w C_g}{T} + \frac{2\lambda_0 L_g}{T^{\frac{2}{3}}}. \tag{31}$$

Since the initial optimality gap $\Delta := J(\theta^*) - J(\theta_0)$. Then we have $J(\theta_T) - J(\theta_0) \leq J(\theta^*) - J(\theta_0) = \Delta$. Plugging Eq. (30) and Eq. (31) into Eq. (12), we have

$$\frac{1}{T} \sum_{t=0}^{T-1} \mathbb{E} \left\| \nabla J(\theta_t) \right\| \lesssim \frac{\Delta}{\lambda_0 T^{\frac{1}{3}}} + \frac{2\sigma}{\beta_0 \sqrt{NK} T^{\frac{1}{3}}} + \frac{L_g + C_w C_g}{\beta_0 T^{\frac{1}{3}}} + \frac{2\sqrt{3} \lambda_0 L_g}{\sqrt{NK\beta_0} T^{\frac{1}{3}}} + \frac{2\sigma \sqrt{\beta_0}}{\sqrt{NK} T^{\frac{1}{3}}}$$

$$+ \frac{2\beta_0 \sigma}{T^{\frac{2}{3}}} + \frac{2\lambda_0 (L_g + L)}{T^{\frac{2}{3}}}. \tag{32}$$

Set $\beta_0 = (NK)^{\frac{1}{3}}$ and $\lambda_0 = (NK)^{\frac{1}{3}}$, we have

$$\frac{1}{T} \sum_{t=0}^{T-1} \mathbb{E} \left\| \nabla J(\theta_t) \right\| \lesssim \frac{\Delta}{\lambda_0 T^{\frac{1}{3}}} + \frac{2\sigma}{\beta_0 \sqrt{NK} T^{\frac{1}{3}}} + \frac{L_g + C_w C_g}{\beta_0 T^{\frac{1}{3}}} + \frac{2\sqrt{3} \lambda_0 L_g}{\sqrt{NK\beta_0} T^{\frac{1}{3}}} + \frac{2\sigma \sqrt{\beta_0}}{\sqrt{NK} T^{\frac{1}{3}}}$$

$$+ \frac{2\beta_0 \sigma}{T^{\frac{2}{3}}} + \frac{2\lambda_0(L_g + L)}{T^{\frac{2}{3}}}$$

$$\leq \mathcal{O}\left( \frac{\Delta + L_g + C_w C_g + \sigma}{(NKT)^{\frac{1}{3}}} + \frac{(L_g + L + \sigma)(NK)^{\frac{1}{3}}}{T^{\frac{2}{3}}} \right)$$

$$\leq \mathcal{O}\left( \frac{\Delta + \hat{L}}{(NKT)^{\frac{1}{3}}} + \frac{\hat{L}(NK)^{\frac{1}{3}}}{T^{\frac{2}{3}}} \right), \tag{33}$$

where $\hat{L} = L + L_g + C_w C_g + \sigma$. By setting $NK \leq \mathcal{O}(\sqrt{T})$, we have $\frac{(NK)^{\frac{1}{3}}}{T^{\frac{2}{3}}} \propto \mathcal{O}\left( (NKT)^{-\frac{1}{3}} \right)$, and thus

$$\frac{1}{T} \sum_{t=0}^{T-1} \mathbb{E} \left\| \nabla J(\theta_t) \right\| \leq \mathcal{O}\left( \frac{\Delta + \hat{L}}{(NKT)^{\frac{1}{3}}} \right).$$

## E CONVERGENCE ANALYSIS FOR PFEDPG-HA

---

**Algorithm 2** Description of PFEDPG-HA

---

1: **Input:** Initial model $\theta_{-1} = \theta_0$; control variates $c_{-1}^{(i)} = \frac{1}{K} \sum_{k=0}^{K-1} w^{(i)} \left( \tau_0^{(i,k)} \mid \theta_{-1}, \theta_0^{(i,k)}(\alpha) \right)$

$\quad \cdot g_i \left( \tau_0^{(i,k)} \mid \theta_{-1} \right)$ for all $i$; aggregated control variates $c_{-1} = \frac{1}{N} \sum_i c_{-1}^{(i)}$; momentum $u_{-1} = c_{-1}$;

$\quad$ local learning rate $\eta$; global learning rate $\lambda$; momentum parameter $\beta$

2: **for** $t = 0, 1, \ldots, T-1$ **do**

3: $\quad$ **for** each agent $i \in [N]$ **do**

4: $\quad\quad$ Initialize local model: $\theta_t^{(i,0)} = \theta_t$

5: $\quad\quad$ **for** $k = 0, 1, \ldots, K-1$ **do**

6: $\quad\quad\quad$ Sample $\alpha \sim \text{Uniform}[0, 1]$

7: $\quad\quad\quad$ Compute interpolated model: $\theta_t^{(i,k)}(\alpha) = \alpha \theta_{t-1} + (1-\alpha)\theta_t^{(i,k)}$

8: $\quad\quad\quad$ Sample trajectory $\tau_t^{(i,k)} \sim p^{(i)}(\tau \mid \theta_t^{(i,k)}(\alpha))$

9: $\quad\quad\quad$ Compute $v_t^{(i,k)} = \theta_t^{(i,k)} - \theta_{t-1}$

10: $\quad\quad\quad$ Compute correction term:

11: $\quad\quad\quad\quad \Lambda_t^{(i,k)} \triangleq \left\langle \nabla \log p \left( \tau_t^{(i,k)} \mid \theta_t^{(i,k)}(\alpha) \right), v_t^{(i,k)} \right\rangle g_i \left( \tau_t^{(i,k)} \mid \theta_t^{(i,k)}(\alpha) \right)$

$\quad\quad\quad\quad\quad + \nabla \left\langle g_i \left( \tau_t^{(i,k)} \mid \theta_t^{(i,k)}(\alpha) \right), v_t^{(i,k)} \right\rangle$

12: $\quad\quad\quad$ Compute update direction:

13: $\quad\quad\quad\quad u_t^{(i,k)} = \beta \left( w^{(i)} \left( \tau_t^{(i,k)} \mid \theta_t^{(i,k)}, \theta_t^{(i,k)}(\alpha) \right) g_i \left( \tau_t^{(i,k)} \mid \theta_t^{(i,k)} \right) - c_{t-1}^{(i)} + c_{t-1} \right)$

$\quad\quad\quad\quad\quad + (1-\beta) \left[ u_t + \Lambda_t^{(i,k)} \right]$

14: $\quad\quad\quad$ Update local model: $\theta_t^{(i,k+1)} = \theta_t^{(i,k)} + \eta \frac{u_t^{(i,k)}}{\left\| u_t^{(i,k)} \right\|}$

15: $\quad\quad$ **end for**

16: $\quad\quad$ Update control variate $c_t^{(i)} = \frac{1}{K} \sum_{k=0}^{K-1} w^{(i)} \left( \tau_t^{(i,k)} \mid \theta_t^{(i,k)}, \theta_t^{(i,k)}(\alpha) \right) g_i \left( \tau_t^{(i,k)} \mid \theta_t^{(i,k)} \right)$

17: $\quad\quad$ Send $\Delta_t^{(i)} = \theta_t^{(i,K)} - \theta_t$ and $c_t^{(i)}$ to server

18: $\quad$ **end for**

19: $\quad$ Server aggregates: $\hat{u}_t = \frac{1}{\eta N K} \sum_{i=1}^{N} \Delta_t^{(i)}$

20: $\quad$ Update global model: $\theta_{t+1} = \theta_t + \lambda \hat{u}_t$

21: $\quad$ Aggregate control variate $c_t = c_{t-1} + \frac{1}{N} \sum_{i=1}^{N} (c_t^{(i)} - c_{t-1}^{(i)})$

22: $\quad$ Aggregate momentum $u_t = \beta \left( \frac{1}{N} \sum_{i=1}^{N} (c_t^{(i)} - c_{t-1}^{(i)}) + c_{t-1} \right) + (1-\beta) u_{t-1}$

23: **end for**

---

According to the updating rule of PFEDPG-HA, we can rewrite $\Lambda_t^{(i,k)}$ as

$$\Lambda_t^{(i,k)} = \left( \nabla \log p^{(i)} \left( \tau_t^{(i,k)} \mid \theta_t^{(i,k)}(\alpha) \right) \right)^\top v_t^{(i,k)} \nabla \Phi_i \left( \tau_t^{(i,k)} \mid \theta_t^{(i,k)}(\alpha) \right)$$

$$+ \nabla^2 \Phi_i \left( \tau_t^{(i,k)} \mid \theta_t^{(i,k)}(\alpha) \right) v_t^{(i,k)}, \tag{34}$$

where $\Phi_i(\tau \mid \theta) = \sum_{h=0}^{H-1} \sum_{j=h}^{H-1} \gamma^j \mathcal{R}^{(i)}(s_j, a_j) \log \pi_\theta(a_h, s_h)$ and $v_t^{(i,k)} = \theta_t^{(i,k)} - \theta_{t-1}$. Note that

$$\mathbb{E}_{\alpha \sim \mathcal{U}[0,1], \tau \sim p^{(i)} \left( \tau \mid \theta_t^{(i,k)}(\alpha) \right)} \left[ \Lambda_t^{(i,k)} \right] = \nabla J \left( \theta_t^{(i,k)} \right) - \nabla J \left( \theta_{t-1} \right).$$

Moreover, we have $\Lambda_t^{(i,k)} := \widehat{\nabla}_i^2 \left( \theta_t^{(i,k)}(\alpha), \tau_t^{(i,k)}, v_t^{(i,k)} \right)$, where

$$\widehat{\nabla}_i^2 \left( \theta_t^{(i,k)}(\alpha), \tau_t^{(i,k)} \right) = \nabla \Phi_i \left( \tau_t^{(i,k)} \mid \theta_t^{(i,k)}(\alpha) \right) \nabla \log p^{(i)} \left( \tau_t^{(i,k)} \mid \theta_t^{(i,k)}(\alpha) \right)^\top$$

$$+ \nabla^2 \Phi_i \left( \tau_t^{(i,k)} \mid \theta_t^{(i,k)}(\alpha) \right),$$

and $\mathbb{E}_{\tau \sim p^{(i)}(\tau|\theta_t^{(i,k)}(\alpha))} \left[ \widehat{\nabla}_i^2 \left( \theta_t^{(i,k)}(\alpha), \tau \right) \right] = \nabla^2 J_i \left( \theta_t^{(i,k)}(\alpha) \right).$

**Proposition E.1.** *(Lemma 4.1 in Shen et al. (2019)) Under Assumption 4.1, for all $\theta$ and $i \in [N]$, we have*

$$\left\| \widehat{\nabla}_i^2(\theta, \tau) \right\| \leq \sqrt{\frac{H^2 G^4 R_{\max}^2 + M^2 R_{\max}^2}{(1-\gamma)^4}} = \widetilde{L},$$

*where $\tau$ is a trajectory sampled according to $p^{(i)}(\tau \mid \theta)$.*

**Lemma E.2.** *Under Assumptions 4.1, 4.2, and 4.3, the disparity $\frac{1}{T} \sum_{t=0}^{T-1} \mathbb{E} \|\nabla J(\theta_t) - u_t\|$ is upper bounded by:*

$$\frac{1}{T} \sum_{t=0}^{T-1} \mathbb{E} \|\nabla J(\theta_t) - u_t\|$$

$$\leq \frac{1}{\beta T} \left( \frac{\sigma}{\sqrt{NK}} + \frac{(C_w C_g + L_g + \widetilde{L})\eta K}{2} \right)$$

$$+ \sqrt{\frac{4\beta K}{3N}} \eta(L_g + 2C_w C_g) + \sqrt{8\beta} \lambda (C_w C_g + \alpha) + \sqrt{\frac{4\beta}{NK}} \sigma$$

$$+ 6C_w C_g \lambda + 3C_w C_g \eta K + \sqrt{2} L_g \eta K + 2\widetilde{L}(\eta K + \lambda) + 12(\eta K + \lambda)(L_g + C_w C_g)$$

$$+ \sqrt{\frac{\eta \sqrt{K} \sigma (L_g + C_w C_g)}{2\sqrt{N}}} + \sqrt{\frac{\sigma \lambda (L_g + C_w C_g)}{\sqrt{NK}}} + \sqrt{\frac{2\sigma C_w C_g \lambda}{\sqrt{NK}}} + \sqrt{\frac{(C_w C_g + L_g)\sigma \eta \sqrt{K}}{2\sqrt{N}}}.$$

$$(35)$$

**Lemma E.3.** *Under Assumptions 4.1, 4.2, and 4.3, the gradient dissimilarity $\frac{1}{NKT} \sum_{t=0}^{T-1} \mathbb{E} \left[ \sum_i \left\| u_t^{(i,k)} - u_t \right\| \right]$ is upper bounded by:*

$$\frac{1}{NKT} \sum_{t=1}^{T} \mathbb{E} \left[ \sum_{i,k} \left\| u_t^{(i,k)} - u_t \right\| \right]$$

$$\leq 2\beta \left( \left(1 + \frac{1}{K}\right) \sigma + \eta K(3C_w C_g + 3L_g + \widetilde{L}) + (L_g + 6C_w C_g + 2\widetilde{L})\lambda \right). \quad (36)$$

The proof of Lemma E.2 is given in Section E.1 and the proof of Lemma E.3 is given in Section E.2. Together, these results provide two key components toward establishing Theorem 4.5, with the detailed proof presented in Section E.3.

## E.1 PROOF OF LEMMA E.2

Since $\mathcal{E}_t := \nabla J(\theta_t) - u_t$, we have

$$\mathcal{E}_t = \nabla J(\theta_t) - \frac{\beta}{NK} \sum_{i,k} w^{(i)} \left( \tau_t^{(i,k)} \mid \theta_t^{(i,k)}, \theta_t^{(i,k)}(\alpha) \right) g_i \left( \tau_t^{(i,k)} \mid \theta_t^{(i,k)} \right) + \frac{\beta}{N} \sum_i (c_{t-1}^{(i)} - c_{t-1})$$

$$- (1-\beta) \left( u_{t-1} \mp \nabla J(\theta_{t-1}) + \frac{1}{NK} \sum_{i,k} \Lambda_t^{(i,k)} \right)$$

$$= (1-\beta)\mathcal{E}_{t-1}$$

$$+ \beta \underbrace{\left( \nabla J(\theta_t) - c_{t-1} - \frac{1}{NK} \sum_{i,k} \left( w^{(i)} \left( \tau_t^{(i,k)} \mid \theta_t^{(i,k)}, \theta_t^{(i,k)}(\alpha) \right) g_i \left( \tau_t^{(i,k)} \mid \theta_t^{(i,k)} \right) - c_{t-1}^{(i)} \right) \right)}_{:= \tilde{z}_t}$$

$$+ (1-\beta) \underbrace{\left( \frac{1}{NK} \sum_{i,k} \Lambda_t^{(i,k)} + \nabla J(\theta_t) - \nabla J(\theta_{t-1}) \right)}_{:= \tilde{q}_t}$$

$$= (1-\beta)^t \mathcal{E}_0 + \sum_{\tau=1}^{t} \tilde{q}_\tau (1-\beta)^{t+1-\tau} + \sum_{\tau=1}^{t} \beta \tilde{z}_\tau (1-\beta)^{t-\tau}. \tag{37}$$

Taking norms and expectations, we can get that

$$\mathbb{E}\left\|\mathcal{E}_t\right\| \leq (1-\beta)^t \mathbb{E}\left\|\mathcal{E}_0\right\| + \left(\mathbb{E}\left\|\sum_{\tau=1}^{t} \tilde{q}_\tau (1-\beta)^{t+1-\tau}\right\|^2\right)^{\frac{1}{2}} + \left(\mathbb{E}\left\|\sum_{\tau=1}^{t} \beta \tilde{z}_\tau (1-\beta)^{t-\tau}\right\|^2\right)^{\frac{1}{2}}. \tag{38}$$

Since $\theta_{-1} = \theta_0$, $c_{-1}^{(i)} = \frac{1}{K}\sum_{k=0}^{K-1} w^{(i)}\left(\tau_0^{(i,k)} \mid \theta_{-1}, \theta_0^{(i,k)}(\alpha)\right) g_i\left(\tau_0^{(i,k)} \mid \theta_{-1}\right)$, $c_{-1} = \frac{1}{N}\sum_i c_{-1}^{(i)}$, and $u_{-1} = c_{-1}$, we have

$$\mathbb{E}\left\|\mathcal{E}_0\right\| = \mathbb{E}\left\|\nabla J(\theta_0) - \frac{\beta}{NK}\sum_{i,k} w^{(i)}\left(\tau_0^{(i,k)} \mid \theta_{-1}, \theta_0^{(i,k)}(\alpha)\right) g_i\left(\tau_0^{(i,k)} \mid \theta_{-1}\right)\right.$$

$$+ \frac{1}{NK}\sum_{i,k}\left(\beta w^{(i)}\left(\tau_0^{(i,k)} \mid \theta_{-1}, \theta_0^{(i,k)}(\alpha)\right) g_i\left(\tau_0^{(i,k)} \mid \theta_{-1}\right)\right.$$

$$\left.\left. + (1-\beta)\Lambda_0^{(i,k)} - w^{(i)}\left(\tau_0^{(i,k)} \mid \theta_0^{(i,k)}, \theta_0^{(i,k)}(\alpha)\right) g_i\left(\tau_0^{(i,k)} \mid \theta_0^{(i,k)}\right)\right)\right\|$$

$$= \mathbb{E}\left\|\nabla J(\theta_{-1}) \mp \frac{1}{NK}\sum_{i,k} w^{(i)}\left(\tau_0^{(i,k)} \mid \theta_{-1}, \theta_0^{(i,k)}(\alpha)\right) g_i\left(\tau_0^{(i,k)} \mid \theta_{-1}\right)\right.$$

$$\left. + \frac{1}{NK}\sum_{i,k}(1-\beta)\Lambda_0^{(i,k)} - \frac{1}{NK}\sum_{i,k} w^{(i)}\left(\tau_0^{(i,k)} \mid \theta_0^{(i,k)}, \theta_0^{(i,k)}(\alpha)\right) g_i\left(\tau_0^{(i,k)} \mid \theta_0^{(i,k)}\right)\right\|$$

$$\leq \frac{\sigma}{\sqrt{NK}} + \frac{(1-\alpha)C_w C_g + L_g}{NK}\sum_{i,k}\mathbb{E}\left\|\theta_0^{(i,k)} - \theta_{-1}\right\| + \frac{\alpha C_w C_g}{NK}\sum_{i,k}\mathbb{E}\left\|\theta_0^{(i,k)} - \theta_{-1}\right\|$$

$$+ \frac{1-\beta}{NK}\sum_{i,k}\mathbb{E}\left\|\widehat{\nabla}_i^2\left(\theta_t^{(i,k)}, \tau_t^{(i,k)}\right) v_t^{(i,k)}\right\|$$

$$\leq \frac{\sigma}{\sqrt{NK}} + \frac{C_w C_g + L_g}{NK}\sum_{i,k}\mathbb{E}\left\|\theta_0^{(i,k)} - \theta_{-1}\right\| + \frac{(1-\beta)\widetilde{L}}{NK}\sum_{i,k}\mathbb{E}\left\|v_t^{(i,k)}\right\|$$

$$\leq \frac{\sigma}{\sqrt{NK}} + \frac{C_w C_g + L_g + (1-\beta)\widetilde{L}}{NK}\sum_{i,k}\mathbb{E}\left\|\theta_0^{(i,k)} - \theta_0\right\| + \frac{(1-\beta)\widetilde{L}}{NK}\sum_{i,k}\mathbb{E}\left\|\theta_0 - \theta_{-1}\right\|$$

$$\leq \frac{\sigma}{\sqrt{NK}} + \frac{(1-\beta)\widetilde{L}\eta K}{2} + \frac{(C_w C_g + L_g)\eta K}{2}. \tag{39}$$

Since

$$\mathbb{E}\left[\tilde{z}_t \mid \mathcal{F}_t\right]$$

$$= \mathbb{E}_{\{\tau_t^{(i,k)}\}, \forall i,k}\left[\tilde{z}_t\right]$$

$$= \mathbb{E}_{\{\tau_t^{(i,k)}\}, \forall i,k}\left[\nabla J(\theta_t) - c_{t-1} - \frac{1}{NK}\sum_{i,k}\left(w^{(i)}\left(\tau_t^{(i,k)} \mid \theta_t^{(i,k)}, \theta_t^{(i,k)}(\alpha)\right) g_i\left(\tau_t^{(i,k)} \mid \theta_t^{(i,k)}\right) - c_{t-1}^{(i)}\right)\right]$$

$$= \nabla J(\theta_t) - \frac{1}{NK}\sum_{i,k}\nabla J_i(\theta_t^{(i,k)}). \tag{40}$$

Then, for any $0 \leq t_1 < t_2 \leq T-1$, we have

$$\mathbb{E}\left\langle \tilde{z}_{t_1}, \tilde{z}_{t_2}\right\rangle$$

$$= \mathbb{E} \left\langle \tilde{z}_{t_1}, \mathbb{E}\left[\tilde{z}_{t_2} \mid \mathcal{F}_{t_2}\right] \right\rangle$$

$$= \mathbb{E} \left\langle \frac{1}{NK} \sum_{i,k} \left( \nabla J_i(\theta_{t_1}) - \nabla J_i(\theta_{t_1}^{(i,k)}) \right), \frac{1}{NK} \sum_{i,k} \left( \nabla J_i(\theta_{t_2}) - \nabla J_i(\theta_{t_2}^{(i,k)}) \right) \right\rangle$$

$$+ \mathbb{E} \left\langle \frac{1}{NK} \sum_{i,k} \left( \nabla J_i(\theta_{t_1}^{(i,k)}) - w^{(i)} \left( \tau_t^{(i,k)} \mid \theta_t^{(i,k)}, \theta_t^{(i,k)}(\alpha) \right) g_i \left( \tau_{t_1}^{(i,k)} \mid \theta_{t_1}^{(i,k)} \right) \right), \right.$$

$$\left. \frac{1}{NK} \sum_{i,k} \left( \nabla J_i(\theta_{t_2}) - \nabla J_i(\theta_{t_2}^{(i,k)}) \right) \right\rangle$$

$$= \mathbb{E} \left\langle \frac{1}{NK} \sum_{i,k} \left( w^{(i)} \left( \tau_{t_1}^{(i,k)} \mid \theta_{t_1}, \theta_{t_1}^{(i,k)}(\alpha) \right) g_i \left( \tau_{t_1}^{(i,k)} \mid \theta_{t_1} \right) \right.\right.$$

$$\left. - w^{(i)} \left( \tau_{t_1}^{(i,k)} \mid \theta_{t_1}^{(i,k)}, \theta_{t_1}^{(i,k)}(\alpha) \right) g_i \left( \tau_{t_1}^{(i,k)} \mid \theta_{t_1}^{(i,k)} \right) \right),$$

$$\frac{1}{NK} \sum_{i,k} \left( w^{(i)} \left( \tau_{t_2}^{(i,k)} \mid \theta_{t_2}, \theta_{t_2}^{(i,k)}(\alpha) \right) g_i \left( \tau_{t_2}^{(i,k)} \mid \theta_{t_2} \right) \right.$$

$$\left.\left. - w^{(i)} \left( \tau_{t_2}^{(i,k)} \mid \theta_{t_2}^{(i,k)}, \theta_{t_2}^{(i,k)}(\alpha) \right) g_i \left( \tau_{t_2}^{(i,k)} \mid \theta_{t_2}^{(i,k)} \right) \right) \right\rangle$$

$$+ \mathbb{E} \left\langle \frac{1}{NK} \sum_{i,k} \left( \nabla J_i(\theta_{t_1}^{(i,k)}) - w^{(i)} \left( \tau_{t_1}^{(i,k)} \mid \theta_{t_1}^{(i,k)}, \theta_{t_1}^{(i,k)}(\alpha) \right) g_i \left( \tau_{t_1}^{(i,k)} \mid \theta_{t_1}^{(i,k)} \right) \right), \right.$$

$$\frac{1}{NK} \sum_{i,k} \left( w^{(i)} \left( \tau_{t_2}^{(i,k)} \mid \theta_{t_2}, \theta_{t_2}^{(i,k)}(\alpha) \right) g_i \left( \tau_{t_2}^{(i,k)} \mid \theta_{t_2} \right) \right.$$

$$\left.\left. - w^{(i)} \left( \tau_{t_2}^{(i,k)} \mid \theta_{t_2}^{(i,k)}, \theta_{t_2}^{(i,k)}(\alpha) \right) g_i \left( \tau_{t_2}^{(i,k)} \mid \theta_{t_2}^{(i,k)} \right) \right) \right\rangle$$

$$\leq \frac{6C_w^2 C_g^2}{NK} \sum_{i,k} \mathbb{E} \left\| \theta_{t_1} \mp \theta_{t_1-1} - (\alpha \theta_{t_1-1} + (1-\alpha)\theta_{t_1}^{(i,k)}) \right\|^2 + \frac{6L_g^2}{NK} \sum_{i,k} \mathbb{E} \left\| \theta_{t_1}^{(i,k)} - \theta_{t_1} \right\|^2$$

$$+ \frac{6C_w^2 C_g^2}{NK} \sum_{i,k} \mathbb{E} \left\| \theta_{t_1}^{(i,k)} - (\alpha \theta_{t_1-1} + (1-\alpha)\theta_{t_1}^{(i,k)}) \right\|^2$$

$$+ \frac{\sigma}{\sqrt{NK}} \left( \frac{C_w C_g}{NK} \sum_{i,k} \mathbb{E} \left\| \theta_{t_2} \mp \theta_{t_2-1} - (\alpha \theta_{t_2-1} + (1-\alpha)\theta_{t_2}^{(i,k)}) \right\| \right.$$

$$\left. + \frac{L_g}{NK} \sum_{i,k} \mathbb{E} \left\| \theta_{t_2}^{(i,k)} - \theta_{t_2} \right\| + \frac{C_w C_g}{NK} \sum_{i,k} \mathbb{E} \left\| \theta_{t_2}^{(i,k)} - (\alpha \theta_{t_2-1} + (1-\alpha)\theta_{t_2}^{(i,k)}) \right\|^2 \right)$$

$$\leq \frac{(36-6\alpha)C_w^2 C_g^2}{NK} \sum_{i,k} \mathbb{E} \left\| \theta_{t_1} - \theta_{t_1-1} \right\|^2 + \frac{(18-6\alpha)C_w^2 C_g^2}{NK} \sum_{i,k} \mathbb{E} \left\| \theta_{t_1}^{(i,k)} - \theta_{t_1} \right\|^2$$

$$+ \frac{6L_g^2}{NK} \sum_{i,k} \mathbb{E} \left\| \theta_{t_1}^{(i,k)} - \theta_{t_1} \right\|^2 + \frac{\sigma}{\sqrt{NK}} \left( \frac{2C_w C_g}{NK} \sum_{i,k} \mathbb{E} \left\| \theta_{t_2} - \theta_{t_2-1} \right\| \right.$$

$$\left. + \frac{C_w C_g}{NK} \sum_{i,k} \mathbb{E} \left\| \theta_{t_2}^{(i,k)} - \theta_{t_2} \right\| + \frac{L_g}{NK} \sum_{i,k} \mathbb{E} \left\| \theta_{t_2}^{(i,k)} - \theta_{t_2} \right\| \right)$$

$$\leq (36-6\alpha)C_w^2 C_g^2 \lambda^2 + (6-2\alpha)C_w^2 C_g^2 \eta^2 K^2 + 2L_g^2 \eta^2 K^2$$

$$+ \frac{2\sigma C_w C_g \lambda}{\sqrt{NK}} + \frac{C_w C_g \sigma \eta \sqrt{K}}{2\sqrt{N}} + \frac{L_g \sigma \eta \sqrt{K}}{2\sqrt{N}}. \tag{41}$$

Further, we have

$$
\mathbb{E}\|\tilde{z}_t\|^2 \leq \mathbb{E}\left\|\nabla J(\theta_t) - \frac{1}{NK}\sum_{i,k} w^{(i)}\left(\tau_t^{(i,k)} \mid \theta_t^{(i,k)}, \theta_t^{(i,k)}(\alpha)\right) g_i\left(\tau_t^{(i,k)} \mid \theta_t^{(i,k)}\right)\right\|^2
$$

$$
= \mathbb{E}\left\|\frac{1}{NK}\sum_{i,k} \nabla J_i(\theta_t) \mp w^{(i)}\left(\tau_t^{(i,k)} \mid \theta_t, \theta_t^{(i,k)}(\alpha)\right) g_i\left(\tau_t^{(i,k)} \mid \theta_t\right)\right.
$$

$$
\left. - w^{(i)}\left(\tau_t^{(i,k)} \mid \theta_t^{(i,k)}, \theta_t^{(i,k)}(\alpha)\right) g_i\left(\tau_t^{(i,k)} \mid \theta_t^{(i,k)}\right)\right\|^2
$$

$$
\leq 4\left(\frac{2C_w^2 C_g^2 + L_g^2}{N^2 K^2}\sum_{i,k} \mathbb{E}\left\|\theta_t^{(i,k)} - \theta_t\right\|^2 + 2(C_w^2 C_g^2 + \alpha^2)\mathbb{E}\|\theta_t - \theta_{t-1}\|^2 + \frac{\sigma^2}{NK}\right)
$$

$$
\leq \frac{4}{3N}\eta^2 K(L_g^2 + 2C_w^2 C_g^2) + 8\lambda^2(C_w^2 C_g^2 + \alpha^2) + \frac{4\sigma^2}{NK}. \tag{42}
$$

Then we have

$$
\mathbb{E}\left\|\sum_{\tau=1}^t \beta\tilde{z}_\tau(1-\beta)^{t-\tau}\right\|^2
$$

$$
\leq \beta\mathbb{E}\|\tilde{z}_\tau\|^2 + \mathbb{E}\langle\tilde{z}_{\tau_1}, \tilde{z}_{\tau_2}\rangle
$$

$$
\leq \frac{4\beta}{3N}\eta^2 K(L_g^2 + 2C_w^2 C_g^2) + 8\beta\lambda^2(C_w^2 C_g^2 + \alpha^2) + \frac{4\beta\sigma^2}{NK}
$$

$$
+ (36 - 6\alpha)C_w^2 C_g^2\lambda^2 + (6 - 2\alpha)C_w^2 C_g^2\eta^2 K^2 + 2L_g^2\eta^2 K^2
$$

$$
+ \frac{2\sigma C_w C_g\lambda}{\sqrt{NK}} + \frac{C_w C_g\sigma\eta\sqrt{K}}{2\sqrt{N}} + \frac{L_g\sigma\eta\sqrt{K}}{2\sqrt{N}}. \tag{43}
$$

Since $\beta \leq 1$, taking square root on both sides of the above inequality yields

$$
\left(\mathbb{E}\left\|\sum_{\tau=1}^t \beta\tilde{z}_\tau(1-\beta)^{t-\tau}\right\|^2\right)^{\frac{1}{2}}
$$

$$
\leq \sqrt{\frac{4\beta K}{3N}}\eta(L_g + 2C_w C_g) + \sqrt{8\beta}\lambda(C_w C_g + \alpha) + \sqrt{\frac{4\beta}{NK}}\sigma
$$

$$
+ \sqrt{36 - 6\alpha}C_w C_g\lambda + \sqrt{6 - 2\alpha}C_w C_g\eta K + \sqrt{2}L_g\eta K
$$

$$
+ \sqrt{\frac{2\sigma C_w C_g\lambda}{\sqrt{NK}}} + \sqrt{\frac{(C_w C_g + L_g)\sigma\eta\sqrt{K}}{2\sqrt{N}}}. \tag{44}
$$

Similarly, for any $t$, we have

$$
\mathbb{E}[\tilde{q}_t \mid \mathcal{F}_t] = \mathbb{E}_{\{\tau_t^{(i,k)}\}, \forall i,k}\left[\Lambda_t^{(i,k)}\right] = \nabla J_i(\theta_t^{(i,k)}) - \nabla J(\theta_{t-1}). \tag{45}
$$

Then, for any $0 \leq t_1 < t_2 \leq T - 1$, we have

$$
\mathbb{E}\langle\tilde{q}_{t_1}, \tilde{q}_{t_2}\rangle
$$

$$
= \mathbb{E}\langle\tilde{q}_{t_1}, \mathbb{E}[\tilde{q}_{t_2} \mid \mathcal{F}_{t_2}]\rangle
$$

$$
= \mathbb{E}\left\langle\frac{1}{NK}\sum_{i,k}\left(\nabla J_i(\theta_{t_1-1}) - \nabla J_i(\theta_{t_1}^{(i,k)})\right), \frac{1}{NK}\sum_{i,k}\left(\nabla J_i(\theta_{t_2-1}) - \nabla J_i(\theta_{t_2}^{(i,k)})\right)\right\rangle
$$

$$
+ \mathbb{E}\left\langle\frac{1}{NK}\sum_{i,k}\left(\nabla J_i(\theta_{t_1}^{(i,k)}) - w^{(i)}\left(\tau_{t_1}^{(i,k)} \mid \theta_{t_1}^{(i,k)}, \theta_{t_1}^{(i,k)}(\alpha)\right) g_i\left(\tau_{t_1}^{(i,k)} \mid \theta_{t_1}^{(i,k)}\right)\right),
$$

$$\frac{1}{NK} \sum_{i,k} \left( \nabla J_i(\theta_{t_2-1}) - \nabla J_i(\theta_{t_2}^{(i,k)}) \right) \Big\rangle$$

$$= \mathbb{E} \Big\langle \frac{1}{NK} \sum_{i,k} \left( w^{(i)} \left( \tau_{t_1}^{(i,k)} \mid \theta_{t_1-1}, \theta_{t_1}^{(i,k)}(\alpha) \right) g_i \left( \tau_{t_1}^{(i,k)} \mid \theta_{t_1-1} \right) \right.$$

$$\left. - w^{(i)} \left( \tau_{t_1}^{(i,k)} \mid \theta_{t_1}^{(i,k)}, \theta_{t_1}^{(i,k)}(\alpha) \right) g_i \left( \tau_{t_1}^{(i,k)} \mid \theta_{t_1}^{(i,k)} \right) \right),$$

$$\frac{1}{NK} \sum_{i,k} \left( w^{(i)} \left( \tau_{t_2}^{(i,k)} \mid \theta_{t_2-1}, \theta_{t_2}^{(i,k)}(\alpha) \right) g_i \left( \tau_{t_2}^{(i,k)} \mid \theta_{t_2-1} \right) \right.$$

$$\left. - w^{(i)} \left( \tau_{t_2}^{(i,k)} \mid \theta_{t_2}^{(i,k)}, \theta_{t_2}^{(i,k)}(\alpha) \right) g_i \left( \tau_{t_2}^{(i,k)} \mid \theta_{t_2}^{(i,k)} \right) \right) \Big\rangle$$

$$+ \mathbb{E} \Big\langle \frac{1}{NK} \sum_{i,k} \left( \nabla J_i(\theta_{t_1}^{(i,k)}) - w^{(i)} \left( \tau_{t_1}^{(i,k)} \mid \theta_{t_1}^{(i,k)}, \theta_{t_1}^{(i,k)}(\alpha) \right) g_i \left( \tau_{t_1}^{(i,k)} \mid \theta_{t_1}^{(i,k)} \right) \right),$$

$$\frac{1}{NK} \sum_{i,k} \left( w^{(i)} \left( \tau_{t_2-1}^{(i,k)} \mid \theta_{t_2-1}, \theta_{t_2}^{(i,k)}(\alpha) \right) g_i \left( \tau_{t_2}^{(i,k)} \mid \theta_{t_2-1} \right) \right.$$

$$\left. - w^{(i)} \left( \tau_{t_2}^{(i,k)} \mid \theta_{t_2}^{(i,k)}, \theta_{t_2}^{(i,k)}(\alpha) \right) g_i \left( \tau_{t_2}^{(i,k)} \mid \theta_{t_2}^{(i,k)} \right) \right) \Big\rangle$$

$$\leq \frac{3L_g^2 + 3C_w^2 C_g^2}{NK} \sum_{i,k} \mathbb{E} \left\| \theta_{t_1}^{(i,k)} - \theta_{t_1-1} \right\|^2 + \frac{3L_g^2 + 3C_w^2 C_g^2}{NK} \sum_{i,k} \mathbb{E} \left\| \theta_{t_2}^{(i,k)} - \theta_{t_2-1} \right\|^2$$

$$\leq \frac{6L_g^2 + 6C_w^2 C_g^2}{NK} \sum_{i,k} \mathbb{E} \left\| \theta_{t_1}^{(i,k)} - \theta_{t_1} \right\|^2 + \frac{6L_g^2 + 6C_w^2 C_g^2}{NK} \sum_{i,k} \mathbb{E} \left\| \theta_{t_1} - \theta_{t_1-1} \right\|^2$$

$$+ \frac{6L_g^2 + 6C_w^2 C_g^2}{NK} \sum_{i,k} \mathbb{E} \left\| \theta_{t_2}^{(i,k)} - \theta_{t_2} \right\|^2 + \frac{6L_g^2 + 6C_w^2 C_g^2}{NK} \sum_{i,k} \mathbb{E} \left\| \theta_{t_2} - \theta_{t_2-1} \right\|^2$$

$$+ \frac{\sigma}{\sqrt{NK}} \frac{L_g + C_w C_g}{NK} \sum_{i,k} \mathbb{E} \left\| \theta_{t_2}^{(i,k)} - \theta_{t_2} \right\| + \frac{\sigma}{\sqrt{NK}} \frac{L_g + C_w C_g}{NK} \sum_{i,k} \mathbb{E} \left\| \theta_{t_2} - \theta_{t_2-1} \right\|$$

$$\leq 4\eta^2 K^2 (L_g^2 + C_w^2 C_g^2) + 12\lambda^2 (L_g^2 + C_w^2 C_g^2) + \frac{\eta\sqrt{K}\sigma(L_g + C_w C_g)}{2\sqrt{N}} + \frac{\sigma\lambda(L_g + C_w C_g)}{\sqrt{NK}}. \tag{46}$$

Further, we have

$$\mathbb{E} \left\| \tilde{q}_t \right\|^2 \leq \mathbb{E} \left\| \frac{1}{NK} \sum_{i,k} \Lambda_t^{(i,k)} + \nabla J(\theta_t) - \nabla J(\theta_{t-1}) \right\|^2$$

$$\leq \frac{1}{N^2 K^2} \sum_{i,k} \mathbb{E} \left\| \Lambda_t^{(i,k)} \right\|^2$$

$$\leq \frac{1}{N^2 K^2} \sum_{i,k} \mathbb{E} \left\| \widehat{\nabla}_i^2 \left( \theta_{r,k}^{(i)}, \tau_{r,k}^{(i)} \right) v_{r,k}^{(i)} \right\|^2$$

$$\leq \frac{1}{N^2 K^2} \sum_{i,k} \widetilde{L}^2 \, \mathbb{E} \left\| v_{r,k}^{(i)} \right\|^2$$

$$\leq \frac{2\widetilde{L}^2}{NK} \sum_{i,k} \mathbb{E} \left\| \theta_{t_2}^{(i,k)} - \theta_{t_2} \right\|^2 + \frac{2\widetilde{L}^2}{NK} \sum_{i,k} \mathbb{E} \left\| \theta_{t_2-1} - \theta_{t_2} \right\|^2$$

$$\leq \frac{2}{3} \eta^2 K^2 \widetilde{L}^2 + 2\widetilde{L}^2 \lambda^2. \tag{47}$$

Then we can get that

$$
\mathbb{E} \left\| \sum_{\tau=1}^{t} \beta \tilde{q}_\tau (1-\beta)^{t+1-\tau} \right\|^2
$$

$$
\leq \beta \mathbb{E} \|\tilde{q}_\tau\|^2 + \mathbb{E} \langle \tilde{q}_{\tau_1}, \tilde{q}_{\tau_2} \rangle
$$

$$
\leq \frac{2}{3} \eta^2 K^2 \widetilde{L}^2 + 2\widetilde{L}^2 \lambda^2 + 4\eta^2 K^2 (L_g^2 + C_w^2 C_g^2) + 12\lambda^2 (L_g^2 + C_w^2 C_g^2)
$$

$$
+ \frac{\eta \sqrt{K} \sigma (L_g + C_w C_g)}{2\sqrt{N}} + \frac{\sigma \lambda (L_g + C_w C_g)}{\sqrt{NK}}. \tag{48}
$$

Since $\beta \leq 1$, taking square root on both sides of the above inequality yields

$$
\left( \mathbb{E} \left\| \sum_{\tau=1}^{t} \tilde{q}_\tau (1-\beta)^{t+1-\tau} \right\|^2 \right)^{\frac{1}{2}} \leq 2\widetilde{L}(\eta K + \lambda) + 12(\eta K + \lambda)(L_g + C_w C_g)
$$

$$
+ \sqrt{\frac{\eta \sqrt{K} \sigma (L_g + C_w C_g)}{2\sqrt{N}}} + \sqrt{\frac{\sigma \lambda (L_g + C_w C_g)}{\sqrt{NK}}}, \tag{49}
$$

where we use the inequality $\sqrt{a+b} \leq \sqrt{a} + \sqrt{b}$ for any $a, b \geq 0$.

Plugging Eq. (39), Eq. (44), and Eq. (49) into Eq. (38), we have

$$
\mathbb{E} \left\| \mathcal{E}^t \right\|
$$

$$
\leq (1-\beta)^t \left( \frac{\sigma}{\sqrt{NK}} + \frac{(1-\beta)\widetilde{L}\eta K}{2} + \frac{(C_w C_g + L_g)\eta K}{2} \right)
$$

$$
+ \sqrt{\frac{4\beta K}{3N}} \eta (L_g + 2C_w C_g) + \sqrt{8\beta} \lambda (C_w C_g + \alpha) + \sqrt{\frac{4\beta}{NK}} \sigma
$$

$$
+ \sqrt{36 - 6\alpha} C_w C_g \lambda + \sqrt{6 - 2\alpha} C_w C_g \eta K + \sqrt{2} L_g \eta K
$$

$$
+ 2\widetilde{L}(\eta K + \lambda) + 12(\eta K + \lambda)(L_g + C_w C_g)
$$

$$
+ \sqrt{\frac{2\sigma C_w C_g \lambda}{\sqrt{NK}}} + \sqrt{\frac{(C_w C_g + L_g)\sigma \eta \sqrt{K}}{2\sqrt{N}}} + \sqrt{\frac{\eta \sqrt{K} \sigma (L_g + C_w C_g)}{2\sqrt{N}}} + \sqrt{\frac{\sigma \lambda (L_g + C_w C_g)}{\sqrt{NK}}}. \tag{50}
$$

Summing the above inequality over $t$ yields

$$
\frac{1}{T} \sum_{t=0}^{T-1} \mathbb{E} \left\| \mathcal{E}^t \right\|
$$

$$
\leq \frac{1}{\beta T} \left( \frac{\sigma}{\sqrt{NK}} + \frac{(1-\beta)\widetilde{L}\eta K}{2} + \frac{(C_w C_g + L_g)\eta K}{2} \right)
$$

$$
+ \sqrt{\frac{4\beta K}{3N}} \eta (L_g + 2C_w C_g) + \sqrt{8\beta} \lambda (C_w C_g + \alpha) + \sqrt{\frac{4\beta}{NK}} \sigma
$$

$$
+ \sqrt{36 - 6\alpha} C_w C_g \lambda + \sqrt{6 - 2\alpha} C_w C_g \eta K + \sqrt{2} L_g \eta K
$$

$$
+ 2\widetilde{L}(\eta K + \lambda) + 12(\eta K + \lambda)(L_g + C_w C_g)
$$

$$
+ \sqrt{\frac{2\sigma C_w C_g \lambda}{\sqrt{NK}}} + \sqrt{\frac{(C_w C_g + L_g)\sigma \eta \sqrt{K}}{2\sqrt{N}}} + \sqrt{\frac{\eta \sqrt{K} \sigma (L_g + C_w C_g)}{2\sqrt{N}}} + \sqrt{\frac{\sigma \lambda (L_g + C_w C_g)}{\sqrt{NK}}}. \tag{51}
$$

### E.2 PROOF OF LEMMA E.3

Recall that $u_t^{(i,k)} = \beta \left( w^{(i)} \left( \tau_t^{(i,k)} \mid \theta_t^{(i,k)}, \theta_t^{(i,k)}(\alpha) \right) g_i \left( \tau_t^{(i,k)} \mid \theta_t^{(i,k)} \right) - c_{t-1}^{(i)} + c_{t-1} \right) + (1 - \beta)(u_{t-1} + \Lambda_t^{(i,k)})$, and

$$
\begin{aligned}
u_t &= \frac{1}{NK} \sum_{i,k} u_t^{(i,k)} \\
&= \beta \left( \frac{1}{N} \sum_i \left( \frac{1}{K} \sum_{k=0}^{K-1} w^{(i)} \left( \tau_t^{(i,k)} \mid \theta_t^{(i,k)}, \theta_t^{(i,k)}(\alpha) \right) g_i \left( \tau_t^{(i,k)} \mid \theta_t^{(i,k)} \right) - c_{t-1}^{(i)} \right) + c_{t-1} \right) \\
&\quad + (1 - \beta) \left( u_{t-1} + \frac{1}{NK} \sum_{i,k} \Lambda_t^{(i,k)} \right).
\end{aligned}
\tag{52}
$$

Then, we have

$$
\begin{aligned}
&\mathbb{E} \left[ \frac{1}{NK} \sum_{i,k} \left\| u_t^{(i,k)} - u_t \right\| \right] \\
&= \mathbb{E} \left[ \frac{\beta}{NK} \sum_{i,k} \left\| w^{(i)} \left( \tau_t^{(i,k)} \mid \theta_t^{(i,k)}, \theta_t^{(i,k)}(\alpha) \right) g_i \left( \tau_t^{(i,k)} \mid \theta_t^{(i,k)} \right) - c_{t-1}^{(i)} \right. \right. \\
&\qquad \left. - \frac{1}{N} \sum_i \left( \frac{1}{K} \sum_{k=0}^{K-1} w^{(i)} \left( \tau_t^{(i,k)} \mid \theta_t^{(i,k)}, \theta_t^{(i,k)}(\alpha) \right) g_i \left( \tau_t^{(i,k)} \mid \theta_t^{(i,k)} \right) - c_{t-1}^{(i)} \right) \right\| \\
&\qquad \left. + \frac{1-\beta}{NK} \sum_{i,k} \left\| \Lambda_t^{(i,k)} - \frac{1}{NK} \sum_{i,k} \Lambda_t^{(i,k)} \right\| \right] \\
&\leq \frac{2\beta}{NK} \sum_{i,k} \mathbb{E} \left\| w^{(i)} \left( \tau_t^{(i,k)} \mid \theta_t^{(i,k)}, \theta_t^{(i,k)}(\alpha) \right) g_i \left( \tau_t^{(i,k)} \mid \theta_t^{(i,k)} \right) - c_{t-1}^{(i)} \right\| \\
&\qquad + \frac{2(1-\beta)}{NK} \sum_{i,k} \mathbb{E} \left\| \Lambda_t^{(i,k)} \right\| \\
&\leq \frac{2\beta}{NK} \sum_{i,k} \mathbb{E} \left\| w^{(i)} \left( \tau_t^{(i,k)} \mid \theta_t^{(i,k)}, \theta_t^{(i,k)}(\alpha) \right) g_i \left( \tau_t^{(i,k)} \mid \theta_t^{(i,k)} \right) \right. \\
&\qquad \left. \mp \nabla J_i(\theta_t^{(i,k)}) \mp \nabla J_i(\theta_t) \mp \nabla J_i(\theta_{t-1}) - c_{t-1}^{(i)} \right\| \\
&\qquad + \frac{2\widetilde{L}(1-\beta)}{NK} \sum_{i,k} \mathbb{E} \left\| \theta_t^{(i,k)} - \theta_t \right\| + 2\widetilde{L}(1-\beta) \mathbb{E} \left\| \theta_t - \theta_{t-1} \right\| \\
&= \frac{2\beta}{NK} \sum_{i,k} \mathbb{E} \left\| w^{(i)} \left( \tau_t^{(i,k)} \mid \theta_t^{(i,k)}, \theta_t^{(i,k)}(\alpha) \right) g_i \left( \tau_t^{(i,k)} \mid \theta_t^{(i,k)} \right) - \nabla J_i(\theta_t^{(i,k)}) \right. \\
&\qquad + w^{(i)} \left( \tau_t^{(i,k)} \mid \theta_t^{(i,k)}, \theta_t^{(i,k)}(\alpha) \right) g_i \left( \tau_t^{(i,k)} \mid \theta_t^{(i,k)} \right) - w^{(i)} \left( \tau_t^{(i,k)} \mid \theta_t, \theta_t^{(i,k)}(\alpha) \right) g_i \left( \tau_t^{(i,k)} \mid \theta_t \right) \\
&\qquad + w^{(i)} \left( \tau_t^{(i,k)} \mid \theta_t, \theta_t^{(i,k)}(\alpha) \right) g_i \left( \tau_t^{(i,k)} \mid \theta_t \right) - w^{(i)} \left( \tau_t^{(i,k)} \mid \theta_{t-1}, \theta_t^{(i,k)}(\alpha) \right) g_i \left( \tau_t^{(i,k)} \mid \theta_{t-1} \right) \\
&\qquad \left. + \nabla J_i(\theta_{t-1}) - c_{t-1}^{(i)} \right\| \\
&\qquad + \frac{2\widetilde{L}(1-\beta)}{NK} \sum_{i,k} \mathbb{E} \left\| \theta_t^{(i,k)} - \theta_t \right\| + 2\widetilde{L}(1-\beta) \mathbb{E} \left\| \theta_t - \theta_{t-1} \right\|
\end{aligned}
$$

$$\leq 2\beta \left( \sigma + \frac{(3-2\alpha)C_wC_g + 2L_g}{NK} \sum_{i,k} \mathbb{E} \left\| \theta_t^{(i,k)} - \theta_t \right\| + (L_g + (5-3\alpha)C_wC_g)\mathbb{E} \left\| \theta_t - \theta_{t-1} \right\| \right)$$

$$+ \frac{2\beta}{N} \sum_i \widetilde{\phi}_t^{(i)} + \frac{2\widetilde{L}(1-\beta)}{NK} \sum_{i,k} \mathbb{E} \left\| \theta_t^{(i,k)} - \theta_t \right\| + 2\widetilde{L}(1-\beta)\mathbb{E} \left\| \theta_t - \theta_{t-1} \right\|$$

$$\overset{(a)}{\leq} 2\beta \left( \sigma + \frac{\eta K((3-2\alpha)C_wC_g + 2L_g)}{2} + (L_g + (5-3\alpha)C_wC_g)\lambda \right.$$

$$\left. + \widetilde{L}(1-\beta)\eta K + 2\widetilde{L}(1-\beta)\lambda \right) + \frac{2\beta\sigma}{K} + \beta\eta K(C_wC_g + L_g) + 2\beta\lambda C_wC_g$$

$$\leq 2\beta \left( \left( 1 + \frac{1}{K} \right) \sigma + \eta K(3C_wC_g + 3L_g + \widetilde{L}) + (L_g + 6C_wC_g + 2\widetilde{L})\lambda \right), \tag{53}$$

where (a) is derived by

$$\widetilde{\phi}_t^{(i)} = \mathbb{E} \left\| \frac{1}{K} \sum_k \left( \nabla J_i(\theta_t) - w^{(i)} \left( \tau_t^{(i,k)} \mid \theta_t^{(i,k)}, \theta_t^{(i,k)}(\alpha) \right) g_i \left( \tau_t^{(i,k)} \mid \theta_t^{(i,k)} \right) \right) \right\|$$

$$= \mathbb{E} \left\| \frac{1}{K} \sum_k \left( \nabla J_i(\theta_t) \mp w^{(i)} \left( \tau_t^{(i,k)} \mid \theta_t, \theta_t^{(i,k)}(\alpha) \right) g_i \left( \tau_t^{(i,k)} \mid \theta_t \right) \right. \right.$$

$$\left. \left. - w^{(i)} \left( \tau_t^{(i,k)} \mid \theta_t^{(i,k)}, \theta_t^{(i,k)}(\alpha) \right) g_i \left( \tau_t^{(i,k)} \mid \theta_t^{(i,k)} \right) \right) \right\|$$

$$\leq \frac{\sigma}{K} + \frac{C_wC_g + L_g}{K} \sum_k \mathbb{E} \left\| \theta_t^{(i,k)} - \theta_t \right\| + C_wC_g\mathbb{E} \left\| \theta_t - \theta_{t-1} \right\|$$

$$\leq \frac{\sigma}{K} + \frac{1}{2}\eta K(C_wC_g + L_g) + \lambda C_wC_g. \tag{54}$$

Summing the above inequality over $t$, we have

$$\frac{1}{NKT} \sum_{t=1}^{T} \mathbb{E} \left[ \sum_{i,k} \left\| u_t^{(i,k)} - u_t \right\| \right]$$

$$\leq 2\beta \left( \left( 1 + \frac{1}{K} \right) \sigma + \eta K(3C_wC_g + 3L_g + \widetilde{L}) + (L_g + 6C_wC_g + 2\widetilde{L})\lambda \right). \tag{55}$$

### E.3 PROOF OF THEOREM 4.5

Set $\beta = \frac{\beta_0}{T^{\frac{2}{3}}}, \lambda = \frac{\lambda_0}{T^{\frac{2}{3}}}, \eta = \frac{1}{KT}$. From Lemma E.2, we can get that

$$\frac{1}{T} \sum_{t=0}^{T-1} \mathbb{E} \left\| \nabla J(\theta_t) - u_t \right\|$$

$$\leq \frac{1}{\beta_0 T^{1/3}} \left( \frac{C_wC_g + L_g + \widetilde{L}}{2T} + \frac{\sigma}{\sqrt{NK}} \right)$$

$$+ \sqrt{\frac{4\beta_0}{3NK}} \frac{L_g + 2C_wC_g}{T^{\frac{4}{3}}} + \frac{\sqrt{8\beta_0}\lambda_0(C_wC_g + \alpha)}{T} + \sqrt{\frac{4\beta_0}{NK}} \frac{\sigma}{T^{\frac{1}{3}}}$$

$$+ \frac{6C_wC_g\lambda_0}{T^{\frac{2}{3}}} + \frac{3C_wC_g + \sqrt{2}L_g}{T} + \frac{1}{T^{\frac{1}{3}}} \sqrt{\frac{2\sigma C_wC_g\lambda_0}{\sqrt{NK}}} + \frac{2}{T^{\frac{1}{2}}} \sqrt{\frac{(C_wC_g + L_g)\sigma}{2\sqrt{NK}}}$$

$$+ 2\widetilde{L} \left( \frac{1}{T} + \frac{\lambda_0}{T^{\frac{2}{3}}} \right) + 12 \left( \frac{1}{T} + \frac{\lambda_0}{T^{\frac{2}{3}}} \right)(L_g + C_wC_g) + \frac{1}{T^{\frac{1}{3}}} \sqrt{\frac{\sigma\lambda_0(L_g + C_wC_g)}{\sqrt{NK}}}$$

$$\lesssim \frac{\sigma}{\beta_0\sqrt{NK}T^{\frac{1}{3}}} + \frac{1}{T^{\frac{1}{3}}}\sqrt{\frac{2\sigma C_w C_g \lambda_0}{\sqrt{NK}}} + \frac{1}{T^{\frac{1}{3}}}\sqrt{\frac{\sigma\lambda_0(L_g + C_w C_g)}{\sqrt{NK}}} + \frac{\sigma}{T^{\frac{1}{3}}}\sqrt{\frac{4\beta_0}{NK}}. \tag{56}$$

Similarly, from Lemma E.3, we have

$$\frac{1}{NKT}\sum_{t=1}^{T}\mathbb{E}\left[\sum_{i,k}\left\|u_t^{(i,k)} - u_t\right\|\right]$$

$$\leq \frac{2\beta_0}{T^{\frac{2}{3}}}\left(\left(1 + \frac{1}{K}\right)\sigma + \frac{3C_w C_g + 3L_g + \widetilde{L}}{T} + \frac{\lambda_0(L_g + 6C_w C_g + 2\widetilde{L})}{T^{\frac{2}{3}}}\right). \tag{57}$$

Plugging Eq. (56) and Eq. (57) into Eq. (12), we have

$$\frac{1}{T}\sum_{t=0}^{T-1}\mathbb{E}\left\|\nabla J(\theta_t)\right\|$$

$$\lesssim \frac{\Delta}{\lambda_0 T^{\frac{1}{3}}} + \frac{2\sigma}{\beta_0\sqrt{NK}T^{\frac{1}{3}}} + \frac{2}{T^{\frac{1}{3}}}\sqrt{\frac{2\sigma C_w C_g \lambda_0}{\sqrt{NK}}} + \frac{2}{T^{\frac{1}{3}}}\sqrt{\frac{\sigma\lambda_0(L_g + C_w C_g)}{\sqrt{NK}}}$$

$$+ \frac{4\sigma\sqrt{\beta_0}}{\sqrt{NK}T^{\frac{1}{3}}} + \frac{\lambda_0 L}{T^{\frac{2}{3}}} + \frac{2\beta_0\sigma}{T^{\frac{2}{3}}}. \tag{58}$$

Set $\beta_0 = (NK)^{\frac{1}{3}}$ and $\lambda_0 = (NK)^{\frac{1}{3}}$, we have

$$\frac{1}{T}\sum_{t=0}^{T-1}\mathbb{E}\left\|\nabla J(\theta_t)\right\|$$

$$\lesssim \frac{\Delta}{\lambda_0 T^{\frac{1}{3}}} + \frac{2\sigma}{\beta_0\sqrt{NK}T^{\frac{1}{3}}} + \frac{2}{T^{\frac{1}{3}}}\sqrt{\frac{2\sigma C_w C_g \lambda_0}{\sqrt{NK}}} + \frac{2}{T^{\frac{1}{3}}}\sqrt{\frac{\sigma\lambda_0(L_g + C_w C_g)}{\sqrt{NK}}}$$

$$+ \frac{4\sigma\sqrt{\beta_0}}{\sqrt{NK}T^{\frac{1}{3}}} + \frac{\lambda_0 L}{T^{\frac{2}{3}}} + \frac{2\beta_0\sigma}{T^{\frac{2}{3}}}$$

$$\leq \mathcal{O}\left(\frac{\Delta + \sqrt{L_g + C_w C_g} + \sigma}{(NKT)^{\frac{1}{3}}} + \frac{(\sigma + L)(NK)^{\frac{1}{3}}}{T^{\frac{2}{3}}}\right)$$

$$\leq \mathcal{O}\left(\frac{\Delta + \bar{L}}{(NKT)^{\frac{1}{3}}} + \frac{\bar{L}(NK)^{\frac{1}{3}}}{T^{\frac{2}{3}}}\right), \tag{59}$$

where $\bar{L} = L + \sqrt{L_g + C_w C_g} + \sigma$. By setting $NK \leq \mathcal{O}(\sqrt{T})$, we have $\frac{(NK)^{\frac{1}{3}}}{T^{\frac{2}{3}}} \propto \mathcal{O}\left((NKT)^{-\frac{1}{3}}\right)$, and thus

$$\frac{1}{T}\sum_{t=0}^{T-1}\mathbb{E}\left\|\nabla J(\theta_t)\right\| \leq \mathcal{O}\left(\frac{\Delta + \bar{L}}{(NKT)^{\frac{1}{3}}}\right).$$

## F    EXPERIMENT CONFIGURATIONS AND ADDITIONAL RESULTS

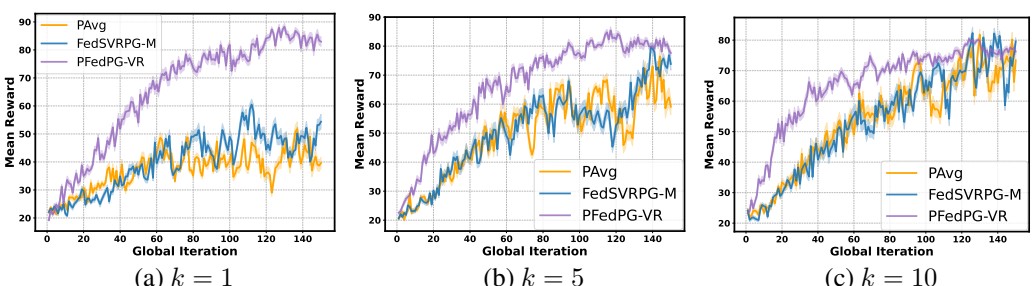

(a) $k = 1$                    (b) $k = 5$                    (c) $k = 10$

Figure 3: Mean rewards for VR-based methods on the CartPole task with varying numbers of local updates ($k = 1, 5, 10$).

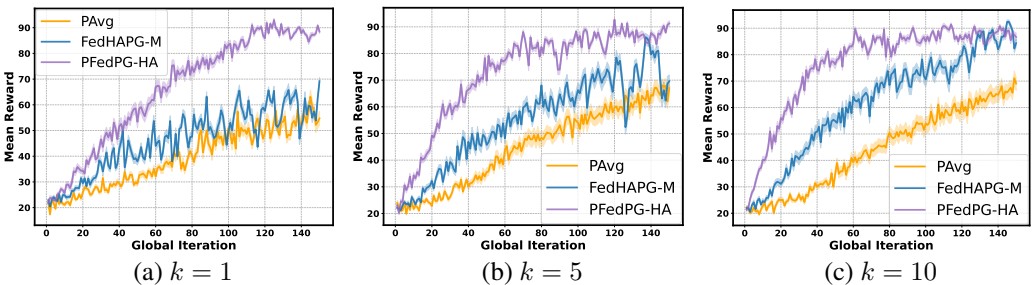

(a) $k = 1$                    (b) $k = 5$                    (c) $k = 10$

Figure 4: Mean rewards for HA-based methods on the CartPole task with varying numbers of local updates ($k = 1, 5, 10$).

Table 3: Mean rewards under different values of $\beta$ on the CartPole task.

| Algorithm | $\beta = 0.1$ | $\beta = 0.3$ | $\beta = 0.5$ | $\beta = 0.7$ | $\beta = 0.9$ |
|---|---|---|---|---|---|
| PFEDPG-HA | $92.01_{\pm 2.93}$ | $91.86_{\pm 2.94}$ | $91.93_{\pm 3.05}$ | $91.77_{\pm 3.03}$ | $91.11_{\pm 3.03}$ |
| PFEDPG-VR | $83.23_{\pm 3.86}$ | $80.67_{\pm 4.34}$ | $78.75_{\pm 4.43}$ | $78.60_{\pm 4.24}$ | $79.37_{\pm 4.38}$ |
| FEDHAPG-M | $41.58_{\pm 4.10}$ | $50.92_{\pm 4.14}$ | $58.73_{\pm 5.35}$ | $84.81_{\pm 5.14}$ | $89.96_{\pm 3.78}$ |
| FEDSVRPG-M | $81.55_{\pm 4.14}$ | $79.97_{\pm 4.20}$ | $75.77_{\pm 5.35}$ | $78.38_{\pm 5.01}$ | $75.69_{\pm 4.92}$ |

The experiments were conducted on a host machine equipped with an Intel(R) Xeon(R) W9-3475X CPU running at 2.20 GHz (maximum turbo frequency: 4.80 GHz), featuring 36 physical cores and 72 threads. The system was configured with 256 GB of DDR5 ECC RAM and a single NVIDIA(R) RTX(TM) A6000 GPU with 48 GB of memory.

### F.1    DETAILS OF TABULAR CASE

We consider random MDPs consisting of $N = 10$ environments. Each MDP features 5 discrete states and 5 actions. The number of global iterations is set to $T_{\max} = 1$, and the discount factor is $\gamma = 0.9$. The state transition kernel is generated randomly, with each element sampled from a Bernoulli distribution. The number of local updates is set to $K = 32$. Different levels of environment heterogeneity are evaluated using 1,600 random seeds. The local stepsize is set to $\eta = 0.05$ for all baseline methods. For our proposed PFEDPG-VR algorithm, the stepsize strictly follows the theoretical specification outlined in Theorem 4.4.

### F.2    DETAILS OF DRL CASE

CartPole is a widely used benchmark in reinforcement learning that involves balancing a pole on a moving cart—commonly known as the inverted pendulum problem. The cart moves along a frictionless track, and the agent must apply forces to the left or right to prevent the pole from falling or the cart from exceeding predefined boundaries. Each episode begins with the pole slightly tilted, and

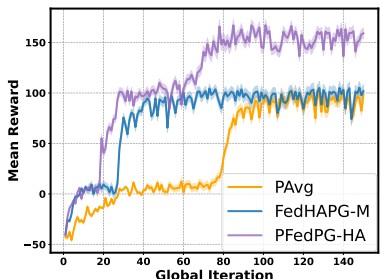

Figure 5: Mean rewards versus global iterations on the HalfCheetah task.

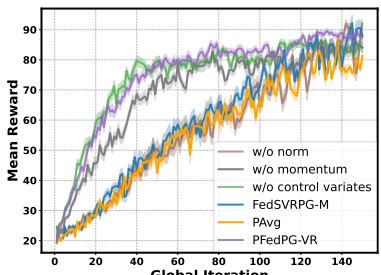

Figure 6: Mean rewards versus global iterations for ablation studies on the CartPole task.

the agent receives a reward at each time step that the pole remains upright. The episode terminates if the pole tilts beyond a critical angle or if the cart moves outside the allowed range.

HalfCheetah is another popular benchmark, focusing on continuous control tasks. The agent is a simplified robotic model inspired by a cheetah, emphasizing the lower body. Its structure consists of multiple connected joints and segments that must be coordinated to achieve forward motion. The goal is to maximize forward velocity while minimizing energy use and avoiding unstable movements. At each step, the agent receives a reward that encourages faster locomotion and penalizes inefficient or erratic actions. The main challenge lies in learning control strategies that balance speed, stability, and smooth motion.

For the DRL experiments, we use the configuration with $N = 5$ agents and $K = 10$ local updates. All algorithms are evaluated over 150 rounds to ensure a fair comparison. For the baseline methods, we set the local stepsize to 0.005 and the global stepsize to 0.6, which correspond to the best-performing choices identified for this general setting through a comprehensive grid search. Whenever $N$, $K$, or problem-dependent constants change, we repeat this procedure and report the best-performing baseline under the new configuration. The same protocol is applied to all ablation variants: each variant undergoes its own sweep, and we report its best-performing curve. For our proposed algorithms, the stepsize is chosen to strictly follow the theoretical prescriptions provided in Theorem 4.4 for PFEDPG-VR and Theorem 4.5 for PFEDPG-HA. All experiments are performed over 10 independent runs with different random seeds, and all reported results use mean $\pm$ standard error throughout.

### F.3 Experiment on Different Numbers of Local Updates

As shown in Figure 3 and Figure 4, we evaluate the impact of varying the number of local updates on the performance of all methods. The results indicate that, in both VR-based and HA-based settings, increasing the number of local updates generally leads to improved convergence behavior across all methods. Crucially, our proposed algorithms consistently achieve the best performance under all local update configurations. This demonstrates both the strong convergence behavior of our methods and the effectiveness of the problem-parameter-free design. Unlike baseline methods, which often require careful stepsize tuning to perform well, our methods reliably reach optimal convergence rates without any hyperparameter adjustment. These findings further underscore the robustness and practical applicability of our approach in FRL scenarios.

### F.4 Experiment on Varying Values of Momentum Coefficient $\beta$

To further demonstrate the robustness of our approach, we perform a sensitivity analysis of the momentum coefficient $\beta$ on the CartPole task. As shown in Table 3, both PFEDPG-VR and PFEDPG-HA maintain strong performance across the full range of $\beta$ values tested, while the baseline methods exhibit noticeable fluctuations and occasional drops when $\beta$ is not well tuned. These results highlight that our problem-parameter-free FRL framework not only achieves superior performance but also ensures stable and reliable learning under varying momentum settings, reinforcing its effectiveness and practical utility in RL tasks.

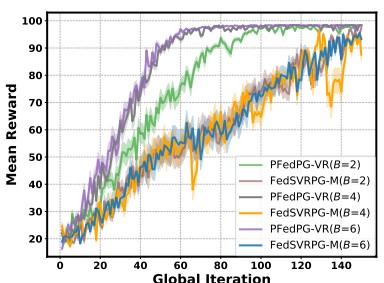

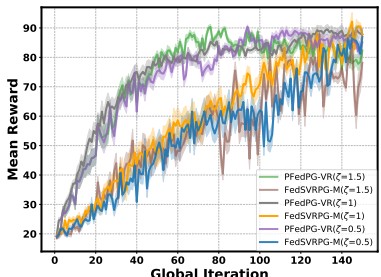

Figure 7: Mean rewards on the Cart-Pole task with different numbers of trajectories sampled per local update.

Figure 8: Mean rewards on the Cart-Pole task with different values of $R_{\max}$.

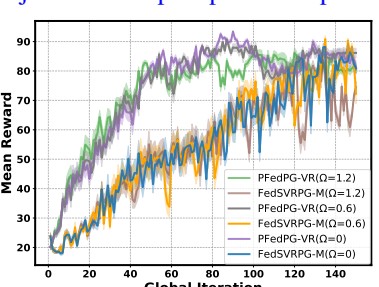

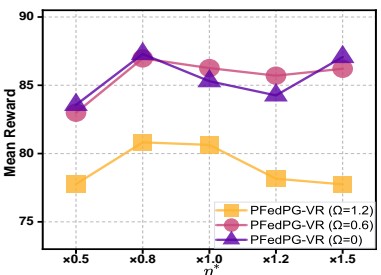

Figure 9: Mean rewards on the Cart-Pole task across varying degrees of reward heterogeneity.

Figure 10: Mean rewards of our method versus learning rates under different reward heterogeneity levels.

## F.5 EXPERIMENT FOR ABLATION STUDIES

To validate the effectiveness of our problem-parameter-free design, we conduct ablation studies to evaluate the contribution of each key component: momentum, gradient normalization, and control variates. The results are summarized in Figure 6.

In Figure 6, PAVG represents the standard policy gradient method without any enhancements. FEDSVRPG-M integrates momentum acceleration into the update rule, and PFEDPG-VR is our complete method that incorporates momentum, gradient normalization, and control variates. The other three show the results of removing one component at a time from PFEDPG-VR: specifically, removing momentum (w/o momentum), gradient normalization (w/o norm), and control variates (w/o control variates).

The results clearly show that PFEDPG-VR achieves the highest performance, outperforming all baselines and ablated variants. This demonstrates that momentum, gradient normalization, and control variates are indispensable for achieving optimal performance in our problem-parameter-free framework. Importantly, all baseline methods and ablated variants required extensive hyperparameter tuning to reach their best reported performance, while PFEDPG-VR selects the optimal learning rate directly from known system parameters, making it a practically deployable and problem-parameter-free solution.

## F.6 EXPERIMENTS UNDER VARYING PROBLEM-DEPENDENT PARAMETERS

To better understand how the problem-dependent parameters in our bounds are reflected in practice, we conduct a set of experiments on the CartPole task.

**Influence of gradient variance $\sigma$.** Gradient variance $\sigma$ directly affects the stochasticity of the updates and appears explicitly in our convergence bounds. To examine its impact, we vary the number of sampled trajectories $B$ per local update, using $B \in \{2, 4, 6\}$. A smaller $B$ leads to noisier gradient estimates, whereas a larger $B$ produces gradients with lower variance. For each value of $B$, we keep all other hyperparameters fixed and train both PFEDPG-VR and FEDSVRPG-M. As shown in Figure 7, PFEDPG-VR remains stable and consistently performs better than the baseline across

all choices of $B$. This behavior is in line with our algorithmic design: variance reduction, control variates, and momentum together reduce the effect of gradient noise, while the normalized updates keep each stepsize controlled, preventing occasional large gradients from causing unstable jumps in the parameters.

**Influence of $R_{\max}$-dependent constants.** We further examine the effect of the constants that depend on the reward bound $R_{\max}$, namely $L = HR_{\max}(M + HG^2)/(1 - \gamma)$, $L_g = HMR_{\max}/(1 - \gamma)$, and $C_g = HGR_{\max}/(1 - \gamma)$. During training, we multiply the environment reward by a scalar factor $\zeta$, with $\zeta \in \{0.5, 1.0, 1.5\}$. This scaling keeps the optimal policy unchanged but makes $R_{\max}$, and therefore $L$, $L_g$, and $C_g$, larger or smaller in proportion to $\zeta$. We train all methods under these scaled rewards and evaluate their performance in the original environment. The resulting learning curves are shown in Figure 8. PFEDPG-VR demonstrates stable and strong performance for all values of $\zeta$. This matches the theory: the normalized updates keep the effective step length controlled, so increasing the reward scale mainly changes the constants in the bound, not the stability region of the algorithm, and the problem-parameter-free stepsizes continue to work well without retuning.

**Influence of reward heterogeneity.** Finally, we consider a reward–heterogeneous setting in which all agents share the same CartPole dynamics and initial-state distribution, but each agent uses a different shaped reward. Specifically, we wrap each local environment with a potential-based shaping term $r_t^{(i)} = r_t + \gamma \Phi_i(s_{t+1}) - \Phi_i(s_t)$, where $\Phi_i$ is a state-dependent potential. The coefficients of $\Phi_i$ are scaled by a heterogeneity-strength parameter $\Omega$: for agent $i$, the shaping weights are multiplied by $(1 + \Omega d_i)$ with fixed offsets $d_i \in [-1, 1]$. When $\Omega$ is close to zero, all agents have almost identical reward functions; as $\Omega$ grows, the reward functions become increasingly different across agents. By Lemma C.2, the variance of the IS weights is upper-bounded by a term proportional to $\|\theta_1 - \theta_2\|^2$ scaled by the constant $C_w$. Thus, greater reward heterogeneity leads to larger parameter discrepancies between agents, placing the algorithm in regimes with higher IS-weight variance and making training more challenging. We run PFEDPG-VR and FEDSVRPG-M for $\Omega \in \{0, 0.6, 1.2\}$ and plot the mean reward in Figure 9. Our method remains stable and competitive as reward heterogeneity increases. This is consistent with the algorithmic design: control variates mitigate the drift caused by heterogeneous rewards, importance sampling corrects for policy mismatch, and the combination of momentum and normalization keeps the global update direction smooth and well scaled.

In the same reward-heterogeneous setting, we also test the sensitivity of our problem-parameter-free stepsizes. For each $\Omega \in \{0, 0.6, 1.2\}$, we multiply the theoretically prescribed local stepsize $\eta^*$ of PFEDPG-VR by factors in $\{0.5, 0.8, 1.0, 1.2, 1.5\}$ and record the final mean reward. Figure 10 shows that performance varies only mildly across these multipliers and heterogeneity levels. This suggests that the problem-parameter-free stepsizes given by our theory are strong default choices in practice, and that small scalar adjustments can further refine performance at very low tuning cost.

# G  THE USE OF LARGE LANGUAGE MODELS (LLMS)

We used LLMs solely to polish the writing and grammar of this paper.

