# OpenReview forum: "Towards Federated Reinforcement Learning Free of Problem-Parameter-Based Tuning"
_ICLR.cc/2026/Conference — Submitted to ICLR 2026_

### Official Review · Reviewer_wY6d · 2025-10-28

**Soundness:** 3
**Presentation:** 2
**Contribution:** 3
**Rating:** 4
**Confidence:** 4

**Summary:**

This paper investigates Federated Reinforcement Learning (FRL) under environment heterogeneity. The authors propose a problem-parameter-free FRL algorithm that does not require hyperparameter tuning (learning rates/stepsizes) or rely on unknown problem-specific parameters (e.g., smoothness constants, heterogeneity bounds, gradient variances) via a local adaptive stepsize. They develop federated policy gradient algorithms (PFEDPG-VR, PFEDPG-HA) with momentum and prove that these achieve state-of-the-art convergence rates.

**Strengths:**

1. The paper addresses a genuine practical problem - the need for manual hyperparameter tuning in federated reinforcement learning (FRL) that depends on unknown problem-specific parameters.

1. They demonstrate that the proposed algorithm can achieve state-of-the-art convergence rates.

1. They demonstrate the superior performance of the proposed method compared with baselines FEDSVRPG-M (Wang et al. 2024a) and PAVG.

**Weaknesses:**

1. Limited Novelty Beyond Stepsize: The core algorithmic structure, employing momentum-based updates combined with variance reduction (VR) or Hessian-aided (HA) techniques for heterogeneous FRL, closely follows the approach proposed by Wang et al. (2024a). The primary novel component appears to be the integration of the adaptive stepsize mechanism to achieve the "parameter-free" property for tuning. The paper could benefit from a more detailed discussion clarifying the specific algorithmic design differences compared to Wang et al. (2024a) beyond the stepsize rule itself and elaborating on how the normalization specifically circumvents the reliance on problem constants in the stepsize formulas and what the technical contribution is in deriving the same convergence rate with the adaptive stepsize.

2. Unclear dependency of problem-specific parameters in the convergence rate bound: It seems that the convergence bound is dependent on many problem-specific parameters (G, M, W, H, $\gamma$, R), which are roughly written as $\hat{L}$ and $\overline{L}$ in Theorem 4.4 and 4.5. However, given that the consequence of using a learning rate not considering problem-specific parameters may appear in the convergence rate, characterizing the dependency of the parameters on the convergence rate seems very critical. It would be better to present the exact effect of the problem-specific parameters on the convergence more clearly in the theorems.

3. Lack of comparison on the dependency of problem-specific parameters: The paper does not analyze whether the dependence of its convergence rate on the aggregated constants ($\hat{L}$, $\overline{L}$) is better, worse, or the same compared to the dependence shown in Wang et al. (2024a) (which uses problem-dependent stepsizes). Since the main contribution is being "parameter-free," it is crucial to understand if this advantage in tuning comes at the cost of potentially worse dependencies on the problem-specific parameters in the convergence rate bound. This comparative analysis regarding the rate's parameter dependency is missing.


[1] Han Wang, Sihong He, Zhili Zhang, Fei Miao, and James Anderson. Momentum for the win: Collaborative federated reinforcement learning across heterogeneous environments. arXiv preprint arXiv:2405.19499, 2024a.

**Questions:**

See the weaknesses.

---

> ### Author Response · Authors · 2025-11-20
> **Point-to-Point Responses**
>
> **We sincerely appreciate the reviewer for recognizing our contributions and for the constructive comments. Our point-to-point responses to concerns on Weaknesses and Questions are given below.**
>
> **Reply to Weakness 1:**
>
> Thank you for your valuable comment. Our work explores variance-reduced [1] and Hessian-aided [2] FRL methods and, like [3], incorporates momentum—a technique that has become standard for accelerating convergence in federated settings [4]–[5]. However, our key contribution lies in the development of the **first problem-parameter-free FRL framework**, which fundamentally distinguishes our approach.
>
> Specifically, our algorithmic design differs from [3] in two key ways:
>
> ### 1. Gradient normalization and problem-parameter-free stepsizes
>
> We introduce per-step gradient normalization at every local update, $\theta_t^{(i,k+1)}=\theta_t^{(i,k)}+\eta\frac{u_t^{(i,k)}}{\Vert u_t^{(i,k)}\Vert}$, which guarantees that $\Vert\theta_t^{(i,k+1)}-\theta_t^{(i,k)}\Vert=\eta$, $\forall i, k, t$. This normalization effectively acts as an adaptive learning rate: it takes larger steps when gradients are small (where more aggressive exploration is beneficial) and smaller steps when gradients are large (where careful progress is needed).
>
> Normalization is also the technical mechanism that enables problem-parameter-free learning rates:
>
> **(i).** Since every local step has norm $\eta$, the total drift of a local model within one communication round is deterministically bounded by $K\eta$. Thus, the per-round model movement can be controlled directly through $\eta$ and $K$, without inserting any unknown problem parameters into the stepsize rules.
>
> **(ii).** In the convergence recursion, all problem-dependent constants appear only as multiplicative factors attached to $\eta$; they no longer determine the scaling of $\eta$ itself. This lets us choose $\eta$ purely as a function of the system parameters $(N,K,T)$ to balance the ascent and variance terms.
>
> This is precisely how gradient normalization eliminates the “oracle” stepsizes used in previous FRL frameworks—stepsizes that normally depend on unknown problem parameters—while still yielding the same optimal convergence rates. In short, gradient normalization effectively decouples the update magnitude from unknown problem parameters, which is the key mechanism that enables a problem-parameter-free stepsize.
>
> ### 2. Control variates
>
> Control variates act as a drift-correction mechanism that aligns local updates with the global objective and reduces gradient variance induced by heterogeneous environments. Specifically, they reduce the difference between the average local gradients and the true global gradient, so the update direction still accurately reflects $\nabla J(\theta_t)$ even when agents have different rewards, transition kernels, or initial-state distributions. At a technical level, the control-variate terms make the heterogeneity-induced error vanish asymptotically, which is what allows our analysis to establish convergence without imposing explicit bounds on environment discrepancies across agents.
>
> ### Synergy of momentum, gradient normalization, and control variates
>
> Building on the above design, momentum smooths noise by accumulating gradients across agents and rounds, accelerates progress in directions of consistent agreement, and stabilizes the normalized updates. Together with gradient normalization and control variates, this yields a synergistic triad:
> - Local gradient normalization for adaptive stepsize control,
> - Client-side momentum for accelerated convergence,
> - Control variates for mitigating agent-drift under heterogeneity.
>
> Removing any of these components either reintroduces the need for problem-parameter-based tuning or forces one to assume explicit heterogeneity bounds.
>
> Moreover, **our theoretical analysis takes a fundamentally different path from prior FRL works** such as [3] and [6]. Specifically:
> - Handling normalized gradients: Gradient normalization introduces mathematical complexities in tracking update dynamics, requiring new techniques to analyze its interaction with momentum.
> - Momentum analysis: Momentum creates coupling across updates; we develop new bounds to capture its interaction with control variates.
> - Problem-parameter-free guarantees: We prove convergence without relying on problem-dependent constants, using novel tools to show that normalization compensates for the absence of tuned stepsizes.
>
> These challenges necessitated new proof techniques not present in prior FRL analyses. Thus, our problem-parameter-free framework represents a fundamental advancement for the practical deployment of FRL algorithms. In the revised paper, we have (i) added a more detailed explanation in Subsection 3.1 on how normalization removes dependence on problem constants in stepsize formulas, and (ii) clarified in Section 5 how our algorithmic design differs from [3] while more explicitly highlighting our contributions.

---

> > ### Author Response · Authors · 2025-11-20
> > **Point-to-Point Responses**
> >
> > **Reply to Weakness 2:**
> >
> > Thank you for your thoughtful comment. The dependence on all problem-specific parameters—consolidated from our theoretical analysis—is already made explicit in Theorems 4.4 and 4.5 in our paper. For a more detailed explanation of how they influence the convergence bound, we respectfully refer you to our response to Weakness 3.
> >
> > **Reply to Weakness 3:**
> >
> > Thank you for your insightful comment. We now address the dependence on problem-specific parameters both theoretically and empirically.
> >
> > ### (i). Theoretical comparison with [3]
> >
> > We explicitly characterize the dependence on problem constants in the revised paper. In Theorem 4.4, our convergence bound has *linear* dependence on $(L,L_g,C_w,C_g,\sigma)$; in Theorem 4.5, it is *linear* in $(L,\sigma)$ and *square-root* in $(L_g,C_w,C_g)$. In comparison, [3] attains *cube-root* dependence on most constants—while incurring a *$2/3$-order* dependence on $C_w$—via problem-dependent stepsize balancing. Our guarantees, however, are problem-parameter-free and do not assume explicit environment heterogeneity bounds.
> >
> > The linear/square-root dependence on certain problem constants in our bounds arises because gradient normalization fixes the effective step length without invoking unknown problem parameters, while momentum aggregates updates without the oracle-style balancing that compresses constant factors. This trade-off—accepting a slightly higher-order dependence on some problem constants in order to obtain problem-parameter-free guarantees—is standard in problem-parameter-free analyses (e.g., [7]-[9]). Importantly, our bounds share the same best-known leading order in $T$ (with linear speedups in $N$ and $K$) as [3], [10], so for moderately large $T$ the asymptotic rate dominates and the practical influence of these constant factors diminishes quickly. Given the practical gains from eliminating problem-specific tuning, this modest increase in constant order is widely regarded as acceptable and well-justified in the current literature.
> >
> > We summarize this comparison explicitly in Remark 4.8 of the revised paper.

---

> > > ### Author Response · Authors · 2025-11-20
> > > **Point-to-Point Responses**
> > >
> > > ### (ii). Empirical sensitivity to problem constants
> > >
> > > To complement the theory, Appendix F.6 in the revised paper now includes experiments that illustrate how these problem constants affect performance in practice:
> > >
> > > - **Gradient variance ($\sigma$).** In Figure 7, we vary the number of sampled trajectories per update, which affects the gradient variance. Across this range, our method behaves stably and remains competitive relative to baselines whose learning rates are tuned by grid search.
> > >
> > > - **IS-weight variance.** In Figure 9, we vary the reward-heterogeneity level, which increases policy discrepancy across agents. According to Lemma C.2 in Appendix C of the paper, the variance of the IS weights is bounded by a term that grows with the squared distance between policy parameters and is scaled by the constant $C_w$. Therefore, higher reward heterogeneity drives the algorithm into regimes where the IS-weight variance becomes larger, making the learning problem more challenging. The results demonstrate that our method remains stable and maintains competitive performance even under these high-variance conditions.
> > >
> > > - **Other problem-dependent constants ($L, L_g, C_g$).** In Figure 8, we perform a reward–scaling experiment: we multiply the reward by a factor $\zeta > 0$ during training, which preserves the optimal policy but scales the reward bound $R_{\max}$. This in turn could scale the constants $L = H R_{\max} {(M + H G^2)}/{(1 - \gamma)}$, $L_g = H M R_{\max} / (1 - \gamma)$, and $C_g = H G R_{\max} / (1 - \gamma)$. Thus, increasing $\zeta$ could proportionally increase $L$, $L_g$, and $C_g$. We train all methods under these scaled rewards and evaluate their performance in the original environment, allowing us to directly test robustness to changes in these reward-dependent constants. Across this range, our method again demonstrates stable and superior performance, indicating robustness with respect to variations in these problem-dependent constants.
> > >
> > > Taken together, these theoretical and empirical results show that while our problem-parameter-free design leads to slightly higher dependence on some problem constants, our convergence rates still match the best-known results in [3], [10]. This level of constant dependence is inherent to problem-parameter-free methods and is commonly observed in related works (e.g., [7]–[9]). Moreover, for moderately large $T$, the asymptotic rate dominates, making the impact of these constant factors negligible in practice. Empirically, our methods remain robust under variation in the underlying problem parameters.
> > >
> > > **References:**
> > >
> > > [1] PAGE-PG: A simple and loopless variance-reduced policy gradient method with probabilistic gradient estimation.
> > >
> > > [2] Hessian aided policy gradient.
> > >
> > > [3] Momentum for the win: Collaborative federated reinforcement learning across heterogeneous environments.
> > >
> > > [4] Momentum benefits non-iid federated learning simply and provably.
> > >
> > > [5] FedCM: Federated learning with client-level momentum.
> > >
> > > [6] Federated reinforcement learning with environment heterogeneity.
> > >
> > > [7] Two sides of one coin: the limits of untuned SGD and the power of adaptive methods.
> > >
> > > [8] Problem-parameter-free decentralized nonconvex stochastic optimization.
> > >
> > > [9] Tuning-free stochastic optimization.
> > >
> > > [10] Stochastic policy gradient methods: Improved sample complexity for fisher-non-degenerate policies.
> > >
> > > **Thank you once again for your thoughtful review and constructive feedback.**

---

### Official Review · Reviewer_hFbc · 2025-10-30

**Soundness:** 3
**Presentation:** 3
**Contribution:** 3
**Rating:** 8
**Confidence:** 2

**Summary:**

The paper introduces a federated reinforcement learning (FRL) framework that removes the need for manual step-size tuning, enabling privacy-preserving intelligent decision-making in distributed settings. The framework unifies adaptive policy and momentum updates with carefully designed control variables at both the client and server, thereby obviating manual step-size selection. The authors further propose two algorithms: TFEDPG-VR, which employs a variance-reduced policy-gradient update, and TFEDPG-HA, which enhances performance via a Hessian-aided adjustment. Theoretical analysis shows that both algorithms achieve optimal sample complexity O(ε^-3) and communication complexity O(ε^-2), with linear speedups as the number of participating agents and local update steps increase. Experiments corroborate superior performance and robustness, demonstrating that the methods effectively avoid hyperparameter tuning even under numerous unknowns, thus simplifying practical deployment and improving adaptability and generalization.

**Strengths:**

Although I am not major in the federate learning but only for reinforcement learning. From the perspective of a researcher in the field of reinforcement learning, this paper excels in terms of methodological innovation, theoretical integrity, and experimental completeness, I think it is a solid paper.

**Weaknesses:**

The empirical evaluation lacks statistical rigor. Learning curves appear to be based on a single run without confidence intervals or standard deviations. Reporting results over multiple independent seeds with appropriate statistical measures (e.g., mean ± standard deviation or 95% confidence intervals) is necessary to substantiate the claimed performance and improve the credibility of the results.

**Questions:**

You can observe the weakness.

---

> ### Author Response · Authors · 2025-11-20
> **Point-to-Point Responses**
>
> **We sincerely appreciate the reviewer for recognizing our contributions and for the constructive comments. Our point-to-point responses to concerns on Weaknesses and Questions are given below.**
>
> **Reply to Weaknesses and Questions:**
>
> Thank you for your valuable comment. We would like to clarify that our empirical results are not based on a single run. All experiments in the paper on deep RL tasks were conducted over 10 independent runs with different random seeds, and we have reported the results in the paper using **mean ± standard error** throughout. We have updated the experimental section to state this statistical reporting more clearly.
>
> **Thank you once again for your thoughtful review and constructive feedback.**

---

### Official Review · Reviewer_78RH · 2025-10-30

**Soundness:** 3
**Presentation:** 2
**Contribution:** 3
**Rating:** 6
**Confidence:** 4

**Summary:**

This paper presents a novel problem-parameter-free framework for Federated Reinforcement Learning (FRL) to address the critical challenge of manual hyperparameter tuning in heterogeneous environments. The authors propose two algorithms, PFedPG-VR and PFedPG-HA, which integrate adaptive step sizes, momentum-based updates, and control variates, achieving state-of-the-art convergence guarantees.

**Strengths:**

**Novel and Impactful Framework:** The proposed problem-parameter-free FRL framework is both innovative and practically significant. It eliminates the need for manual hyperparameter tuning, thereby saving substantial time and improving the stability of the learning algorithms.

**Free of Arbitrary Heterogeneity:** The use of momentum-based update removes the need for restrictive bounded-heterogeneity assumptions. Consequently, the convergence guarantees are independent of any additive error terms related to the degree of heterogeneity.

**State-of-the-Art Theoretical Performance:** The proposed algorithms are proved to attain the optimal sample complexity of $O(\varepsilon^{-3})$ and communication complexity of $O(\varepsilon^{-2})$, matching the state of the art for policy-based federated reinforcement learning.

**Weaknesses:**

**Weakness 1: Unclear Contributions of Algorithmic Design.** The algorithmic design contributions are not clearly justified. Both the control variate and normalization components lack sufficient theoretical or intuitive motivation in the main text. While the paper claims that these techniques help remove heterogeneity-dependent parameters, their specific roles in achieving this effect remain unclear. Moreover, recent work [1] achieves convergence under arbitrary heterogeneity without incorporating either control variates or normalization, suggesting that these components may not be essential to the claimed theoretical improvements. This issue is particularly important given that, according to the ablation study, removing the control variate component leads to only marginal differences in practical performance compared to the full algorithm. A more detailed explanation is needed to clarify the concrete contributions of these design choices.

**Weakness 2: Lack of Reward Heterogeneity Experiments.** While the proposed algorithms are theoretically designed to handle arbitrary environment heterogeneity, the experimental validation primarily focuses on heterogeneity in transition dynamics and initial state distributions. No explicit experiments are provided to assess performance under reward heterogeneity, where agents have different reward functions.

[1] Wang, Han, et al. "Momentum for the Win: Collaborative Federated Reinforcement Learning across Heterogeneous Environments." International Conference on Machine Learning. PMLR, 2024.

**Questions:**

1. How do control variates specifically contribute to the “problem-parameter-free” framework? From the ablation study, removing control variates seems to result in very similar performance. Theoretically, prior work [1] achieves convergence under arbitrary heterogeneity without using control variates.

2. The paper suggests that normalization is a key design for achieving a parameter-free learning rate. Could the authors provide a more technical or intuitive explanation to support this claim?

3. Could the authors include experiments on reward heterogeneity, as it is another common form of environmental heterogeneity?

4. Could the authors compare the convergence results presented in Theorems 4.5 and 4.6 with those in [1]?

[1] Wang, Han, et al. "Momentum for the Win: Collaborative Federated Reinforcement Learning across Heterogeneous Environments." International Conference on Machine Learning. PMLR, 2024.

---

> ### Author Response · Authors · 2025-11-20
> **Point-to-Point Responses**
>
> **We sincerely appreciate the reviewer for recognizing our contributions and for the constructive comments. Our point-to-point responses to concerns on Weaknesses and Questions are given below.**
>
> **Reply to Weakness 1 and Questions 1&2:**
>
> Thank you for your valuable comment. Below we clarify the concrete contributions of each component.
>
> ### 1. Clarifying the role and novelty of control variates
>
> - **Specific role.** Control variates act as a drift-correction mechanism, aligning local updates with the global objective and reducing gradient variance caused by heterogeneous environments. This leads to more stable and consistent updates across agents. From a theoretical perspective, the control-variate term ensures that the heterogeneity-induced error—i.e., the gap between the aggregated local gradient and the true global gradient—contracts asymptotically, allowing us to establish convergence without requiring explicit bounds on environment discrepancies (e.g., differences in reward functions, state transition kernels, or initial state distributions).
>
> - **Why control variates remain essential in our setting.** While [1] achieves convergence under arbitrary heterogeneity without control variates, it still needs problem-dependent stepsize tuning based on unknown problem parameters. In our problem-parameter-free setting, control variates are needed to keep the update direction sufficiently accurate so that a system-parameter–based stepsize remains valid.
>
> - **Regarding the ablation.** The ablated variant (without control variates) appears competitive only because we re-tune its learning rate via grid search. This variant loses problem-parameter-free guarantees and cannot use our theoretically prescribed stepsizes. Thus the empirical gap can be small, but the theoretical necessity remains fundamental.
>
> We have clarified the role of control variates more explicitly in the Subsection 3.1 of the revised paper.
>
> ### 2. Why normalization is key to achieving a problem-parameter-free learning rate
>
> Gradient normalization ensures each update has controlled length: $\Vert\theta_t^{(i,k+1)}-\theta_t^{(i,k)} \Vert=\eta \frac{u_t^{(i,k)}}{\Vert u_t^{(i,k)}\Vert}=\eta$, $\forall i, k, t$, which effectively adjusts the learning rate according to the local optimization landscape. Specifically, this design automatically allows larger steps in regions with small gradients (where more aggressive exploration is beneficial) and smaller steps in steep regions (where careful progress is needed). Beyond this intuition, normalization enables two key theoretical properties:
>   1. Since every local step has norm $\eta$, the total drift of a local model within one communication round is deterministically bounded by $K\eta$. Consequently, we can control model update in one communication round directly through $\eta$ and $K$, without needing any unknown problem parameters.
>   2. In the convergence recursion, all problem-dependent constants multiply at most $\eta$ or $K\eta$; they do not determine the scaling of $\eta$ itself. This structure allows us to set $\eta$ purely as a function of the system parameters $(N,K,T)$ to balance the ascent and variance terms. The problem-specific constants then only affect the multiplicative constants in the final bound, so we retain the optimal convergence rate while the stepsizes remain problem-parameter-free.
>
> Thus, normalization decouples step length from unknown problem parameters—this is the core reason the algorithm becomes problem-parameter-free. We have provided a more detailed technical and intuitive explanation of the role of normalization in Subsection 3.1 of the revised paper.
>
> ### 3. Why momentum and control variates remain indispensable
>
> Although [1] uses momentum alone—an approach that is now standard for mitigating heterogeneity in federated optimization [2]–[3]—and our ablations can perform reasonably once manually tuned, both momentum and control variates are essential within our problem-parameter-free framework.
>
> - **Momentum**
>   - smooths noise by accumulating gradients across agents and rounds,
>   - accelerates optimization progress in consistent directions, and
>   - stabilizes normalized updates, which can otherwise be sensitive to noise.
>
> - **Control variates**
>   - correct agent drift due to heterogeneous environments, and
>   - ensure the update direction remains a reliable surrogate for $\nabla J(\theta_t)$.
>
> Together with normalization, they form a synergistic triad:
>
> 1. **Normalization** eliminates dependence on unknown problem-specific parameters.
> 2. **Momentum** stabilizes and accelerates normalized updates.
> 3. **Control variates** reduce heterogeneity-induced drift so normalization can safely use system-parameter–based stepsizes.
>
> Removing any component reintroduces the need for tuning or requires environment heterogeneity bounds.

---

> > ### Author Response · Authors · 2025-11-20
> > **Point-to-Point Responses**
> >
> > ### 4. Summary: main contribution of our method
> >
> > Our main contribution is to develop the **first FRL algorithms whose hyperparameters depend solely on the known system parameters** $(N,K,T)$, enabling an effectively problem-parameter-free methodology that removes environment heterogeneity bounds. Even without the problem-parameter-based tuning, our methods still **match the best-known convergence rates in $\epsilon$**. Achieving this is technically nontrivial: the interaction between gradient normalization, momentum, and control variates introduces complex analytical coupling under non-stationary, heterogeneous environments, and controlling this interplay requires **fundamentally new proof techniques beyond standard RL or federated optimization frameworks**. We have revised the main text to provide a clearer justification of the necessity and contributions of our work in Section 5 of the revised paper.
> >
> > **Reply to Weakness 2 and Question 3:**
> >
> > Thank you for your thoughtful suggestion. In the revised paper, we have added a new experiment that explicitly evaluates our methods under reward heterogeneity. Specifically, we consider a federated CartPole setting where all agents share the same dynamics and initial-state distribution, but each agent is assigned a different reward function via state-dependent reward shaping. The degree of reward heterogeneity is controlled by a scalar heterogeneity strength parameter $\Omega$: when $\Omega$ is close to 0, all agents use almost identical shaping weights and thus face nearly the same reward function; as we increase the $\Omega$, the shaping weights for different agents are pushed further apart (e.g., some agents are rewarded more for keeping the pole upright, others are penalized more for position or velocity deviations), which creates genuinely different reward landscapes. In addition, we perform a learning-rate sensitivity analysis under different reward-heterogeneity settings by scaling our theoretically prescribed learning rates with several nearby multipliers to test the robustness of our method.
> >
> > The results reported in Figure 9 of Appendix F.6 show that our proposed method remains stable across a range of reward-heterogeneity levels and maintains clear performance advantages over the baselines, even as the reward heterogeneity increases. For the learning-rate sensitivity under reward heterogeneity, the results in Figure 10 of Appendix F.6 show that performance changes only mildly, indicating that our problem-parameter-free stepsizes are robust and provide reasonable default choices even in the presence of reward heterogeneity.
> >
> > **Reply to Question 4:**
> >
> > Thank you for your helpful question. We have now provided a direct comparison between our convergence guarantees and those in [1]. As summarized in Table 1 and further discussed in Remark 4.6, the convergence rates established in Theorems 4.5 and 4.6 match those obtained in [1]. Specifically, both our methods and [1] achieve the same leading-order rate in $\epsilon$ and the same linear speedup in $N$ and $K$. The key distinction is that our methods obtain these rates without the problem-parameter-based tuning, whereas the stepsizes in [1] depend on problem-specific constants.
> >
> > Moreover, we also compare the dependence on problem-specific parameters in the convergence bounds with [1] in Remark 4.8. To further complement the theory, we include additional experiments in Figures 7 and 8 of Appendix F.6, where we evaluate both our method and [1] under varying levels of the key problem-dependent constants. The results show that our approach remains stable and performs competitively across all tested settings, indicating resilience to variations in these problem-specific parameters even though our algorithm does not rely on them for tuning. Together, these theoretical and empirical comparisons demonstrate that our method matches the convergence guarantees of [1], achieves the same linear speedup, and additionally offers the advantage of operating in a problem-parameter-free manner.
> >
> > **References:**
> >
> > [1] Momentum for the win: Collaborative federated reinforcement learning across heterogeneous environments.
> >
> > [2] FedCM: Federated learning with client-level momentum.
> >
> > [3] Momentum benefits non-iid federated learning simply and provably.
> >
> > **Thank you once again for your thoughtful review and constructive feedback.**

---

### Official Review · Reviewer_MNvV · 2025-10-31

**Soundness:** 3
**Presentation:** 3
**Contribution:** 2
**Rating:** 4
**Confidence:** 4

**Summary:**

The paper proposes a “problem-parameter-free” FRL framework with two instantiations: PFEDPG-VR (variance-reduced policy gradient) and PFEDPG-HA (adds Hessian-vector corrections). Core ingredients are: (i) client & server control variates, (ii) client-side momentum, and (iii) per-step normalization (fixed-length local updates). The stepsizes (η,λ,β) are set as functions of (N,K,T), avoiding unknown smoothness/variance/heterogeneity constants. The theory shows convergence to an ε-stationary point with sample complexity O(ε−3) and communication O(ε−2), with linear speedups in N and K. Experiments on random tabular MDPs and MuJoCo (CartPole, HalfCheetah) report moderate gains and robustness to learning-rate choice.

**Strengths:**

* Clear engineering of several stabilizing tricks (control variates, momentum, normalized steps) into a coherent FRL algorithm.

* No heterogeneity magnitudes are required to set hyperparameters, unlike prior FRL that bake heterogeneity bounds into stepsizes. The rates still hold without “neighborhood” convergence caveats.

* The analysis preserves linear gains in agents and local updates while keeping communication as commonly observed in FRL works. The proofs seem clear and easy to follow (I haven’t checked all of them in full detail but generally seem OK)

**Weaknesses:**

* The algorithms recombine known components already explored in FRL/RL (client momentum, control-variate corrections akin to SCAFFOLD, Variance reduction, Hessian-aided updates) and the paper seems combinatorial. The headline rates (sample O(ε−3), communication O(ε−2), linear N,K speedups) are the same prior work which is underwhelming; the main sell/novel bit is “tuning-free” hyperparameters depending only on (N,K,T). It is not very valuable in practice since the learning rates are generally “tuned” as the upper bounds are generally loose.

* No heterogeneity bounds claim feels oversell. The analysis still assumes bounded log-gradients/Hessians (Assump. 4.1), bounded gradient variance (4.2), and bounded IS-weight variance (4.3). In heterogeneous RL, IS weights can blow up; assuming a global bound is strong. Assumption 4.3 also seems particularly strong and i guess will generally be false without restricting the proximity of $\theta_1,\theta_2$, because importance weight can have unbounded or heavy–tailed variance when policies place near–zero mass on actions supported by the other policy. It would be better to cite papers making the assumption next to them and not collectively.
In the theorems you called \hat{L} a constant that depends on H, \gamma, R_{max} etc however I feel the dependence should be explicitly shown as a part of sample complexity bounds as done in most of RL and FRL works cited in your paper.

* Since the whole sell of the paper (strength no 2) is not having to tune learning rates can it becomes important to discuss how you lose on methods which allow tuning in terms of rates/upper bounds and how you lose when the heterogeneity is high, however the paper only says they are equivalent to works which allows tuning in terms of their dependence on $epsilon$, I would encourage the authors to atleast discuss the dependence on heterogeneity parameters like $\sigma$, IS weight variance compared to prior works since they have a same rate in $\epsilon$. I imagine you might lose out on some performance gains compared to these methods which allow environment adaptive learning rate tuning.

* There are some places where the notation and sometimes theory has some minor typos that cause confusion: For eg. in thm 1 you set the learning rate to 1/\sqrt{KT} but in table 1 it become 1/KT, Several places jump from conditional E[⋅∣\mathcal{F}_t]to unconditional without comment. Add a subscript $t$ each time. It's otherwise hard to follow downstream which expectations are conditional. Some polishing will help.

**Questions:**

* If the same learning rate schedules work across different heterogeneity conditions, you might definitely lose on performance compared to methods that do allow manual hyperparameter tuning as commonly done in practice. This is not discussed in the paper. Depending on the discussion, one might be able to use the rates suggested here as a good starting point, irrespective of environment heterogeneity conditions followed by tuning to improve performance (as it is a common practice). Such a discussion will be beneficial to assessing whether the results are interesting in a federated RL context beyond combining gains from commonly known ideas in the literature like Importance sampling, variance reduction (STORM), control variates.

* I understand that the paper is mainly theoretical, but I have a question about the experimental results presented. The baselines appear to use fixed LRs (e.g., 0.005/0.6), while your method is auto-set by theory. Please sweep baselines over sensible grids and report best-of-sweep. Is that being done and the best curves are being reported here?

---

> ### Author Response · Authors · 2025-11-20
> **Point-to-Point Responses**
>
> **We sincerely appreciate the reviewer for recognizing our contributions and for the constructive comments. Our point-to-point responses to concerns on Weaknesses and Questions are given below.**
>
> **Reply to Weaknesses:**
>
> **1.** Thank you for your thoughtful comment. While prior methods have explored momentum, control variates, variance reduction, and Hessian-aided updates individually, our contribution is not the components themselves but the **problem-parameter-free FRL framework** that enables these mechanisms to work together without relying on tuning based on unknown problem parameters. This requires both new algorithmic design and new theoretical tools, allowing our method to operate reliably under environment heterogeneity. This framework is achieved through the careful integration of three essential components:
>
> - **Gradient normalization** acts as an adaptive stepsize mechanism that automatically adjusts the effective learning rate to the local landscape. It takes larger steps when the gradient is small (flat regions) and smaller steps when the gradient is large (steep regions), eliminating the need for manual tuning.
> - **Client-side momentum** accelerates convergence while stabilizing updates by accumulating gradients across agents and rounds.
> - **Control variates** align local updates with the global objective, mitigating drift and variance across heterogeneous agents.
>
> This integration eliminates the bounded-environment heterogeneity assumptions (such as the bound on differences in reward functions, transition kernels, or initial-state distributions) required in prior works [1]-[3] and achieves strong theoretical and empirical performance without knowledge of smoothness constants or other problem-specific parameters. Importantly, achieving this requires **fundamentally new theoretical analysis**, not simply assembling known ideas. Specifically:
>
> - Handling normalized gradients: Gradient normalization introduces mathematical complexities in tracking update dynamics, requiring new techniques to analyze its interaction with momentum.
>
> - Momentum analysis: Momentum creates coupling across updates; we develop new bounds to capture its interaction with control variates.
>
> - Problem-Parameter-free guarantees: We prove convergence without relying on problem-dependent constants, using novel tools to show that normalization compensates for the absence of tuned stepsizes.
>
> These analytical innovations go beyond a simple combination of existing tools. We have clarified these contributions more thoroughly in Section 5 of the revised paper.
>
> The headline rates—sample complexity $O(\epsilon^{-3})$, communication complexity $O(\epsilon^{-2})$, and linear $N,K$ speedups—represent the best-known results in FRL works [4]–[5]. Importantly, we would like to clarify that unlike prior methods that require extensive trial-and-error tuning, our approach provides **usable, theory-driven default stepsizes** that depend only on system-level constants $(N, K, T)$ and do not require estimating unknown problem parameters. Empirically, these theory-prescribed stepsizes achieve competitive and often superior performance compared with baselines whose learning rates are tuned through grid search.
>
> **2.** Thank you for your valuable comment. We would like to clarify that we only eliminate the need to assume bounded environment discrepancies across agents (i.e., no explicit bounds on differences in reward functions, transition kernels, or initial-state distributions) rather than no heterogeneity bounds. This aligns with the concept used in the existing heterogeneous-environment FRL work [5], and contrasts with earlier FRL studies [1]–[3] that require such environment heterogeneity bounds. We have stated this explicitly in the paper to avoid confusion.
>
> Regarding Assumptions 4.1–4.3, these are standard and well-established in policy-gradient analyses. Specifically, Bounded log-gradients/Hessians (Assumption 4.1) have been used in [5]-[9], [11]; Bounded gradient variance (Assumption 4.2) has been used in [5]-[9], [11]; Bounded IS-weight variance (Assumption 4.3) has been used in [5]-[7], [9]-[10]. We have now clearly cited these references next to each assumption in the revised version.
>
> We agree with you that in in policy-gradient RL, importance weights can become heavy-tailed when policies place near-zero probability on actions favored by others. However, we clarify that even in centralized RL settings, the bounded IS-weight variance assumption is widely used to ensure stability and tractable analysis [6]–[7], [9]–[10], and it remains a standard assumption in federated RL with heterogeneous environments as well [5]. We therefore follow this well-established analytical practice, consistent with existing RL theory.
>
> For clarity, the convergence bounds with explicit constant dependencies, consolidated from our theoretical analysis, are stated in Theorems 4.4 and 4.5 in the paper.

---

> > ### Author Response · Authors · 2025-11-20
> > **Point-to-Point Responses**
> >
> > **3.** Thank you for your insightful comment. We address this comment both theoretically and empirically.
> >
> > ### (i). Theoretical Comparison
> >
> > From the theoretical perspective, we already characterize these effects explicitly. In Theorem 4.4, our bound exhibits *linear* dependence on the gradient variance $\sigma$ and *square-root* dependence on the IS-weight variance bound $W$. In Theorem 4.5, the dependence remains *linear* in $\sigma$ and becomes *1/4-root* in $W$. In comparison, [5] achieves *cube-root* dependence on both $\sigma$ and $W$, whereas [12] attains *linear* dependence on $\sigma$ and *1/6-order* on $W$, but its convergence rate is slower than ours. These methods, however, require problem-parameter-dependent stepsize balancing, which relies on knowledge of problem-specific constants. Our approach, by contrast, is problem-parameter-free and removes environment-level heterogeneity bounds, which provides robustness without needing access to such constants.
> >
> > The linear (or square-root) dependence in our results arises because gradient normalization fixes the effective step length without requiring unknown problem parameters, and client-side momentum aggregates updates without the oracle-style balancing that compresses constant factors. This behavior is thus inherent to problem-parameter-free methodologies and aligns with what is commonly observed in related problem-parameter-free methods (e.g., [13]-[15]). Importantly, our convergence bounds share the best-known leading order in $T$ (with linear speedups in $N$ and $K$), so for moderately large $T$ the asymptotic regime dominates and the practical impact of having slightly higher-root constants diminishes rapidly. Given the practical advantages of eliminating environment-adaptive or problem-specific tuning, this modest increase in constant dependence is widely considered acceptable in the current literature. We have incorporated the relevant discussion—covering how our convergence bounds depend on all problem constants—into Remark 4.8 in the revised paper.
> >
> > ### (ii). Empirical Validation
> >
> > To complement the theory, we have also added new experiments in Appendix F.6 to examine how variations in these problem constants relate to performance in practice.
> >
> > - **Gradient variance ($\sigma$)**. In Figure 7, we adjust the number of sampled trajectories per local iteration, which can influence the gradient variance. Across this range, our method continues to perform competitively and shows stable behavior relative to tuned baselines.
> >
> > - **IS-weight variance**. In Figure 9, we vary the heterogeneity level of the reward function, which increases policy discrepancy across agents. According to Lemma C.2 in Appendix C of the paper, the IS-weight variance is bounded by the squared distance between policy parameters. Therefore, larger parameter differences lead to higher IS-weight variance during training. The results show that our method remains stable and maintains performance advantages even as heterogeneity grows.
> >
> > - **Other problem constants ($L$, $L_g$, $C_g$)**. We further include a reward-scaling experiment, where the reward is multiplied by a factor $\zeta>0$ during training. This preserves the optimal policy while scaling the reward bound $R_{\max}$ and consequently the constants $L = H R_{\max} {(M + H G^2)}/{(1 - \gamma)}$, $L_g = H M R_{\max} / (1 - \gamma)$, and $C_g = H G R_{\max} / (1 - \gamma)$. By training all methods under scaled rewards and evaluating them under the original environment, we directly assess robustness to changes in these reward-dependent constants. It can be observed in Figure 8 that our method remains stable across different scales, indicating resilience to variations in $L$, $L_g$, and $C_g$.
> >
> > These additional empirical results support our theoretical claims and demonstrate that, despite slightly higher dependence on certain constants, our problem-parameter-free approach remains robust and competitive across a wide range of problem settings.
> >
> > **4.** Thank you for your helpful comment. We have corrected the typo, added some clarifications, and refined the relevant formulas where appropriate to improve clarity. We appreciate your suggestion, and we are taking care to ensure that the derivations are presented in a way that facilitates understanding.

---

> ### Author Response · Authors · 2025-11-20
> **Point-to-Point Responses**
>
> **Reply to Questions:**
>
> **1.** Thank you for your insightful question. We would like to clarify that across different levels of heterogeneity, our theoretically guided learning-rate settings consistently achieve the best-known convergence order. Since our convergence bounds are derived in an order sense, multiplying the prescribed stepsizes by small constant factors does not change the theoretical rate—these factors affect only the multiplicative constants while preserving the same asymptotic rate.
>
> To examine this effect empirically, we conducted additional experiments under varying levels of reward heterogeneity and re-ran our method using several nearby learning-rate choices. The results (reported in Figure 10 of Appendix F.6) show that when we make small adjustments to the scalar multipliers of our learning rates, performance shows mild variation, which suggests that a slight refinement of our learning rates can modestly improve performance at negligible tuning cost. This supports that our theoretically prescribed learning rates can indeed serve as strong default initializations, and if practitioners wish to tune further, they may obtain minor refinements with only minimal extra effort.
>
> Moreover, the added experiments in Figures 7–9 (Appendix F.6), which vary key problem-dependent constants, further support our claim. In these experiments, our method uses only the theory-prescribed learning rates, while all baseline methods use grid-searched learning rates. Despite this, our method still remains competitive or superior across all tested settings. This demonstrates that our problem-parameter-free stepsizes are not only theoretically justified but also practically useful even compared with baselines that rely on manual tuning.
>
> In summary, all the added experiments further demonstrate that our contribution goes beyond a simple combination of known components, and that the resulting problem-parameter-free framework remains stable across different heterogeneity levels while performing reliably without problem-dependent tuning.
>
> **2.** Thank you for your thoughtful question. The baseline learning rates shown (e.g., 0.005/0.6) are not fixed a priori—they are the best-of-sweep values obtained by a grid search for the $N=5, K=10$ setting. When the configuration changes—such as varying $N$, $K$, or the problem-dependent constants in our additional experiments—we re-run the grid search under the new configuration and report the best baseline performance for that setting (in some cases even exceeding the results reported in the original papers). The same protocol is applied to all ablation variants: for each variant we perform a separate sweep and report its best curve.
>
> **References:**
>
> [1] Federated reinforcement learning with environment heterogeneity.
>
> [2] Fedkl: Tackling data heterogeneity in federated reinforcement learning by penalizing kl divergence.
>
> [3] Federated temporal difference learning with linear function approximation under environmental heterogeneity.
>
> [4] Stochastic policy gradient methods: Improved sample complexity for fisher-non-degenerate policies.
>
> [5] Momentum for the win: Collaborative federated reinforcement learning across heterogeneous environments.
>
> [6] Stochastic variance-reduced policy gradient.
>
> [7] Sample efficient policy gradient methods with recursive variance reduction.
>
> [8] Hessian aided policy gradient.
>
> [9] An improved analysis of (variance-reduced) policy gradient and natural policy gradient methods.
>
> [10] Learning bounds for importance weighting.
>
> [11] Adaptive step-size for policy gradient methods.
>
> [12] Fault-tolerant federated reinforcement learning with theoretical guarantee.
>
> [13] Two sides of one coin: the limits of untuned SGD and the power of adaptive methods.
>
> [14] Problem-parameter-free decentralized nonconvex stochastic optimization.
>
> [15] Tuning-free stochastic optimization.
>
> **Thank you once again for your thoughtful review and constructive feedback.**

---

### Author Response · Authors · 2025-11-29
**Summary of Rebuttal (Part II of II)**

### **1. Clarifying our main contribution and framework**

To better highlight the novelty, we explicitly state that our main contribution is the **problem-parameter-free and environment-heterogeneity-agnostic FRL framework**:

In Section 3.1 and the discussion in Section 5, we now clearly explain how

- Gradient normalization decouples step length from unknown problem scales and enables problem-parameter-free stepsizes,
- Client-side momentum stabilizes and accelerates learning across agents, and
- Control variates mitigate drift and variance under heterogeneous environments.

We would like to emphasize that these components are not merely combined, but are carefully designed to work together to address key challenges in FRL. In particular, our deliberate design ensures that the stepsizes depend solely on known system parameters (e.g., number of agents $N$, number of local updates $K$, total communication rounds $T$)—rather than unknown problem-specific parameters—while still handling arbitrary environment heterogeneity. This makes our distinction from prior FRL methods more explicit, both conceptually and algorithmically.

### **2. Strengthening and detailing our theory**

To make the theory clearer and more interpretable:

- In Theorems 4.4 and 4.5, we now explicitly show the dependence on all problem-specific constants (e.g., $L, L_g, C_w, C_g, \sigma$).
- We added Remark 4.8, which
  - compares our constant dependence with related FRL works, and
  - explains the widely accepted trade-off in problem-parameter-free methods: one typically accepts slightly higher dependence on certain constants in order to avoid problem-parameter-based tuning. This trade-off is consistent with existing problem-parameter-free optimization literature, and since its practical impact diminishes rapidly for moderately large $T$, it is widely regarded as reasonable and well justified given the practical advantages.

We also clarified the **fundamentally new proof techniques** required in our setting. In particular, our theoretical analysis must
- handle normalized gradients and carefully track their interaction with momentum,
- control the coupling between momentum and control variates across rounds and agents, and
- establish convergence with stepsizes that do not rely on any problem-dependent constants.

These clarifications highlight that achieving problem-parameter-free stepsizes under arbitrary environment heterogeneity—while matching the best-known convergence rates—requires nontrivial new theoretical analysis, goes beyond a direct rearrangement of existing components, and addresses key challenges in FRL.

### **3. Additional experiments to further validate robustness of our problem-parameter-free stepsizes**

To further support our theoretical claims and validate that our problem-parameter-free stepsizes are practically robust, we added experiments under varying problem settings in Appendix F.6:

- **Gradient variance:** We vary the number of sampled trajectories per local update (Figure 7), thereby changing the gradient variance, and illustrate that our method remains stable and competitive relative to tuned baselines.

- **Reward scaling ($R_{\max}$) and its dependent constants:** By scaling the reward during training (Figure 8), we change $R_{\max}$, which in turn could scale its associated constants $L, L_g, C_g$. This tests whether our stepsizes remain robust when these problem-dependent constants vary. Our method consistently exhibits stable and strong performance across all scales, again compared to tuned baselines.

- **Reward heterogeneity:** We introduce controlled reward heterogeneity via potential-based shaping (Figure 9), which increases policy discrepancy and could lead to higher importance sampling weight variance. Our method still shows stable and superior performance under these conditions. In addition, we perform learning-rate sensitivity studies under different heterogeneity levels (Figure 10), which further highlight the robustness of our theory-prescribed stepsizes.

Taken together, these additional experiments show that our problem-parameter-free design remains robust across diverse and increasingly challenging problem settings, and that our theoretically derived problem-parameter-free stepsizes are not only analyzable but also practically effective.

### **4. Other improvements**

We have reported all results as mean ± standard error. We also improved notation and refined several explanations to provide a clearer and more technically transparent presentation throughout the paper.

We sincerely hope that these efforts have further clarified the value of our work and are helpful for your evaluation. Thank you once again for your time and dedication throughout this process.

Sincerely,

Authors of the paper

---

### Author Response · Authors · 2025-11-29
**Summary of Rebuttal (Part I of II)**

Dear Area Chair,

We sincerely appreciate your efforts in handling our submission and thank you for your time and dedication throughout this process. In light of the current review procedure, we would like to provide a concise summary of our rebuttal, which we hope will be helpful for the evaluation of our work.

In the reviews, we were encouraged that the reviewers recognized the importance of our work, i.e., a novel problem-parameter-free FRL framework that eliminates the need for manual hyperparameter tuning, addresses a genuine practical problem in FRL, and handles arbitrary environment heterogeneity, supported by clear and well-structured proofs. Reviewers also acknowledged that our methods achieve state-of-the-art convergence rates with linear gains in the number of agents and local updates, and that the paper excels in methodological innovation, theoretical integrity, and experimental completeness.

In response to the reviewers' valuable comments, we have carefully considered all the suggestions and made significant efforts to address them, which we briefly summarize below.

---

### Meta-Review · Area_Chair_fSKY · 2025-12-27

**Summary:**

This paper proposes a problem-parameter-free federated reinforcement learning (FRL) framework whose goal is to eliminate manual step-size tuning under heterogeneous environments. The four reviewers broadly agreed the problem is practically meaningful and that this paper provides a coherent framework combining gradient normalization-based step-sizes, client-side momentum, and control variates. This paper also provides convergence guarantees and empirical gains on standard benchmarks. However, the reviewers raised several key concerns or weaknesses including (i) the novelty of the algorithmic contributions which could be read as a careful combination of known components including adaptive step-sizes with momentum and variance reduction; (ii) the convergence bounds still involve problem-specific quantities or hide them in constants, which are not as claimed to be problem-parameter-free; and (iii) concerns on the limited breadth of evaluation, mostly common benchmark settings, unclear marginal value of individual components from ablations, and missing statistical reporting, e.g., mean±std over seeds and confidence intervals.

**Reviewer Concerns:**

Regarding the concerns on the novelty of the algorithmic contributions, in the rebuttal, the authors explicitly framed their approach as relying on gradient normalization for step-sizes, momentum for stabilization under heterogeneity, and control variates to mitigate drift/variance. The authors argued that these are jointly necessary for the FRL setting.  Even with those clarifications, I would expect that at least one of the lower-score reviewers to remain uncertain that the main conceptual step on normalization to adaptive step-sizes, plus momentum/control-variate integration is sufficiently distinct from prior "adaptive, parameter-free" optimization transferred into FRL. This is more about positioning and crisp novelty statement than correctness.

Regarding the convergence bound to still depend on problem-specific quantities, the authors argued that any remaining constant dependence is an accepted tradeoff in "parameter-free" literature, while still removing the tuning requirement in practice.

In the rebuttal, the authors also indicated they can report mean±std or standard error and clarify this in the revision.  One reviewer questioned whether removing the control variate changes performance only marginally. While the rebuttal argues that it is essential for theory and for difficult regimes, the empirical case may still feel mixed without broader stress tests.

**Reviewer Scores:**

Reviewer MNvV flagged novelty and constant-dependence and some clarity issues. The rebuttal’s clearer novelty positioning and additional experiments would plausibly move them up, though probably not higher because the "is it fundamentally new?" doubt may remain.

Reviewer 78RH was positive on significance and liked the parameter-free framing, while still raising "unclear algorithmic contributions and ablation" and comparison questions. With the direct clarifications and added robustness studies, I’d expect them to stay positive if the revised revision clearly disentangles contributions and includes the requested comparisons.

Reviewer hFbc's comments were superficial and lacked technical depth.

Reviewer wY6d liked the problem and results but raised novelty concerns and worries about dependence on problem parameters in the bounds. The rebuttal’s explicit comparison or justification and additional experiments might help, but I’d still expect only a one-notch move absent a very crisp rewritten related-work and positioning section.

---

### Decision · Program_Chairs · 2026-01-26

Reject